# Single-cell transcriptomic analysis reveals diversity within mammalian spinal motor neurons

Ee Shan Liau [1,2,10], Suoqin Jin [3,4,10], Yen-Chung Chen [2], Wei-Szu Liu[2], Maëliss Calon [5,6,7], Stéphane Nedelec [5,6,7], Qing Nie [4,8,11] ✉ & Jun-An Chen [1,2,9,11] ✉

Spinal motor neurons (MNs) integrate sensory stimuli and brain commands to generate movements. In vertebrates, the molecular identities of the cardinal MN types such as those innervating limb versus trunk muscles are well elucidated. Yet the identities of finer subtypes within these cell populations that innervate individual muscle groups remain enigmatic. Here we investigate heterogeneity in mouse MNs using single-cell transcriptomics. Among limb-innervating MNs, we reveal a diverse neuropeptide code for delineating putative motor pool identities. Additionally, we uncover that axial MNs are subdivided into three molecularly distinct subtypes, defined by mediolaterally-biased Satb2, Nr2f2 or Bcl11b expression patterns with different axon guidance signatures. These three subtypes are present in chicken and human embryos, suggesting a conserved axial MN expression pattern across higher vertebrates. Overall, our study provides a molecular resource of spinal MN types and paves the way towards deciphering how neuronal subtypes evolved to accommodate vertebrate motor behaviors.

Motor behaviors are fundamental in animals, enabling basic survival skills ranging from fight-or-flight to the enormous repertoire of movements underlying complex social interactions. These movements are controlled by spinal cord neurons. Motor neurons (MNs) are the final hub that conveys commands from the central nervous system (CNS) to peripherals. MNs project their axons to target muscles and coordinate muscular contractions. To support the vast heterogeneity of muscular functions, MN diversity is generated during embryonic development to establish precise synaptic and motor circuit connections[1]. Thus, MNs are organized as columnar and pool subtypes, stereotypically positioned along the spinal cord axis (Fig. 1a).

Columnar MNs are spatially grouped according to the body regions they project to. For example, lateral motor column (LMC) MNs are present in the brachial and lumbar spinal cord to control limb muscle movements, whereas medial motor column (MMC) MNs span along the entire rostrocaudal axis of the spinal cord and control epaxial muscles to support body posture (Fig. 1a). Within these columnar MNs, MNs that innervate individual muscle groups can be further categorized into subtypes known as motor pools. More than 40 limb muscle groups have been described in a typical amniote[2], implying that a matching number of motor pool subtypes exist for limbs alone. Although previous studies have systematically mapped the positions

[1]Molecular and Cell Biology, Taiwan International Graduate Program, Academia Sinica and Graduate Institute of Life Sciences, National Defense Medical Center, Taipei 11529, Taiwan. [2]Institute of Molecular Biology, Academia Sinica, Taipei 11529, Taiwan. [3]School of Mathematics and Statistics, Wuhan University, 430072 Wuhan, China. [4]Department of Mathematics, University of California, Irvine, Irvine, CA 92697, USA. [5]Institut du Fer à Moulin, 75005 Paris, France. [6]Inserm, UMR-S 1270, 75005 Paris, France. [7]Science and Engineering Faculty, Sorbonne Université, 75005 Paris, France. [8]Department of Developmental and Cell Biology, University of California, Irvine, Irvine, CA 92697, USA. [9]Neuroscience Program of Academia Sinica, Academia Sinica, Taipei, Taiwan. [10]These authors contributed equally: Ee Shan Liau, Suoqin Jin. [11]These authors jointly supervised this work: Jun-An Chen, Qing Nie. ✉e-mail: qnie@uci.edu; jachen@imb.sinica.edu.tw

of motor pools along the spinal cord according to the position of muscles they innervate, a comprehensive molecular profile of motor pools is still lacking. Furthermore, studies on motor pool diversity have primarily focused on LMC MNs, leaving other columnar MNs underappreciated.

Previous studies have identified several genetic regulators of motor pools in LMC MNs and have shed light on the molecular mechanisms that determine MN identities and how motor pools regulate their target connectivity[1]. These studies suggest that each motor pool is defined by the combinatorial expression of transcription factors (TFs)—including Hox proteins and others such as *Etv4* (*Pea3*), *Runx1* or *Pou3f1* (*Scip*)—and genes encoding axon guidance ligands and receptors[1]. Some motor pool-specific TFs dictate the activity of downstream effector genes[3]. Misexpression of motor pool genes has been shown to alter different stages of the MN differentiation and maturation process, including identity specification, axon arborization and dendritic patterning, as well as axon navigation and circuit formation[4,5]. However, only a few motor pool genes have been identified to date. As a result, apart from a few TFs and guidance cues, the molecular repertoire of MN diversity remains largely unclear. Given that motor pool-specific gene expression drives distinct cell type-specific functions, we hypothesize that there might be a broader motor pool code that can be deciphered by means of comprehensive single-cell spinal MN transcriptomics. Identifying those pool codes would enable genetic targeting of a specific motor pool and reveal the underlying intrinsic or extrinsic mechanisms that establish cell type-specific properties and target connectivity.

Although MNs control muscle contraction in all animals, motor outputs can vary among species. Fish utilize epaxial MNs to facilitate their lateral undulatory swimming patterns, whereas tetrapods use limb MNs and axial MNs to coordinate appendicular muscles for joint movement and trunk muscles for posture maintenance, respectively[6]. Comparative analyses across different species have identified the origin of limb motion and MN subtypes in the LMC[7–9]. Nevertheless, although axial MNs representing ancestral MNs are present in both aquatic and land vertebrates, they support distinct motor behaviors in these two groups[7]. Currently, our understanding of MMC MNs is much more incomplete relative to LMC MNs. It has been hypothesized that MMC subtypes may innervate distinct epaxial muscles[10–12], but a systematic exploration of the molecular heterogeneity of MMC MNs is still lacking. A detailed study of axial MMC neuronal diversity is critical to investigations of how MNs have diversified during vertebrate evolution and might also help decipher how neuronal subtypes have co-evolved with the distinct modes of locomotion displayed by aquatic and terrestrial vertebrates.

In this work, we profile MNs at the embryonic stage to understand the uncharted diversity of motor pools when MNs have differentiated and make selective synaptic connections to form motor circuits given that the two recent single-cell studies of spinal cholinergic neuronal diversity indicated that motor pool identities become subtler in adult mice[13,14]. As MNs account for less than 1% of the cells within the spinal cord[15], we use MN reporter mice (Mnx1-GFP) and establish a robust method to enrich for spinal MNs. Our study has generated a detailed atlas of MN transcriptional identities, including known and unreported segment-restricted MN subtypes. Within LMC MNs, we reveal transcriptional codes for 16 and 10 subclusters from brachial and lumbar limbs, respectively, which may correspond to putative motor pool groups. Moreover, we demonstrate unexpectedly diverse expression of neuropeptides in the LMC MNs. The discovery of underappreciated heterogeneity in axial MNs is validated by different marker expression patterns on distinct axonal branches. Finally, we have explored the MMC subtypes in other vertebrates, including humans. Our results serve as a valuable resource for investigating the multifaceted mechanisms underlying how cell-type-specific features and locomotor circuit development are established.

## Results

### Single-cell profiling of developing spinal MNs

The diverse motor behaviors of mammals are regulated by a rich repertoire of MNs distributed along the spinal cord (Fig. 1a). However, we still lack a comprehensive picture of spinal MN molecular heterogeneity in mammalian embryos. Therefore, we decided to perform single-cell RNA sequencing (scRNA-seq) on mouse spinal MNs at embryonic day 13.5 (E13.5) (Fig. 1b), a stage at which these neurons have acquired pool identities and established axon innervations into target muscles[16]. Given that MNs represent only a tiny fraction of all spinal cells, using the entire tissue as a source would not be an optimal approach for in-depth characterizations of MN diversity[15]. Thus, we isolated spinal cord tissues separately from the rostral and caudal spinal cord segments of a total 12 Mnx1-GFP mouse embryos from 2 pregnant mice in which GFP is expressed at a high level in MNs and in a small population of interneurons[17]. Using fluorescence-activated cell sorting (FACS), we collected cell populations with the highest GFP intensities (GFP^high) and subjected the samples to scRNA-seq (Fig. 1b and Supplementary Fig. 1a–c). As expected, by aligning our scRNA-seq samples using a customized GFP reference, we detected *GFP* transcripts in most cells in both samples (Supplementary Fig. 1d).

Although Mnx1 labels MNs and a population of spinal glutamatergic interneurons (IN) specifically in the spinal cord, we detected sparse *Mnx1* expression in our single-cell dataset, likely due to a limitation of scRNA-seq sensitivity. As spinal MNs are all cholinergic, we thus defined MNs according to their expression of both cholinergic (*Slc18a3*, *Chat*) and spinal MN marker (*Mnx1*), with spinal MNs accounting for ~91% of the total cells upon quality filtering (rostral: 5162 MNs from a total of 5460 cells; caudal: 4699 MNs from a total of 5242 cells) (Supplementary Fig. 2; see "Methods"). Only a tiny population of cells (rostral: 300 cells; caudal: 435 cells) sparsely expressed spinal IN markers (*En1*, *Pou4f1* or *Pax2*), which we excluded from further analyses (Supplementary Fig. 2c–f). Accordingly, our single-cell libraries cover a large number of MNs from brachial to sacral segments, allowing us to construct a more robust molecular atlas of spinal MNs at the embryonic stage.

### Identification of spinal MN subtypes and gene signatures

To uncover MN heterogeneity, we clustered single-cell transcriptomes from the rostral and caudal datasets using Seurat[18] and assigned cluster identities based on the averaged expression of canonical marker genes (see "Methods"). Our analysis resulted in a total of 13 clusters and we were able to assign cell identities to 10 of them (Fig. 1c–f): LMC lateral motor column (*Foxp1*+ and *Aldh1a2*+)[19], MMC medial motor column (*Mecom*+ and *Lhx3*+)[20], PGC preganglionic motor column (*Nos1*+ and *Zeb2*+), HMC hypaxial motor column (*Lhx3*−, *Foxp1*−, *Isl1*+), *Mnx1*+ excitatory interneurons with a glutamatergic (*Slc17a6*+) property[21], and nascent MNs (*Neurod2*^high and *Ebf2*^high)[22,23]. One cluster was denoted as "unknown" as we could not uncover a salient and characteristic gene set for it (Supplementary Fig. 3a). Nevertheless, identifying the major columnar MNs allowed us to characterize detailed MN subtypes within these columnar MNs. Unexpectedly, two clusters appeared to be segmentally restricted and previously unreported, prompting us to identify their distributions in vivo. We named them based on the markers they express: *Calb1*+ MNs (*Calb1* is selectively expressed in the rostral cluster), and the caudally-restricted *Nfib*+*Grm5*+ cluster in which *Nfib* and *Grm5* are the top markers (Fig. 2a). Extensive reports on Calb1 and Grm5 in other spinal cord cell types already exist[24,25], but their patterns of expression in MNs have not been examined in detail before. Therefore, we verified the distributions of these markers in the spinal cord to determine if these two transcriptional clusters correspond to specific MN subtypes.

For Calb1+ MNs, we noted that Mnx1, Calb1 and Chat co-expressing cells were located at the most ventromedial region of the spinal cord at the E13.5, E15.5, and postnatal day 7 (P7) stages (Fig. 2b, c). Moreover,

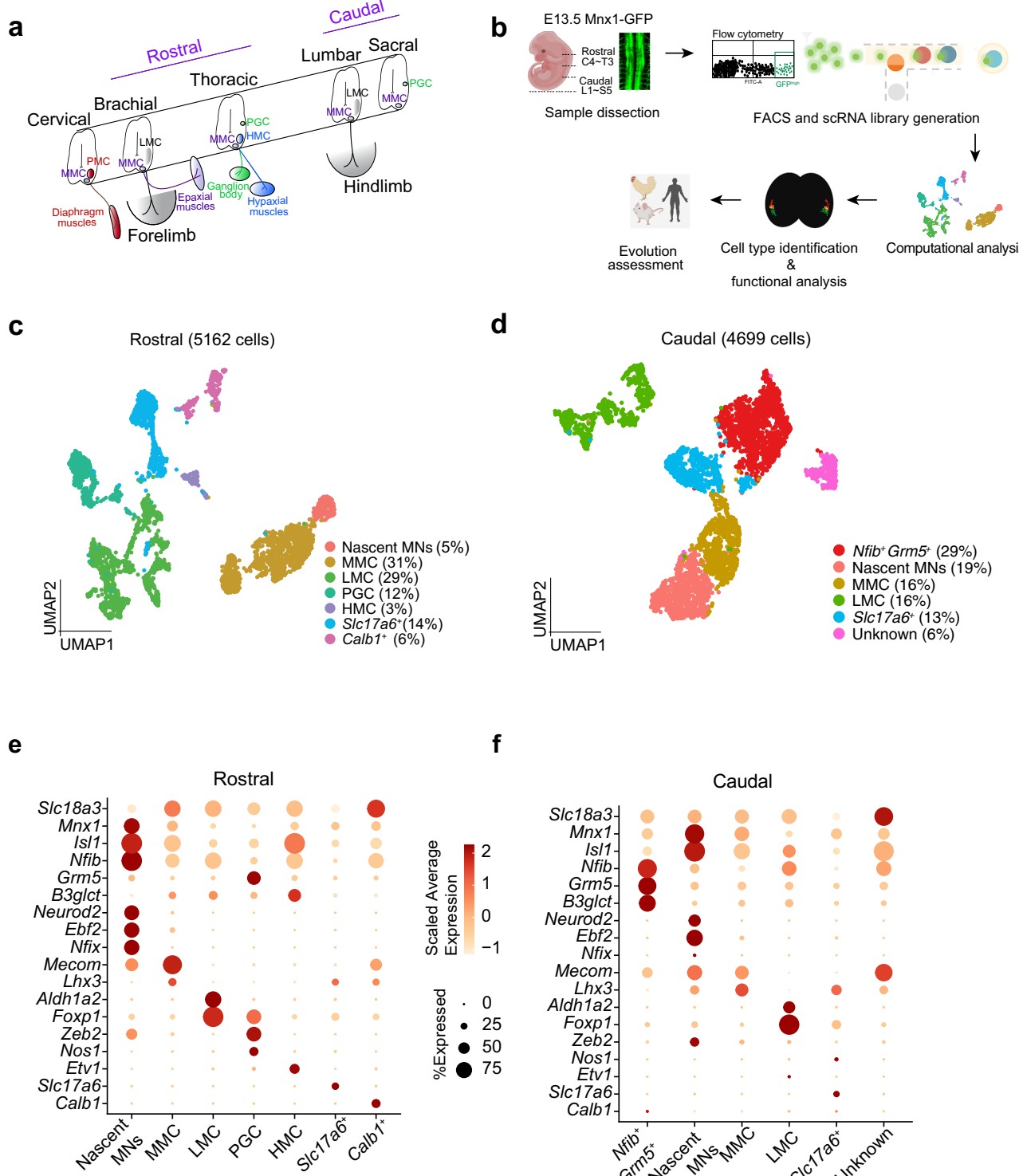

**Fig. 1 | Single-cell transcriptome profiling of E13.5 mouse spinal motor neurons.** **a** Illustration of MN subtypes innervating different muscle targets along the rostrocaudal axis of the spinal cord. Innervation targets and motor column are coded with the same color: PMC phrenic motor column (red), MMC medial motor column (violet), LMC lateral motor column (grey), PGC preganglionic motor column (green), HMC hypaxial motor column (blue). Samples for this study were collected from rostral and caudal segments. **b** Overview of the experimental workflow. **c**, **d** UMAP plots showing cellular heterogeneity of **c** rostral and **d** caudal samples after removing interneuron and non-neuronal cells. Cell populations are color-coded, and cell identities have been annotated based on the expression patterns of known and identified markers in this study. Percentages of each cell population are shown in parentheses. **e**, **f** Dot-plots showing expression patterns of known and identified marker genes (rows) in each cluster (columns) for **e** rostral and **f** caudal MNs. The size of each circle reflects the percentage of cells expressing the genes, and color intensity represents scaled average expression for each gene. Single-cell transcriptomes in this study were produced by collecting cells from 12 embryos of two Mnx1-GFP pregnant mice, pooled together to perform scRNAseq. Rostral and caudal segments were separated by dissection and processed independently.

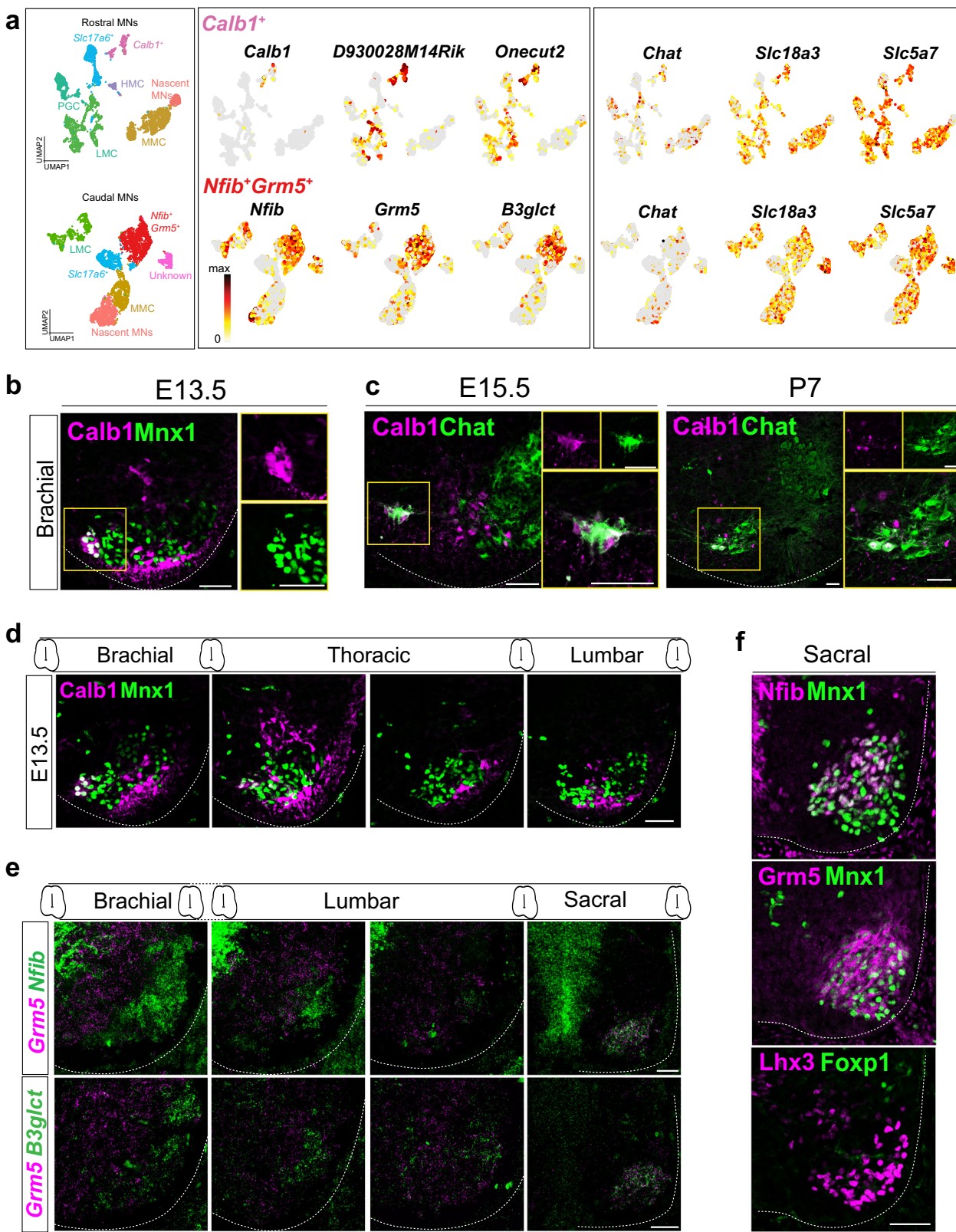

Calb1+ MNs were distinct from Calb1+-only Renshaw cells positioned ventrolateral to the MNs[24] (Fig. 2b). In agreement with our sequencing data (Fig. 1c–f), the Mnx1+Chat+Calb1+ MNs were only observed in the brachial and upper thoracic segments, implying their potential upper body muscle group innervations (Fig. 2d). In contrast, the Nfib+Grm5+ co-expressing neurons were only observed in the sacral spinal segments, with the expression of another top marker−B3glct. (Fig. 2e and

Supplementary Fig. 3b). We noticed that these cell populations were located in the vicinity of MMC MNs and were segregated from Foxp1+ LMC and PGC MNs (Fig. 2f). These observations correspond to our single-cell data showing that the two clusters exhibited slight transcriptomic similarity to MMC MNs (Supplementary Fig. 3c). Thus, we have validated the two previously unreported MN subtypes from single-cell data, which present different gene expression profiles to

**Fig. 2 | Characterization of Calb1⁺ and Nfib⁺Grm5⁺ MNs. a** UMAP visualization of all clusters (left), top three markers enriched in *Calb1⁺* and *Nfib⁺Grm5⁺* clusters (middle), and cholinergic genes (right). **b** Immunostaining reveals ventromedial positioning of the Calb1⁺Mnx1⁺ cells in the E13.5 mouse spinal cord. A high-magnification image is shown in the yellow square. **c** Immunostaining at E15.5 and P7 reveals that Calb1⁺ cells exhibit cholinergic identity by co-expressing Chat. High-magnification images are shown in the yellow square. **d** Calb1⁺Mnx1⁺ cells localize in the brachial and upper thoracic segments. **e** RNAscope-based fluorescence ISH of

*Nfib⁺Grm5⁺* cluster markers show that *Nfib⁺Grm5⁺* and *Grm5⁺B3glct⁺* coexpressing cells are present mainly in the sacral segments. **f** Immunostaining of Mnx1 with Nfib or Grm5 and Foxp1 with Lhx3 in adjacent sacral spinal cord sections (E13.5). Foxp1 and Lhx3 label sacral PGC and MMC MNs, respectively. Note that the Nfib⁺ Grm5⁺ cells are separated from sacral PGC MNs and partially overlap MMC MNs. Dashed lines outline the spinal cord boundary. Scale bars represent 50 μm. All immunostaining and in situ hybridization experiments were repeated on *n* = 3 embryos.

columnar MNs, such as the expression of *Calb1* or *Nfib* plus *Grm5*, and that they are anatomically clustered and localized to certain segments of the spinal cord. Given that Calb1⁺ neurons and sacral MNs are disease-resistant[26], these subtypes might also be important to MN disease and it would be of great interest in the future to determine their peripheral targets. We elaborate further on our findings in the Discussion.

In addition to uncovering MN subtypes, we endeavored to identify additional molecular signatures of known subtypes (Fig. 3a) by identifying differentially expressed genes (DEGs) among the known columnar MNs. HMC MNs were thought to lack a genetic marker[8], yet we have identified a series of marker genes for this subtype (Supplementary Data 1), including *Tac1* (which encodes the tachykinin peptide hormone family, neurokinin A, substance P, neuropeptide K and neuropeptide gamma), which was previously described as being expressed by pain-related projection neurons and interneurons in the spinal cord[27]. However, its expression has not been reported before for MNs. In situ hybridization validated the selective expression of *Tac1* in thoracic HMC MNs, representing a HMC genetic marker (Fig. 3b, right, Fig. 3d). Similarly, we verified *Kitl* as a marker of PGC MNs by in situ hybridization. *Kitl* encodes a ligand for the versatile Kit pathway involved in axon outgrowth, cell migration and cell survival[28], suggesting potential similar roles in PGC MNs. We also compared LMC and MMC neurons (Supplementary Fig. 4a, b). Apart from known genes (LMC: *Foxp1*; MMC: *Mecom/Lhx4/Lifr/Il6st*)[19,29], we uncovered selective expression of IgLON neural adhesion molecules (i.e., encoded by *Ntm* and *Lsamp*) within these neurons (Fig. 3b, left, Fig. 3c, Supplementary Fig. 4b, and Supplementary Data 2). Notably, *Ntm* expression discriminates LMC from *Lsamp*ʰⁱᵍʰ MMC neurons in the E13.5 brachial spinal cord (Fig. 3b, left, Fig. 3c). IgLON neural adhesion molecules are known to regulate axon pathfinding, dendritic arborization and synaptogenesis[30,31], and selective expression of the respective genes might contribute to the differences in neuronal pathfinding and circuit wiring between LMC and MMC MNs. Overall, our scRNA-seq data not only have identified most columnar MNs and revealed undescribed MN subtypes, but have also unveiled a list of markers for different subtypes that could potentially explain their physiological characteristics. Additionally, our results provide an extensive resource to facilitate the development of genetic tools that enable further molecular characterizations of specific subtypes in the future.

## Subtypes within LMC MNs

The identification of major columnar MNs within our MN dataset prompted us to uncover subtypes within these motor columns. Characterization of MN subtypes in the adult stage has revealed previously unreported heterogeneity in PGC MNs[13,14] and the embryonic origin of these subtypes remains unclear. Although we sampled spinal cord tissue mainly at the limb segments, we had included small populations of PGC and HMC MNs within our collected samples. In agreement with observations from a previous study reporting that differential *Isl1* expression could differentiate PGC MNs into PGCa and PGCb subtypes[32], we observed a similar pattern in our single-cell data (Fig. 1c and Supplementary Fig. 4c). However, in this study, our objective was to analyze LMC and MMC MN subtypes, so we did not endeavor to characterize the PGC subclusters further. We used LMCs as a proof-of-principle type to test if transcriptome-based subclusters

match certain known motor pools, as marker genes for some LMC motor pools have been revealed[33]. Accordingly, we clustered the 1,475 cells from brachial LMC MNs. Among brachial LMC MNs, Hoxa5/Hoxc8 proteins define rostrocaudal (rc) identities[3], whereas Isl1/Lhx1 distinguish their mediolateral (ml) positions[34] (Fig. 4a). Therefore, we grouped the cells into four "spatial quadrants" according to their expression of these four cardinal TFs (Fig. 4b, c, see "Methods"). We compared the DEGs between MNs in each "spatial quadrant." Comparison of the *Hoxa5⁺* and *Hoxc8⁺* clusters identified several Hox cofactors (*Pbx1*, *Pbx3*, *Meis1*, and *Meis2*) that are already known to be differentially expressed between rostral (*Hoxa5⁺*, *Hoxc8*) and caudal (*Hoxc8⁺*) MNs[35] (Supplementary Fig. 5a). We also observed enriched expression of *Nrp2* and *Unc5c* in the medial *Isl1⁺* cluster, consistent with previous studies[36,37] (Supplementary Fig. 5b). These results indicate that the assignment of LMC MNs into spatial quadrants corresponds well to in vivo distributions.

Next, we performed subclustering analysis separately on each spatial quadrant (namely, rm: rostral medial, rl: rostral lateral, cm: caudal medial, cl: caudal lateral) (Supplementary Fig. 5c). To determine a reasonable number of subclusters, we adopted a multi-resolution ensemble strategy based on spectral graph theory, which ensured that we captured distinct clusters at the transcriptomic level (Supplementary Fig. 5d, e and Supplementary Data 3, see "Methods"). Our analysis revealed 16 subclusters distributed across the four spatial quadrants (Fig. 4d and Supplementary Fig. 5c). To understand which gene types are diversely expressed in the LMC MNs, we performed ontology (GO) analysis on the top-ranked variable genes among these neurons. We observed that many of these genes encode TFs, cell adhesion and axon guidance molecules, receptors, and ion channels, many of which have been cited as having functions related to cell fate specification, motor pool sorting and positioning, axon pathfinding and synaptogenesis[38,39] (Fig. 4e and Supplementary Fig. 5f). Among the TFs which presented a strong cluster-specific pattern (Fig. 4f), we detected expression of known motor pool markers such as *Etv4* (*Pea3*) in the cm1/2 subcluster, *Runx1* in the rl2 subcluster, and *Pou3f1* (*Scip*) in the cm3 subcluster, corresponding to MNs innervating the pectoralis (pec)[40]/cutaneous maximus (CM)[41], scapulohumeralis (Sca) and flexor carpi ulnaris (FCU) muscles[3,42], respectively (Supplementary Fig. 6a). This result supports our hypothesis that LMC subclusters likely reflect motor pools, which motivated us to examine the remaining unaligned putative motor pools. First, we focused on previously unreported and specifically expressed TFs, since motor pool-specific TFs are usually the major determinants for establishing motor pool identity and properties[43]. As a proof of principle validation experiment, we chose *Zfhx4* that is selectively expressed in the rl1 and rl2 subclusters (Fig. 4f). We corroborated its expression in ~15% of LMC MNs in the rostral lateral region of the brachial segment (Fig. 4g, h), resembling the protein expression pattern of Zfh2 (a *Drosophila* ortholog of *Zfhx4*) in the late-born ventral nerve cord neurons of the fly[44]. We also further observed *Nr2f1* expression in subsets of LMC MNs (Supplementary Fig. 6b). Thus, profiling of our enriched MN population allowed us to recover LMC MN subtypes at an unprecedented resolution, to uncover previously uncharacterized subtypes that might innervate individual muscle groups and to suggest that the cluster-specific TFs they express might be important to establish their presumptive motor pool identity.

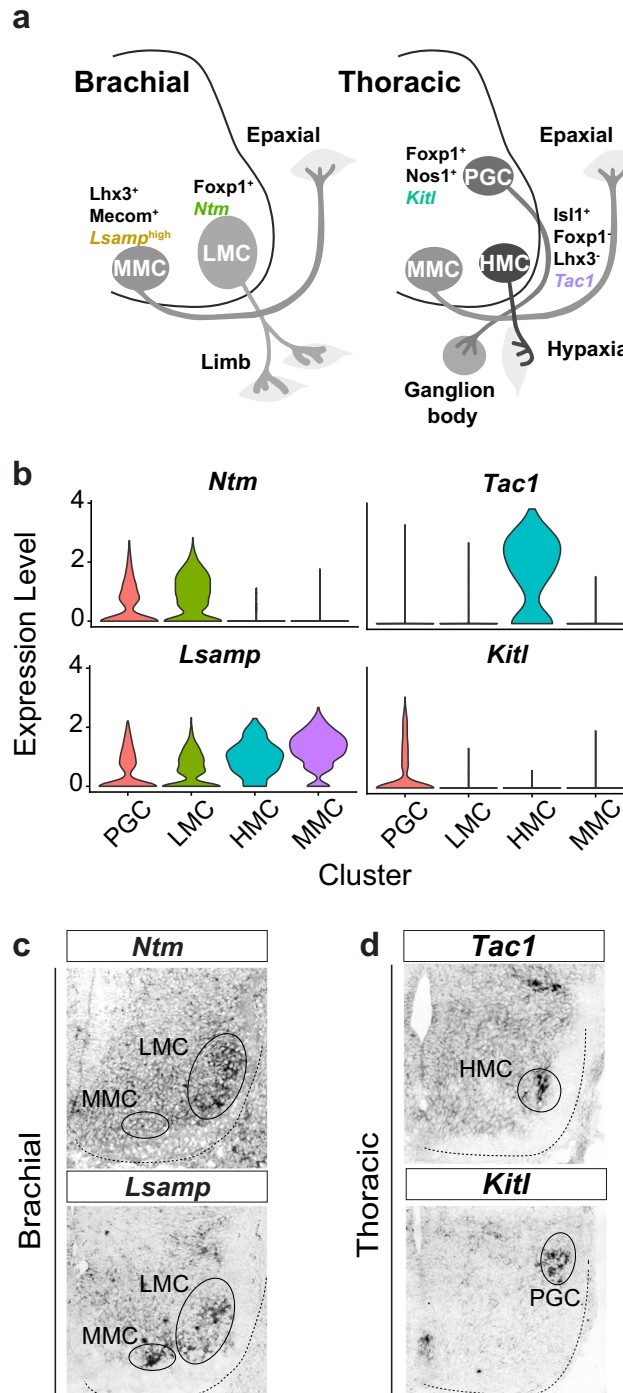

**Fig. 3 | Cluster identity assignment and identification of genetic markers.**
**a** Schematic summary of brachial and thoracic columnar MNs with their positions in the spinal cord and their targeting muscles. The known (black font) and the identified (colored font) molecular signatures in each columnar MN type are labeled. **b** Violin plots reflecting the expression of selected genes for each known MN cluster. **c** In situ hybridization (ISH) of cell adhesion markers identified for LMC (*Ntm*) and MMC (*Lsamp*) MNs. **d** ISH of *Tac1*- and *Kitl*-identified HMC and PGC MNs, respectively. In **c**, **d**, dashed lines outline the spinal cord boundary. Scale bar represents 50 μm. All in situ hybridization experiments were performed on *n* = 3 embryos.

### Unexpectedly diverse neuropeptide expression within LMC MNs

Upon checking the molecular pathways enriched for variable genes in LMC MNs, we noticed a GO term for neuropeptides (Fig. 4e), which are small proteins produced by neurons that usually act on G protein-coupled receptors. Neuropeptides can function via autocrine or paracrine signaling with neurotransmitters in a single neuron type to modulate and expand neuronal function[45]. Although neuropeptides have been used to categorize cell types in the brain and dorsal horn of the spinal cord[46,47], their expressions and functions in MNs have not been characterized. Using the Gini index to assess gene expression variability among LMC subclusters, we observed that neuropeptides are as variable as other reported gene families such as cell adhesion molecules and significantly more variable than housekeeping genes (Fig. 5a; the Gini index was not significantly correlated with either the number of genes within a gene set (correlation = −0.52, *p* value = 0.29), or the average expression levels within a gene set (correlation = −0.03, *p* value = 0.95)). Therefore, we anticipated that neuropeptides could be reliably used to distinguish LMC MNs. Similar to our observation for TFs, we validated a cohort of neuropeptides (*Npy*, *Sst*, *Grp*, *Cbln1*, *Cbln4*, *Pnoc*, and *Penk*) from our single-cell results and indeed observed that they are spatially partitioned in different subsets of LMC MNs using in situ hybridization (Fig. 5b, c and Supplementary Fig. 6c). Combinatorial neuropeptide expression patterns precisely distinguished LMC subtypes. For example, *Sst* and *Npy* co-expression distinguished the rm1 subcluster from the *Npy*-only rl4 subcluster (Fig. 5b, d and Supplementary Fig. 6d–g), whereas *Pnoc* and *Grp* co-expression delineated the cl3 subcluster from the *Pnoc*-only cl4 subcluster (Fig. 5b). Remarkably, we also detected that the *Etv4*+ subcluster (cm1) expressed *Sst* and *Grp*, whereas the *Pou3f1*+ subcluster (cm3) displayed *Penk* expression (Fig. 5e, f). Furthermore, we observed a significantly stronger correlation between the subcluster-specific TFs and neuropeptides compared to the correlation between randomly selected TFs and neuropeptides (Fig. 5g), implying that motor pool TFs in combination with neuropeptides may exert subtype-specific physiological roles that warrant future experimental validation. Strikingly, upon comparing our embryonic MN dataset to previously published adult MN scRNAseq results[13,14], we found that though the embryonic LMC subtype TF/neuropeptide code is largely distinct from that of adult MN subtypes, adult skeletal MN subtypes still manifest a different set of neuropeptide combinations, implying that the neuropeptides might function differentially in both embryonic and adult stages (Supplementary Figs. 7 and 8).

Finally, to investigate whether these specific neuropeptide-expressing motor pools target peripherals by means of selective innervations, we utilized Npy[Cre]; Tau-GFP reporter mice together with Mnx1-RFP to examine the axonal projection patterns of the *Npy*+ subcluster. As expected, we detected that the axons of Npy+ neurons represent only a subset of the MN nerves in the brachial and lumbar plexus (Fig. 5h). Overall, our scRNAseq results indicate that brachial LMCs manifest a complex array of subclusters reflecting a pattern of diverse neuropeptide expression, which might facilitate motor pools elaborating precise sensory-motor and neuromuscular connections. We present the implications of these findings in our Discussion.

### Convergence and divergence between brachial and lumbar LMC MNs

LMC MNs are present in both the forelimbs and hindlimbs[1], and some motor pools, such as *Etv4* subtype[48], are known to be present in both brachial and lumbar segments, indicating that there might be more convergent or divergent subclusters between the two segments. Accordingly, we performed an unbiased cross-segmental molecular comparison of these LMC MN subclusters with Harmony[49] to examine this hypothesis. We first compared gene expression in LMC MNs between brachial and lumbar segments and uncovered 112 DEGs, in which *Hox* genes were most prevalent (Fig. 6a and Supplementary Data 4). This outcome is consistent with previous studies showing that *Hox* genes are the crucial TFs defining segmental MN identities[3]. Additionally, GO analysis of the DEGs demonstrated that pathways—such as MN cell fate specification and development, insulin signaling

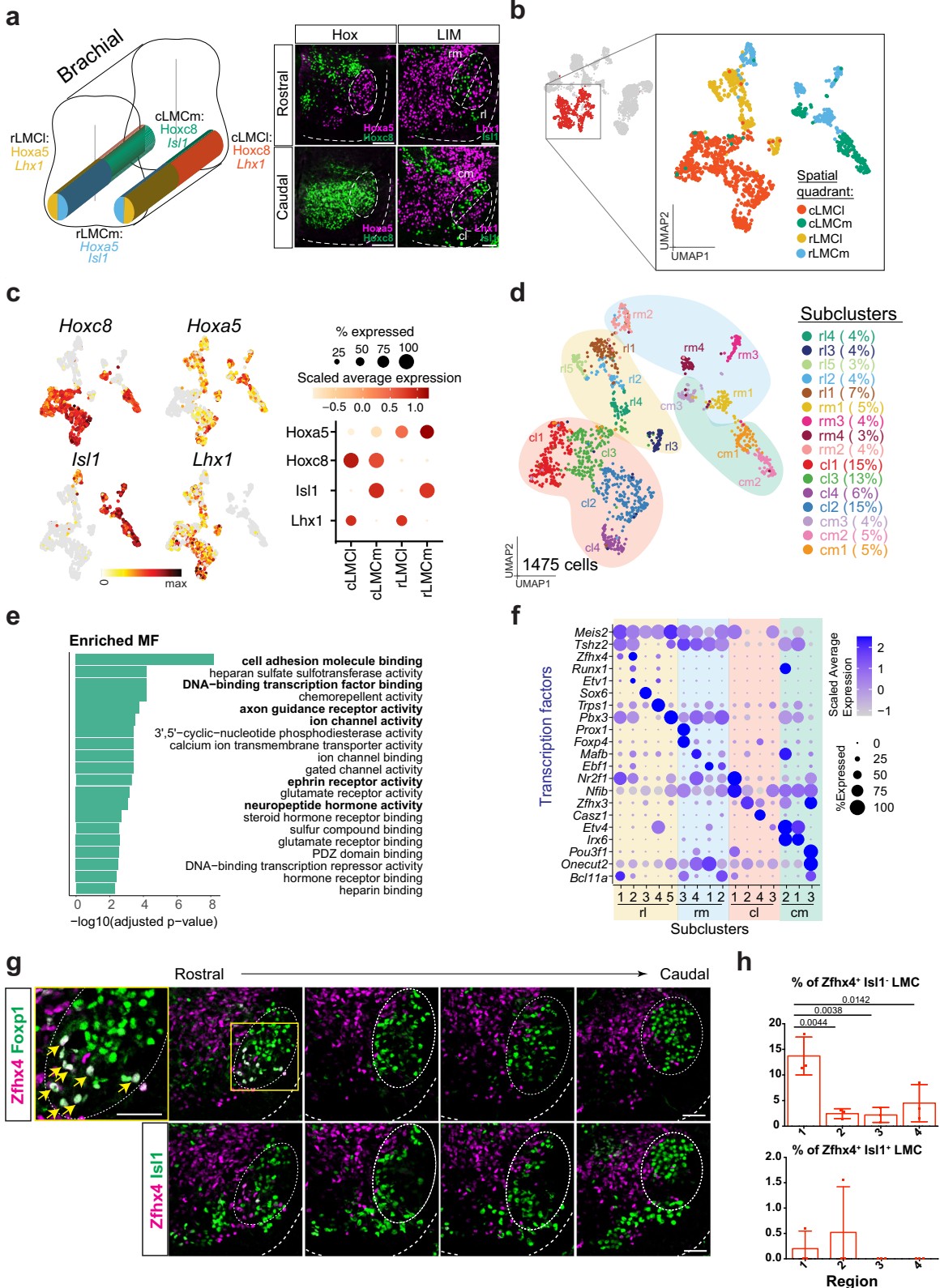

pathway and translation regulation—were more prominent in lumbar than brachial LMC MNs (Fig. 6b). In contrast, neuropeptide-related terms were more enriched in the brachial LMC MNs (Fig. 6b). To validate that the observed differences between limbs are bona fide biological variations rather than batch sampling artifacts, we performed RNAscope multiplexed fluorescence in situ hybridization and quantified cell numbers to confirm higher expression of *Grp* and *Npy*

neuropeptides in the brachial LMC MNs (Fig. 6c, d and Supplementary Fig. 9a). Additionally, we identified important neuropeptide genes such as *Igf1*[50], known to exert a neuroprotective function, as being differentially enriched in Hoxd10[+] medial LMC MNs in the lumbar segment (Fig. 6c, f). Similarly, we observed that these neuropeptide-expressing populations are dynamically distributed across the rostrocaudal axis of the brachial or lumbar spinal cord (Fig. 6d–g and

**Fig. 4 | Delineation of molecular diversity in limb MNs based on differential expression of transcription factors and neuropeptides. a** Schematic diagram of LMC MN subtypes based on their cell body positions in the ventral horn of the spinal cord. Cells residing in the brachial segment are divided into four "spatial quadrants": rLMCl (rostral lateral), rLMCm (rostral medial), cLMCl (caudal lateral) and cLMCm (caudal medial) regions based on the expression patterns of "spatial genes," i.e., Hox and LIM homeodomain factors. Representative immunostaining images of spatial genes (*n* = 3 embryos) are shown on the right. **b** UMAP plot of cells from the brachial LMC cluster (red) as in Fig. 1c, with further subclustering into the four "spatial quadrants." **c** UMAP (left) and dot-plots (right) of *Hoxa5, Hoxc8, Isl1*, and *Lhx1* expression patterns. **d** UMAP plot of all 16 LMC subclusters. Cell proportions for each subcluster are indicated as percentages. The four "spatial quadrants" are shaded to match the colors in **b**. **e** Gene ontology enrichment of molecular functions of DEGs across brachial LMC neurons. Terms of interest in this study are highlighted in bold. **f** Dot-plot showing expression patterns of

representative markers (rows) from the top differentially expressed TFs among LMC subclusters (columns). **g, h** Immunostaining and quantification of the transcription factor Zfhx4/Foxp1/Isl1 on spinal cord sections from different regions of the brachial segment along the rostrocaudal axis (1 → 4) (see "Methods"). A high-magnification image of the LMC MNs (yellow box) is shown, with arrowheads indicating Zfhx4 in Foxp1⁺ LMC cells. Zfhx4⁺Isl1⁺ and Zfhx4⁺Isl1⁻ were calculated against Foxp1⁺ LMC cells. For all panels, Scale bar represents 50 μm. Dashed lines outline the spinal cord boundary and dashed circles demarcate LMC MNs. Single cells were pooled from *n* = 12 embryos of 2 pregnant mice, brachial LMC: 1475 cells. Adjusted *p* values are from **e**: one-sided hypergeometric test followed by a Benjamini–Hochberg correction; **h**: one-way ANOVA with Tukey's multiple comparison test, *n* = 3 embryos. Data are presented as mean ± SD. Only significant values (adjusted *p* value <0.05) are shown. Source data are provided as a Source data file.

Supplementary Fig. 9a, b), corroborating their aforementioned cluster-specific expression (Fig. 5b). The differential neuropeptide expression in either brachial or lumbar LMCs was not a reflection of a developmental timing delay, as when we compared E12.5 brachial and E13.5 lumbar LMC MN datasets, we found that the neuropeptide signaling pathway (GO:0007218) is still more enriched in the E12.5 brachial[23] compared to E13.5 lumbar segment, representing a similar outcome to our findings for segmental gene expression at the same stage (E13.5) (Supplementary Fig. 9c, d). Furthermore, expressions of brachial-enriched *Npy* and *Grp* were consistent from E13.5 to E14.5 (Supplementary Fig. 9e, f). Therefore, our results indicate that, like TFs and cell adhesion molecules, neuropeptides not only exhibit heterogeneous expression among LMC subclusters but also differ between the brachial and lumbar segments.

Next, we probed for correspondences between the brachial and lumbar LMC subclusters. Similar to the above-described approach, we clustered lumbar LMC MNs and identified three "spatial quadrants" (Supplementary Fig. 10a). Further subclustering of each of those spatial quadrants resulted in 10 subclusters and DEG analysis identified their distinct molecular markers (Supplementary Fig. 10b–e and Supplementary Data 3). To effectively compare LMC subclusters between the brachial and lumbar segments, we integrated all LMC cells from all limbs and jointly projected them onto a shared UMAP space using the Harmony algorithm[49] (Fig. 7a–c). By performing quantitative similarity analyses based on the mutual nearest neighbors of each cell in the UMAP space and the Harmony space, we were able to assess the relatedness of LMC cells across the brachial and lumbar segments (Fig. 7d, e and Supplementary Fig. 11a). We observed two principles: (1) medial subclusters are similar between brachial and lumbar segments, as is the case for lateral subclusters in brachial/lumbar LMC MNs, suggesting that both segments share the molecular basis of mediolateral identity, whereas the similarity among rostrocaudal subclusters between limbs is less evident (Fig. 7c–e and Supplementary Fig. 11b); and (2) though the overall molecular profile for mediolateral identity is similar for brachial/lumbar LMCs, a detailed comparison of each subcluster revealed a divergent tendency. Nineteen subclusters, including Etv4⁺, were found to have correspondences between the brachial and lumbar segments (Fig. 7f and Supplementary Fig. 11c), but five and two LMC subclusters were specific to the brachial or lumbar segment, respectively. For example, the *Grp*⁺ cl3 and rl5 subclusters are present in the brachial segment alone (Fig. 7g and Supplementary Fig. 11d). Although Nkx6.1 is known to be expressed in certain Isl1⁺ LMC MNs in brachial and lumbar segments[43], we further identified a Nkx6.1⁺ Isl1⁻ subpopulation, presumably the crl6 cluster, which is only located in the caudal part of the lumbar segment (Fig. 7g–i). Overall, our comparison between brachial and lumbar LMC MNs reveals convergent and divergent gene expression profiles and subtypes between the two limbs, including TFs and neuropeptides, highlighting

how these signature codes might serve as an organizing gene logic within LMC MNs.

## Identification of MMC MN subtypes and molecular signatures

Our identification of more MN subtypes in LMC neurons prompted us to assess the previously underappreciated molecular heterogeneity in epaxial muscle-innervating MMC MNs. We hypothesized that, like LMC MNs, MMC neurons also comprise molecularly distinct subtypes. Indeed, a recent study uncovered molecular diversity within MMC MNs, with ~40% of MMC neurons expressing *Ebf2* and loss of this gene resulting in a reduction in the population of axial MNs[51]. To test our hypothesis, we clustered MMC MNs from the rostral sample and identified three major subclusters, namely *Nr2f2*⁺ (Coup-TFII), *Satb2*⁺, and *Bcl11b*⁺ (Ctip2) (Fig. 8a–c and quantification in Fig. 8d). We validated the expression of these markers in MMC MNs on mouse spinal cord sections at E13.5. Satb2 and Nr2f2 were expressed in MMC MNs in a mutually exclusive manner (Fig. 8c). Although some Bcl11b⁺ cells were co-expressed with Satb2 or Nr2f2, these double-positive cells (Bcl11b⁺Satb2⁺ or Bcl11b⁺Nr2f2⁺) did not manifest salient hallmarks (Fig. 8c and Supplementary Fig. 12a, b). Previously reported Ebf2⁺ MNs were identified in the *Satb2*⁺ subcluster (Supplementary Fig. 12c, d). Our scRNAseq and immunostaining results thus support that MMC MNs display molecular heterogeneity. Given that the spatial arrangement of LMC MNs corresponds to the distal-to-proximal positioning of innervating muscle targets[52], we examined if the MMC subtypes display spatial organization. A topographical analysis revealed a preferential medial localization in the brachial MMC for the Satb2⁺ relative to the Nr2f2⁺ and Bcl11b⁺ MMC MNs (Fig. 8e). We also identified that the three subtypes are distinguishable based on their expression of receptor and axon guidance molecules that are essential for precise axon targeting[53]. In particular, the *Satb2*⁺ and *Bcl11b*⁺ subclusters displayed *Sema3c, Sema6a* and *Nrp2* expression, whereas the *Nr2f2*⁺ subcluster was enriched for *Sema5a, Nrp1* and *Unc5c* expression (Fig. 8f). RNAscope-based multiplex in situ hybridization further confirmed that *Nrp2* expression is more enriched in the *Satb2*⁺ subpopulations (Fig. 8g, h). Their differential expression prompted us to investigate if MMC subtypes project their axons in distinct trajectories. Therefore, first we scrutinized under high magnification the axonal projection patterns of Mnx1-GFP embryos and observed that the dorsal ramus bifurcates into branches that project towards epaxial muscles such as the spinalis and longissimus/iliocostalis muscles (Supplementary Fig. 12e). We also observed preferential expression of axon guidance receptor molecules Nrp2 (enriched in Satb2⁺) and Unc5c (enriched in Nr2f2⁺) in distinct branches of the dorsal ramus (Fig. 8i), suggesting that the molecularly defined MMC subtypes might innervate different epaxial muscle types through distinct routes. Collectively, our results indicate that, similar to LMC MNs, MMC subtypes exhibit preferential mediolateral distributions and they selectively

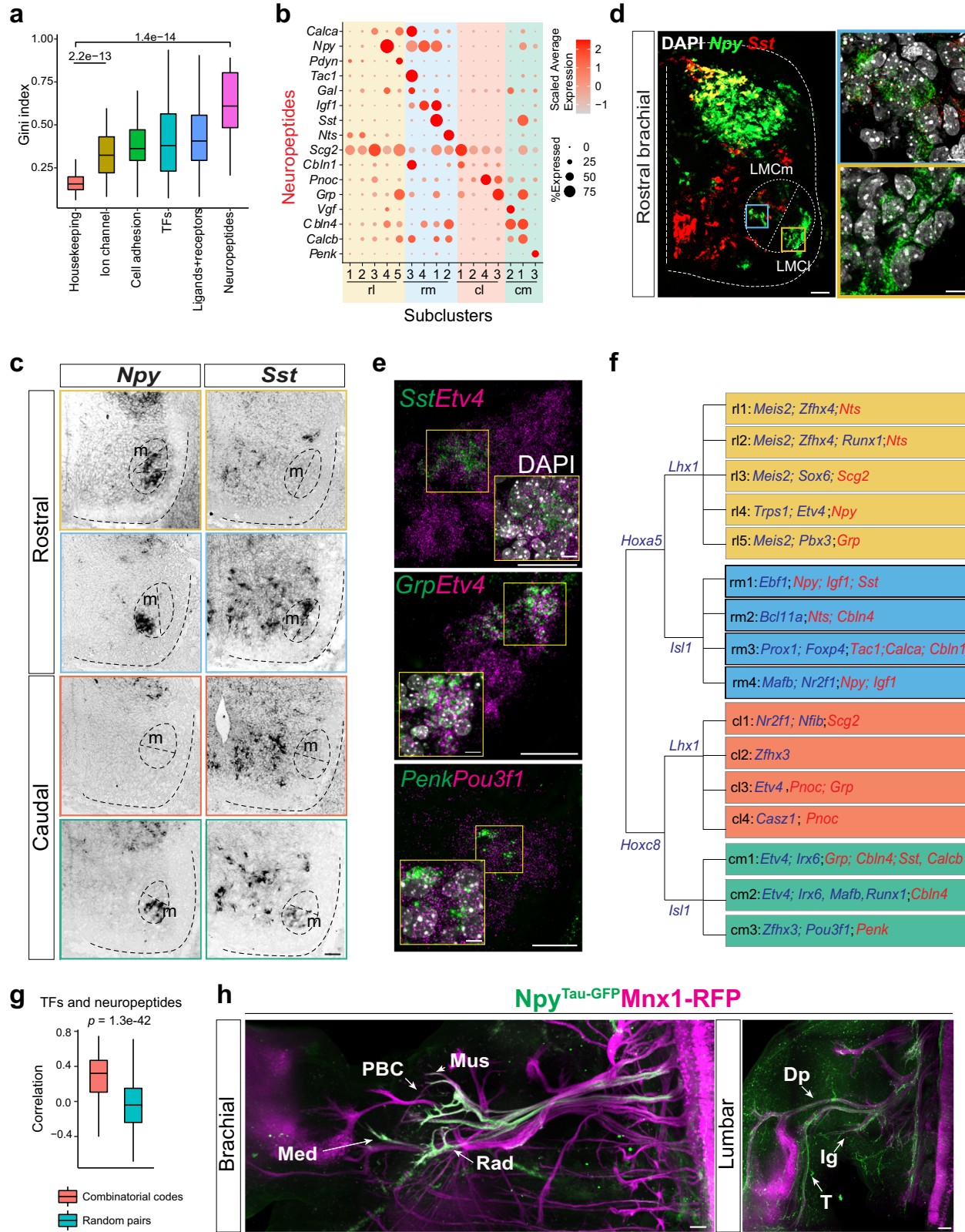

**MMC MN subclusters in chicken and human embryos**
During evolution, MN types have changed dramatically in parallel with the musculoskeletal system to adapt to versatile lifestyles. The epaxial muscle-innervating MMC neurons are regarded as the ancestral MNs,

based on their occurrence in primitive vertebrates such as lamprey[7]. Our identification of MMC subtypes in mouse embryos led us to examine further if similar subtype heterogeneity is conserved across higher vertebrates, including in chickens and humans. In the chicken spinal cord, we detected the MMC marker Mecom in the majority of Lhx3+ MMC MNs (Fig. 9a). Similar to mouse embryos, these MNs also expressed *Nr2f2*, Satb2 and Bcl11b (Fig. 9b–d). Satb2 and *Nr2f2* were

express TFs, receptors and axon guidance molecules that might facilitate high fidelity axonal innervations to the correct epaxial muscles.

**Fig. 5 | Validation of cluster-specific TFs and neuropeptides defining LMC MN diversity. a** Comparison of expression variability of functional gene sets within LMC MNs by the Gini index. Housekeeping genes ($n = 289$) were used as a reference. Genes associated with each functional term were intersected with highly variable genes across LMC MNs (ion channel, $n = 57$; cell adhesion molecules, $n = 61$; TFs, $n = 191$; ligands/receptors, $n = 296$; neuropeptide, $n = 23$). **b** Dot-plot showing expression patterns of representative neuropeptides among LMC subclusters. **c** In situ hybridization (ISH) of *Sst* and *Npy* in LMC MNs. Colored boxes indicate representative images for each "spatial quadrant." **d** RNAscope-based ISH of *Npy* and *Sst* in the rostral brachial spinal cord section. High-magnification images (blue box, *Npy* and *Sst* co-expression in medial LMCs; yellow box, *Npy*-only in lateral LMC) are shown. **e** RNAscope-based ISH of TF and neuropeptide combinatorial expression in LMC neurons. High magnification images are shown (yellow box). **f** Summary of TF (blue) and neuropeptide (red) combinatorial codes that define the

sixteen LMC subclusters. **g** Spearman's correlation of subcluster-specific TFs and neuropeptides relative to random pairs of TFs and neuropeptides ($n = 338$ independent experiments). The three lines (top to bottom) represent the 25th, 50th (i.e., center), and 75th percentiles of data values of the samples. The whiskers extend to the most extreme values (i.e., minima and maxima) within 1.5 times the IQR of the median. **h** Npy$^+$ neuronal axon projection in brachial and lumbar motor nerves. PBC posterior brachial cutaneous, Mus musculocutaneous, Med median, Rad radial, Ig inferior gluteal, T tibial, Dp deep peroneal nerves. For **a**, **g**, two-tailed Wilcoxon rank-sum tests. Each staining experiment was independently repeated on **d**, **e**: $n = 3$ and **h**: $n = 2$ embryos. The spatial information demarcated by the dashed circles and lines was annotated based on immunostaining for *Hox* (*HoxaS/Hoxc8*) and LIM homeodomain TFs (Isl1/Lhx1) on adjacent slides in all figures. Scale bar: high-magnification images, 10 µm; **h**, 100 µm; others 50 µm. r rostral, c caudal, m medial, l lateral.

expressed in a mutually exclusive manner and displayed a preferential mediolateral distribution (Fig. 9c, d and overlay (right)), whereas expression of Bcl11b partially overlapped with Satb2$^+$ MMC MNs. We also performed immunostaining on spinal cord sections from human embryos at Gestational Week (GW) 7.9 (equivalent to mouse E13.5 embryos in which MN columnar identities have been established)[54,55]. Most of the ventromedial MMC MNs were MECOM$^+$ (Fig. 9e), and all of the MMC subtype markers we identified in mice—NR2F2, SATB2 and BCL11B—were detected in human MMC MNs (Fig. 9f–h). A few BCL11B$^+$ cells overlapped with NR2F2$^+$ or SATB2$^+$ cells. Similar to the expression patterns in chicken and mouse embryos, we detected segregation of SATB2$^+$ and NR2F2$^+$ cells among the MECOM$^+$ MMC MNs (Fig. 9h). This segregation was also apparent at GW 10.3, indicating that the distinct subtypes persist in developing human spinal cords (Fig. 9i). Collectively, our findings suggest that Bcl11b, Satb2 and Nr2f2 mark MMC subtypes in chicken, mouse, and human embryos, implying an evolutionarily conserved distribution of MMC subtypes in higher vertebrates. Our results may help decipher how MMC subtypes have evolved to accommodate body posture in different vertebrates displaying various spinal alignment, movement and breathing styles.

## Discussion

Our study has generated a detailed molecular atlas of spinal MNs at the embryonic stage. We profiled MNs from mouse embryos at embryonic stage E13.5, i.e., when most subtype identities have already been established and MNs have acquired their intrinsic properties to connect to appropriate muscle groups. Given the rarity of MNs in the spinal cord, we enriched for MNs using Mnx1-GFP reporter mice and subjected GFP-labeled MNs to FACS before performing single-cell RNA sequencing. This enrichment strategy expanded the total number of postmitotic embryonic MNs we collected to almost 10,000, unlike a previous study that focused on studying whole spinal cord diversity[15]. Most of our major cell clusters reflect motor columns based on known genetic markers, and we identified genetic markers that represent additional means to label and manipulate specific motor columns. Moreover, we present two previously uncharacterized segmentally-restricted cell clusters, which display some shared features with MMC MNs but remain molecularly distinct. Furthermore, we leveraged the large number of MNs in our dataset to permit a more detailed and unbiased analysis of heterogeneity within LMC and MMC MNs, resulting in 16 and 10 LMC subclusters from brachial and lumbar segments, respectively, and 3 MMC subclusters. The unique molecular codes expressed by each subcluster may not only serve as markers, but also provide insights into the intrinsic characteristics of individual cell populations and how robust axon targeting and neural connectivity are achieved.

In addition to known MN subtypes, we discovered two segmentally-restricted cell clusters in MNs, i.e., Calb1$^+$ MNs in the brachial and thoracic region and Nfib$^+$Grm5$^+$ MNs in the sacral segment. Both of these MN subtypes are close to the MMC MN region and

share canonical MMC markers *Mecom* and *Lhx4*. However, these two clusters exhibit low transcriptome similarity to MMC MNs. It is still not clear if the two identified clusters are columnar MNs, or belong to a derivative class of axial MNs that exhibit a segmental preference. It is generally regarded that the epaxial and hypaxial muscles are separated by the horizontal septum, and MMC MNs are believed to innervate the epaxial muscles exclusively via the dorsal rami, whereas HMC MNs control the hypaxial musculature via the ventral rami[6]. However, recent studies have revealed that some ventral rami-targeting MNs have a molecular profile and are medially positioned like MMC MNs that innervate epaxial muscles[56,57]. This scenario highlights the pressing need for a unified and detailed consideration of molecular profiles and muscle innervation patterns at the level of muscle groups. Since the two previously uncharacterized Calb1$^+$ and Nfib$^+$Grm5$^+$ subtypes unveiled in our study also express generic MMC markers and show similar localizations to ventral rami-targeting MNs, it will be tantalizing to investigate further if they innervate the same muscles characterized in that previous study[56] and if their molecular distinctiveness defines the preference for the ventral versus dorsal ramus.

Differential vulnerability is a recurring theme in neurodegenerative diseases affecting MNs, in which motor pools respond differently to disease[58,59]. To date, the lack of means to track neuronal subtypes in fine detail has limited studies of differential vulnerability. Most reported differences in susceptibility relate to more accessible and larger muscles (such as the soleus and gastrocnemius in amyotrophic lateral sclerosis (ALS)) and broader muscle groups (such as the paraspinal muscles versus intercostal muscles in spinal muscular atrophy (SMA)). We envisage that the molecular diversity uncovered in this study will enable tracking and comparisons of fine neuronal subtypes in disease models by means of lineage tracing, potentially revealing directly the molecular nature of subtype-specific susceptibility. Two subtypes identified in this study, rostral Calb1$^+$ and caudal Nfib$^+$ Grm5$^+$, are marked by genes implicated as being protective against MN disease[60]. Several studies in rodents and humans have reported that higher expression levels of Calb1 and Grm5 are associated with resistance to ALS, a MN degenerative disease[60]. Calb1 has been shown to buffer excessive excitotoxic Ca$^{2+}$, and Grm5 overexpression was found to prevent programmed cell death in in vitro cultures of rat cerebellar granule cells at the early maturation stage[60–62]. Further investigation of whether these clusters exhibit selective resistance to neurodegenerative disease would be a unique opportunity to understand the molecular nature underlying "disease resistance" in MNs and provide potential therapeutic targets.

Although LMC MNs are classified according to their innervation of different limb muscles, the molecular differences between these neurons have remained largely unclear. Our study complements and extends the knowledge of LMC MN diversity by describing distinct expression patterns of combinations of transcription factors and neuropeptides among the MN subtypes. We have also validated in vivo a number of markers for these subtypes. Despite the roles of

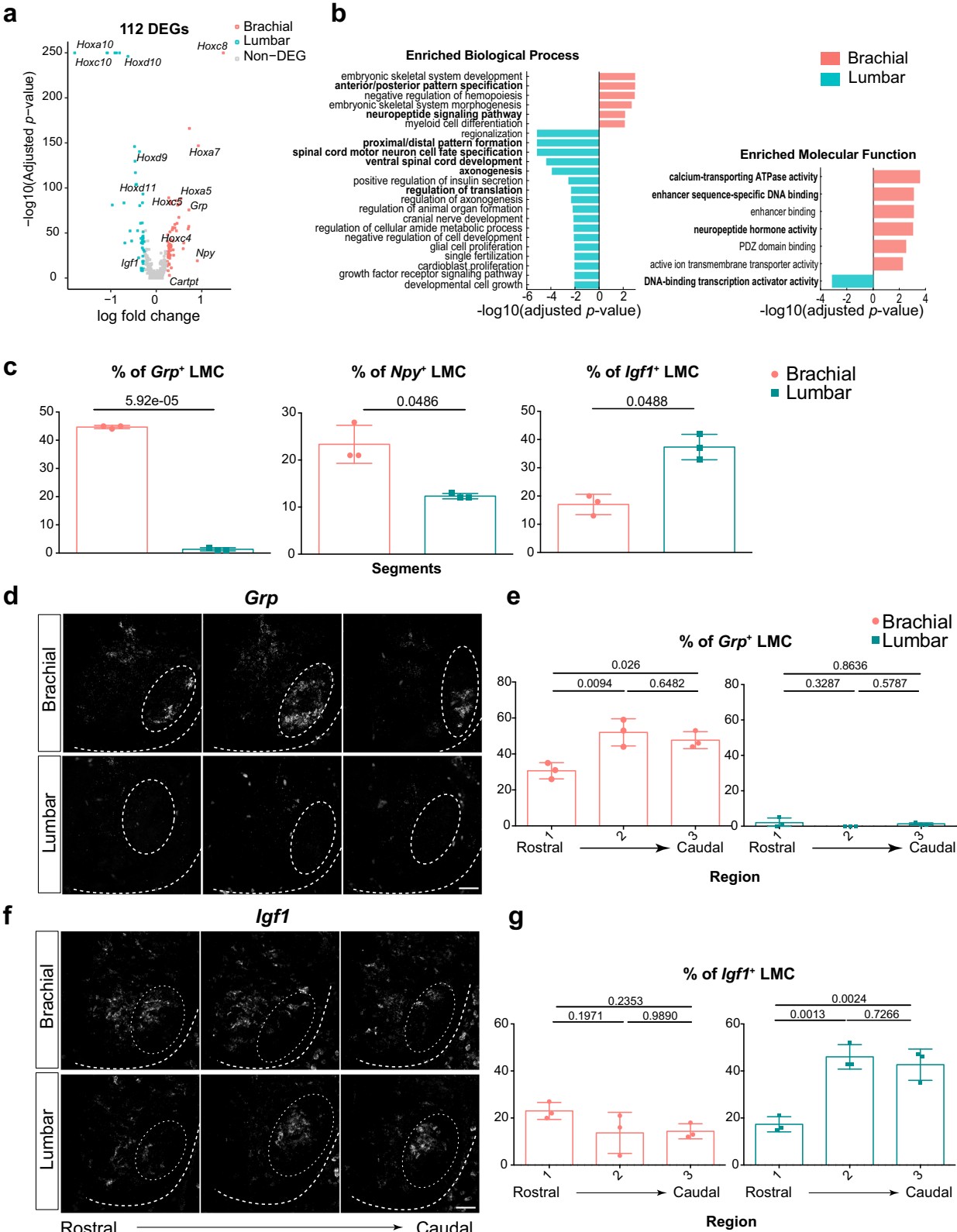

neuropeptides in MN development remaining unclear, their unexpected diversity in neurons could point to functions beyond neurotransmission in embryonic spinal MNs. Indeed, a few neuropeptides in MNs have been assessed in previous studies, which suggested they potentially serve as trophic factors to promote nicotinic acetylcholine receptor (AChR) accumulation and to facilitate formation of neuromuscular junctions[63]. We hypothesize that these neuropeptides might:

(1) modulate motor pool-specific electrophysiology via autocrine signaling; (2) regulate motor circuit formation by guiding premotor interneuronal innervation to MNs, or (3) ensure the fidelity of finer innervation patterns to the peripherals. To better understand the functions of neuropeptides in neurons, it is tempting to investigate if heterogeneous electrophysiological patterns are displayed by MNs and correspond to motor pools, as well as being associated with

**Fig. 6 | Divergent features between brachial and lumbar LMC neurons.**
**a** Volcano plot displaying the DEGs of LMC MNs between segments. Colored dots reflect genes that exhibit significantly differential expression. Genes with an adjusted *p* value <0.05 and log fold-change >0.25 were deemed differentially expressed. The log-transformed fold-change is shown on the x-axis. *Hox* genes are the top-ranked markers differentially expressed between brachial (salmon) and lumbar (turquoise) segments. Note that a few neuropeptide genes (e.g., *Igf1*, *Npy*, *Grp*) were observed in the list of DEGs. **b** Enrichment analysis of DEGs between brachial and lumbar LMC neurons for biological process (BP, left) and molecular functions (MF, right). The bold text highlights terms of interest in this study. Notably, the neuropeptide signaling pathway is shown as enriched in brachial segments. **c** Quantification of the percentage of neuropeptide-expressing cells in *Chat*⁺ or *Slc18a3*⁺ LMC MNs between brachial and lumbar segments.

**d, f** Fluorescence ISH of the **d** brachial-enriched *Grp* and **f** lumbar-enriched *Igf1* neuropeptides. Images are arranged from left to right according to their relative position along the rostrocaudal axis of the brachial (upper panel) and lumbar (bottom panel) spinal cord. **e, g** Quantification of the percentages of neuropeptide-expressing cells in Chat⁺ or Slc18a3⁺-expressing cells within LMCs based on images in **d, f**. Results show the differential distribution of neuropeptide-expressing populations in different regions along the rostrocaudal axis (1→ 3) within the brachial (salmon circle) and lumbar (turquoise square) segments. Adjusted *p* values are from **a**, **b**: one-sided hypergeometric test followed by a Benjamini−Hochberg correction. **c**: two-tailed paired t-test, *n* = 3 embryos. **e, g**: one-way ANOVA with Tukey's multiple comparison test, *n* = 3 embryos. All data are presented as mean ± SD. Source data are provided as a Source data file. Scale bar represents 50 μm.

neuropeptide usage. Characterizing expression patterns of neuropeptide receptors in corticospinal projection neurons, spinal interneurons, and individual muscle groups will help narrow down the candidate receivers of neuropeptide signals and allow examinations of how connectivity is altered upon gain or loss of function of a given neuropeptide in MNs. Moreover, we also noted differential neuropeptide enrichment between brachial and lumbar segments, which might explain the potential differences in motor circuitry between these spinal cord segments. Consistent with this notion, we also report here that not all LMC subclusters are conserved between brachial and lumbar segments. In our pioneering study, while we attempted to compare cluster similarity at the transcriptome level, we also recognize that there might be functionally distinct clusters existing in different segments that are defined by other extrinsic factors, such as innervation target preference or pre-motor circuit involvement. Further experimental validations, including marker staining with axonal tracing, to complement computational analyses, are warranted.

Although we have expanded the number of molecularly defined LMC subtypes, the number does not match the expected ~60 known LMC motor pools. There are a few possible reasons for this outcome. First, though motor pools are expected to be molecularly defined at the transcriptome level, the subtypes we identified could also correspond to one or multiple motor pools that manifest subtle transcriptomic differences. Second, motor pools at the embryonic stage might not be reflected by the transcriptome alone. An increasing number of studies have demonstrated that single-cell multi-omics analyses, such as joint single-cell transcriptomic and epigenomic profiling, can increase the ability to dissect cellular heterogeneity[64,65]. Third, some neuronal identities may only be elicited by signaling or activity during spinal cord wiring at the postnatal stage[66]. Accordingly, a longitudinal examination of spinal MN identity integrated with later time-points may be enlightening[66]. Fourth, although we determined the number of subclusters using a robust ensemble approach based on mathematical spectral graph theory, single-cell transcriptomic data is inherently noisy and sparse in nature, which engenders additional challenges for estimating optimal numbers of clusters from such datasets. Further computational subclustering analysis with in vivo functional validation would likely enhance our understanding of LMC diversity.

Axial MNs that control core muscles represent an evolutionarily ancient neuronal type and they play pivotal functions across most vertebrate species. Diversity within the relatively small population of MMC neurons in the spinal cord is less appreciated due to their largely unknown molecular profiles. In this study, we describe transcriptomic heterogeneity within MMC MNs. Notably, we observed mediolateral expression patterns for Satb2 and Nr2f2. We further speculated that these MMC MN subtypes might innervate different axial muscle groups, such as the spinalis, longissimus, iliocostalis or levator costae muscles[11], since receptor genes specific to either the Satb2⁺ or Nr2f2⁺ subtype are detected in distinct branches of the dorsal ramus. Future studies to confirm functional differences between these subtypes via

retrograde labeling in embryonic muscles will be necessary, albeit technically challenging. In addition, the identified MMC subtype markers of Nr2f2, Satb2 and Bcl11b facilitate region patterning in the brain[67,68]. Whether they exert similar regulatory roles in MMC MNs warrants further investigation. Moreover, the mutually exclusive expression pattern of Nr2f2 and Satb2 suggests potential regulation between these two subtypes, which should prompt future genetic manipulation or lineage tracing studies.

An elegant recent survey revealed that the neuronal subtype essential for walking originated in primitive jawed fish[9], highlighting that the common ancestor of most vertebrates already possessed a sophisticated blueprint for walking. While it remains unclear how axial neuronal subtypes and their innervating motor circuits changed during vertebrate evolution, axial muscles perform distinct functions in different species. For example, despite the presence of axial MNs in both aquatic and land animals, these animals utilize the axial muscles innervated by those MNs distinctly; most fish generate propulsive forces via rhythmic contraction of their axial muscles during swimming, whereas tetrapods employ axial muscles for non-locomotory spinal alignment and breathing[6,69]. It is tempting to speculate that the MMC subtypes co-evolved with different lifestyles and now serve different modes of locomotion. Therefore, we investigated MMC neuronal heterogeneity in embryos of three different vertebrates—chicken, mouse, and human—based on immunostaining of the Nr2f2/Satb2/Bcl11b markers and observed that the same set of proteins consistently labels subsets of MMC MNs in these species. Interestingly, we observed that the SATB2 and NR2F2 mediolateral positioning we observed in mouse embryos might be inverted in humans. Since the transcriptional program underlying MMC subtype identities might be critical to MMC subtype topographical organization, additional experiments on human embryos will be necessary in future to investigate if this inversion leads to a reorganization of the MMC topographical distribution and whether this change is functionally important. The conservation of markers provides genetic access to MMC subtypes for future comparative studies and allows the characterization of gene expression signatures and innervation patterns of MMC subtypes in various species. For example, at least two MN subtypes have been reported in zebrafish;[6] their primary and secondary MNs are characterized based on birth order, cell body size and muscle innervation. The molecular nature of these subtypes has yet to be established, although they are known to express generic MN genes such as the *mnx* paralogs *islet1* and *islet2*[70,71]. It remains to be determined if the markers of mouse MMC subpopulations we have identified herein are also expressed in zebrafish or other aquatic vertebrates. Comparative transcriptomics would be needed to explore similarities and divergences among such species and would require challenging integration analysis of single-cell MNs and in situ hybridization validation experiments.

Unexpectedly, either using trypsin- or papain-based dissociation approaches, we acquired equivalent numbers of MMC (2354 cells) and LMC (2248 cells) MNs, which is inconsistent with the population ratio

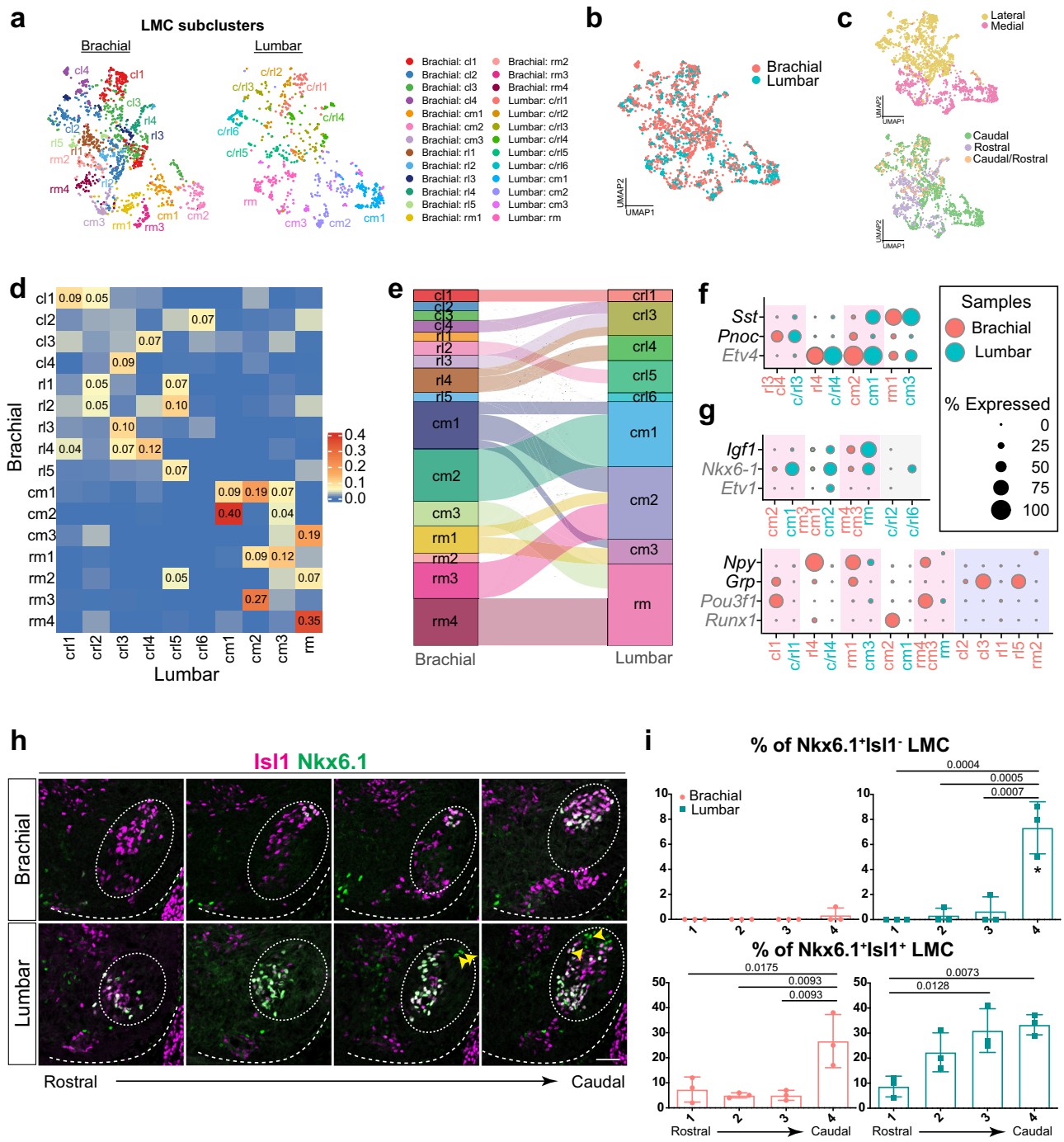

**Fig. 7 | Comparison of subclusters between brachial and lumbar LMC neurons.**
**a** Joint UMAP visualization of all brachial and lumbar LMC MN subclusters.
**b** UMAP showing LMC MNs from both samples are integrated without confounding technical artifacts. **c** LMC MNs with medial and lateral (upper panel), rostral and caudal (bottom panel) identities on the joint UMAP space. **d** Similarity analysis between LMC subclusters from rostral and caudal segments, as reflected by the heatmap, was quantified by the proportion of overlapping mutual nearest neighbors. **e** An alluvial diagram intuitively illustrates the relatedness between brachial and lumbar LMC subclusters. Line thickness reflects the level of similarity between subclusters, and the size of each box reflects total similarity. Lines with similarities <0.09 have been omitted. **f, g** Combined dot-plot showing marker gene expression (color-coded along *Y*-axis: gray for known motor pool markers; black for neuropeptides) in LMC subclusters (*X*-axis) from a merged dataset. Three categories are highlighted: **f** selected genes expressed in both brachial and lumbar subclusters; **g** genes preferentially expressed in brachial (bottom panel)

and lumbar (upper panel) segments. LMC subclusters in different segments have been grouped based on similarity data from **d**. Circle size indicates the percentage of cells expressing the genes. **h** Immunostaining of Nkx6.1/Isl1 along the rostrocaudal axis of brachial (upper panel) and lumbar (bottom panel) segments of the spinal cord. Arrows indicate Nkx6.1-expressing cells in LMCl (Isl1⁻). Dashed lines outline the spinal cord boundary, and LMC MNs are demarcated by circles based on Foxp1 expression. **i** Quantification of the percentages of Nkx6.1⁺ Isl1⁻ (lateral) and Nkx6.1⁺ Isl1⁺ (medial) cells in Foxp1⁺ LMC neurons within different regions of the brachial segment. Asterisk (*) denotes the unique enrichment of the Nkx6.1⁺ Isl1⁻ population in lumbar. Single-cells were pooled from *n* = 12 embryos of 2 pregnant mice, brachial LMC: 1475 cells; lumbar LMC: 774 cells. For **i**, data are presented as mean ± SD, adjusted *p* value was analyzed by one-way ANOVA with Tukey's multiple comparison test, *n* = 3 embryos. Only significant values (adjusted *p* value <0.05) are shown. Source data are provided as a Source Data file. Scale bar, 50 μm.

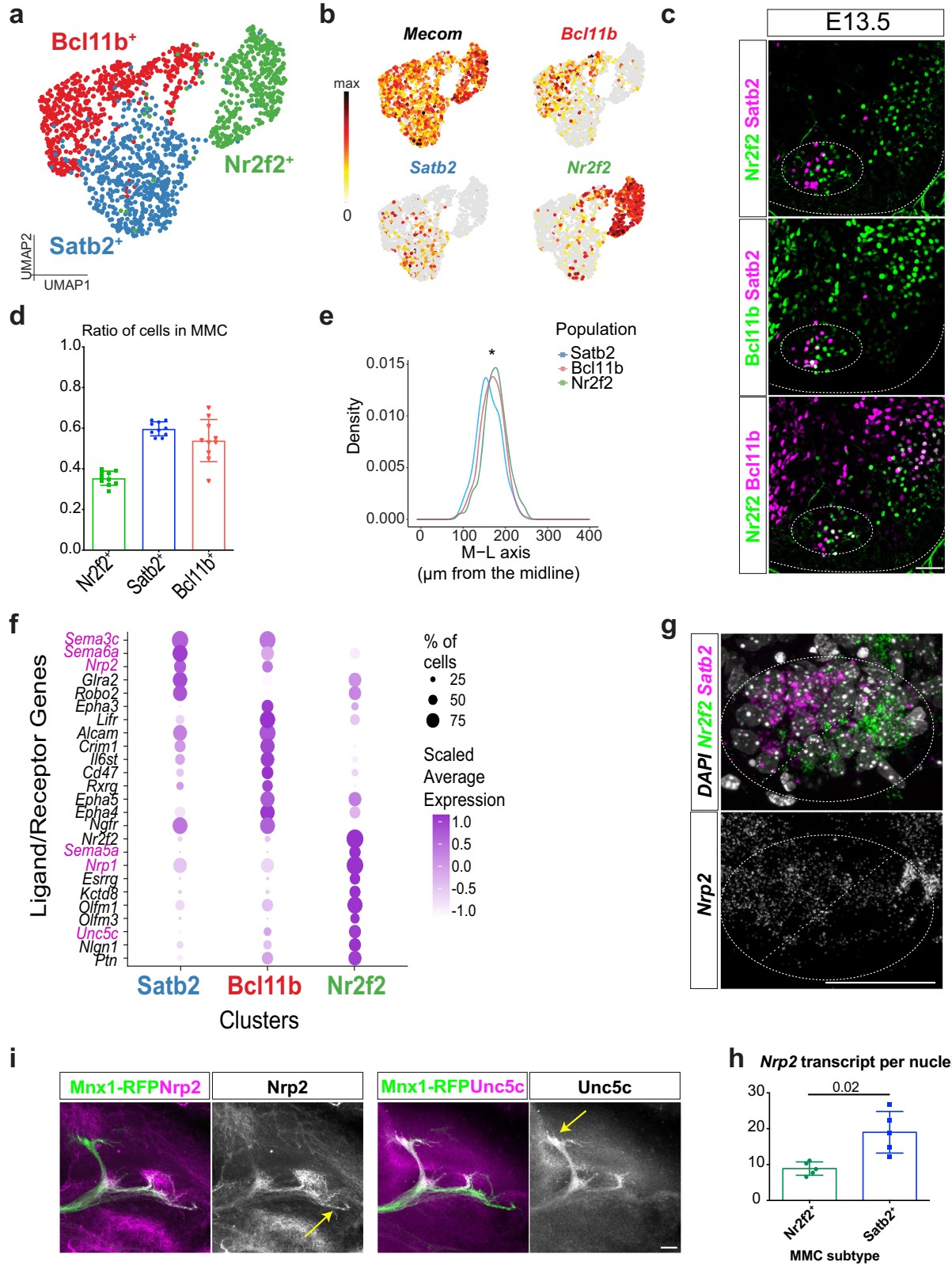

of brachial LMC to MMC MNs (4:1) in E13.5 mouse embryos[72]. This scenario may reflect that LMC MNs are intrinsically vulnerable to shear stress, regardless of how mild a dissociation approach is applied, and it is consistent with previous reports indicating that LMC MNs are more susceptible to degeneration in amyotrophic lateral sclerosis[73]. Additionally, we noticed that although the rostrocaudal identities of spinal cord cell types are determined and maintained by cross-regulation

among Hox proteins[33], our single-cell analysis revealed that *Hox* transcripts, such as *Hoxa5* and *Hoxc8*, can still be co-expressed within the same cell, despite being segregated at the protein level. This outcome is consistent with our previous studies showing that post-transcriptional mechanisms, such as those exerted by microRNAs, are indispensable to the unambiguous governance of spatiotemporal Hox protein expression and for defining rostrocaudal MN subtype

**Fig. 8 | Molecular and functional heterogeneity in MMC neurons. a** UMAP visualization of brachial MMC cells with three molecularly distinct subtypes. **b** Normalized expression of marker genes on UMAP space. *Mecom* is a known marker for MMC MNs and is highly expressed in all MMC neurons, but *Bcl11b*, *Satb2* and *Nr2f2* are enriched in different subtypes. **c** Immunostaining for the MMC subtype markers Satb2, Nr2f2 and Bcl11b in brachial segments of E13.5 spinal cord. Nr2f2 and Satb2 are expressed in a mutually exclusive manner, whereas a fraction of Bcl11b expression overlaps with both. **d** Quantification of subtype ratio in Mecom$^+$ MMC neurons in brachial E13.5 spinal cord. **e** Mediolateral (M-L) density plot of the Satb2$^+$ (blue), Nr2f2$^+$ (green) and Bcl11b$^+$ (red) subtypes in brachial E13.5 spinal cord. **f** Dot-plot showing selective expression of ligand/receptor genes in MMC subtypes, such as differential *Nrp1* and *Nrp2* expression in the *Nr2f2*$^+$ and *Satb2*$^+$ subpopulations, respectively. **g** Representative image of *Nrp2*, *Nr2f2* and *Satb2* multiplexed fluorescence ISH. **h** Quantification of *Nrp2* transcripts per nuclei in the MMC subtype. Total nuclear *Nrp2* transcripts were normalized to the total number of *Satb2*- or *Nr2f2*-expressing cells. **i** Immunostaining of Nrp2 and Unc5c reveals their expression on distinct axonal branches. Mnx1-RFP was used to visualize all branches of the dorsal ramus. Arrows show differences in receptor expression on specific axonal branches, repeated on $n = 3$ embryos. Scale bar for all images in this figure represents 50 μm. Single cells were pooled from $n = 12$ embryos of 2 pregnant mice, MMC: 1594 cells. For quantification analysis, results are shown as mean ± SD. **d:** $n = 10$ embryos for each MMC subtype, average ratios from 2 to 4 sections for each embryo, one-way ANOVA with Tukey's multiple comparison test; **e:** $n = 12$ embryos. Kruskal–Wallis Dunn's multiple comparison test, $p$ values adjusted with Holm method for Satb2-Nr2f2: 1.57e-6; Bcl11b-Nr2f2: 0.0127; Bcl11b-Satb2: 0.0063, *$p < 0.05$; **h:** $n = 5$ embryos, two-tailed paired $t$ test. Source data are provided as a Source data file.

identities in the spinal cord[23]. Thus, although our single-cell gene expression analysis has facilitated the identification of transcriptionally distinct cell types and states, integrating other modalities such as proteomics with our dataset would reveal more information about gene regulation during the cell-type determination process. Furthermore, while we have also attempted to cluster the collected MNs after integrating rostral and caudal datasets (Supplementary Fig. 13), we noticed several integration complications that disagreed with in vivo observations reported by previous and our studies. For example, a small subset of *Tac1*-expressing cells was not differentiated as a single cluster (Supplementary Fig. 13). Some cells expressing the LMC markers (*Foxp1* and *Aldh1a2*) were incorrectly assigned to another cluster which does not express these marker genes (Supplementary Fig. 13). Yet in our separate analysis, we identified the *Tac1*-expressing cluster of cells as the HMC MNs based on known marker expression (*Isl1$^+$Lhx3$^-$Foxp1$^-$*) and validation through *Tac1* in situ hybridization. Given that many previous elegant studies have provided rich knowledge on Hox-mediated segmental MN types[33], which could be served as a reference for us to annotate known segmentally-restricted types with confidence, thus, in our study, we presented separate analyses for rostral and caudal samples to prevent overlooking subtle sample type-intrinsic differences. This also highlights that reliable integration between biological conditions remains challenging[74], and future development with more advanced algorithms to overcome this issue is anticipated.

Overall, our study has deciphered the molecular repertoire of spinal MNs and conservation of subtypes across different vertebrates. Our identification of combinatorial TF/neuropeptide markers for each MN subtype could also facilitate a better understanding of axon targeting and selective neuronal connections, and help uncover differential susceptibility in disease models when seeking disease-protective pathways that could represent therapeutic targets.

## Methods

### Ethical compliance

All mice experimental procedures were performed in accordance to guidelines approved by the Institutional Animal Care and Use Committee (IACUC) at Academia Sinica (protocol number 19-12-1409 and 12-07-389). Human embryo spinal cords were harvested from material obtained following legally induced terminations of pregnancy (first trimester of pregnancy) from the Department of Obstetrics and Gynecology at the Antoine Béclère Hospital (Clamart, France). Fetal age was calculated by measuring the length of limbs and feet according to a developed mathematical model[75] and none of the induced abortions were performed for reasons of fetal abnormality. Tissues were collected in absence of compensation, with written informed consent specifying the purpose of the research, in accordance with legal procedures agreed by the French Biomedicine Agency, Agence de la Biomédecine (authorized project number PFS12-002).

### Mice

Female P50-P150 wildtype C57BL/6J (purchased from the National Laboratory Animal Center, Taiwan), Mnx1-RFP[73], Tau-mGFP[76] (JAX, 021162) and male P50-P180 Mnx1-GFP[17], Npy-IRES$^{Cre/Cre}$ (JAX, 027851)[77] mice were used in the study. Mice were crossed to obtain embryos of desired genotypes. When a copulation plug was observed, the embryo stage was estimated as E0.5. All live animals were maintained in a C57BL/6J background and housed in the specific-pathogen-free (SPF) animal facility of Institute of Molecular Biology, Academia Sinica, with 12-h light/dark cycle, 45–55% humidity, 19–22 °C temperature and had ad libitum access to food and water in their home cages at all times, abiding by the IACUC Academia Sinica guidelines.

### Sample collection for scRNA-seq

Mnx1-GFP mice were crossed to C57BL/6J mice for single-cell spinal MN collection. E13.5 embryos of pregnant mice ($n = 12$ embryos of two pregnant mice) were pooled to perform scRNA-seq. The embryos were decapitated and dissected to isolate spinal cords from rostral (C4-T3) and caudal (L1-S5) segments in Leibovitz's (L-15) medium. These segments were independently processed until single-cell library construction. Tissue dissociation was performed using a Neural Tissue Dissociation Kit (P) (Miltenyi Biotec, 130-092-628) on a gentleMACS dissociator (Miltenyi Biotec, 130-093-235) according to the manufacturer's instructions. Dissociated cells were resuspended in N2B27/DMEM-F12 and neurobasal medium containing N2 (Life Technologies, 17502048) and B27 (Life Technologies, 17504044), 1% penicillin–streptomycin, 2 mM L-glutamine, 0.2 M β-mercaptoethanol and 0.5 μM ascorbic acid, supplemented with 1% inactivated fetal bovine serum (FBS), 1:10,000 Dnase-I (Worthington Biochemical, LS006342), before filtering through a 70-μm strainer (Falcon, 352350). Sorting was carried out using a BDFACSAria III cell sorter (BD BioSciences, USA), with 85 μm nozzle diameter and 45 sheath pressure, to collect GFP$^+$ cells at 4 °C into DMEM medium with 1% FBS. Collected cells were counted and adjusted to a final concentration of 700–1200 cells/μl.

### Single-cell library generation

ScRNA libraries were generated using a 10× Genomics Chromium Controller Instrument (10× Genomics, Pleasanton, CA) and Chromium Single Cell 3' Reagent Kit v2 according to the manufacturers' instructions. In brief, single-cell suspensions were loaded on the Chromium Controller Instrument to generate single-cell Gel Bead-In-Emulsions (GEMs). Upon breaking up of GEMs, the barcoded cDNA was purified and amplified. During library construction, the amplified barcoded cDNA was fragmented, A-tailed, and then adapter-ligated. A sample index was added. Pooled libraries were sequenced using Illumina NextSeq 500 and NovaSeq S4 systems, and then paired-ended (read 1: 28 base pairs, read 2: 91 bp) to an average depth of 85k mean reads per cell.

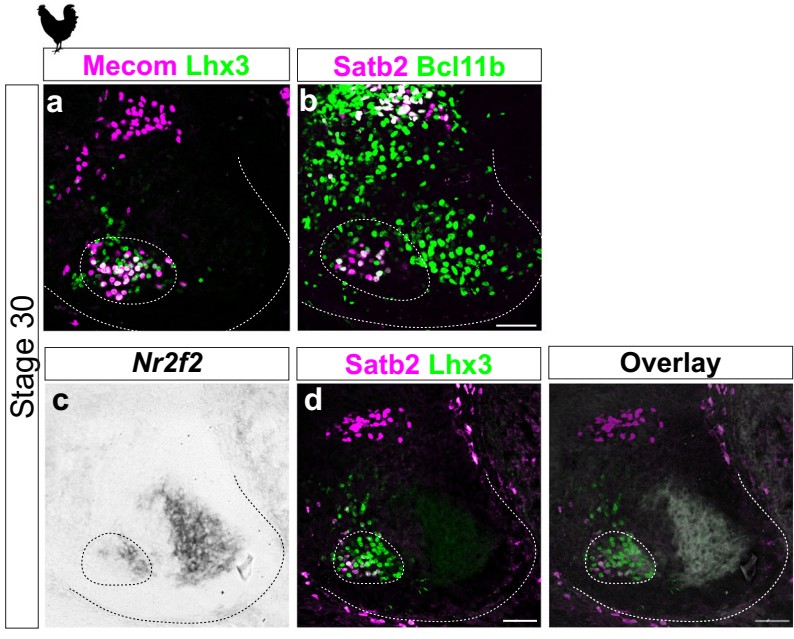

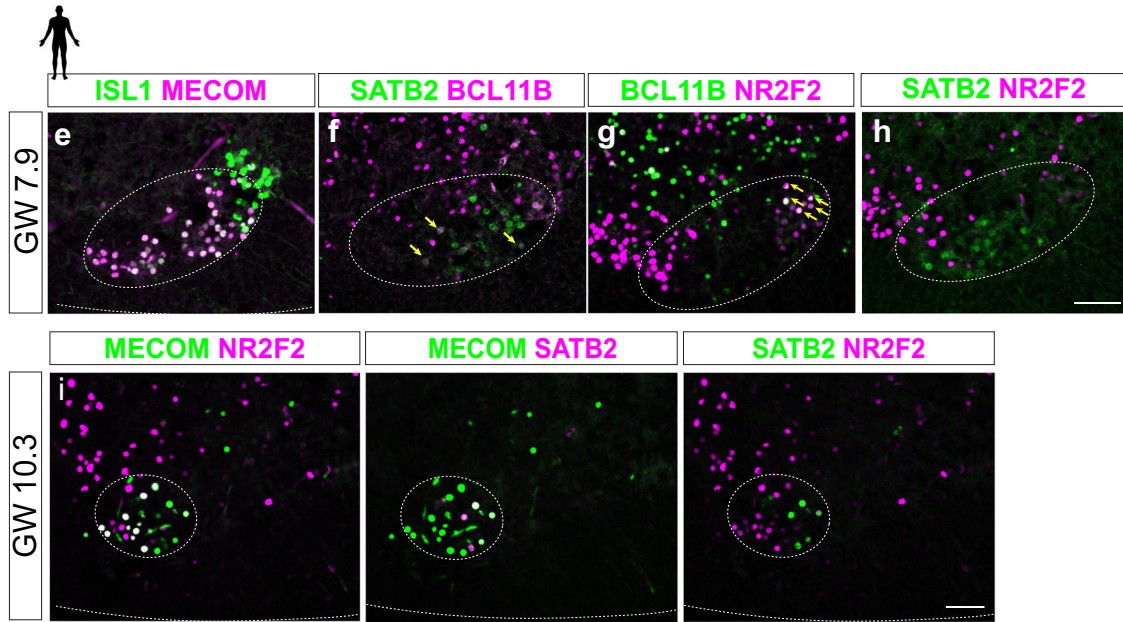

**Fig. 9 | MMC MN subpopulations are conserved in chicken and human spinal cord. a** Immunostaining for Mecom and Lhx3 in Stage 30 chick spinal cord shows that Mecom+ cells are MMC MNs. **b** Immunostaining for Satb2 and Bcl11b reveals expression in MMC MNs, with some cells exhibiting co-expression of these proteins. **c, d** In situ hybridization of **c** Nr2f2 and post-ISH immunostaining for **d** Lhx3 and Satb2 on the same spinal cord section. An overlay image of all markers (right) shows that Satb2 and Nr2f2 expression is mutually exclusive and displays a preferential mediolateral distribution. Dashed lines outline spinal cord boundaries and dashed circles demarcate MMC boundaries based on Lhx3 expression. **e–h** Immunostaining for MECOM and ISL1 in human embryo GW7.9 spinal cord shows that Mecom+ MNs are ISL1+ MNs. SATB2, NR2F2 and BCL11B are expressed in MMC MNs. MMC regions were outlined with dashed circles. Yellow arrows indicate co-expressing cells. **i** NR2F2 and SATB2 exhibit a mutually exclusive expression pattern in the MMC MNs of human embryo GW10.3 spinal cord. Staining was repeated on n = 4 chick embryos and n = 1 human embryo each for GW7.9 and GW10.3 on 2–3 different sections. Scale bar represents 50 μm.

## Preprocessing and quality control of scRNA-seq data

The FASTQ files were processed using the standard Cell Ranger pipeline (version 2.1.1, 10× Genomics) for demultiplexing, mapping to the mm10 reference, filtering, barcoding, and counting unique molecular identifiers (UMI). From the Cell Ranger pipeline, the total number of cells were 5528 and 5288, median numbers of genes detected per cell were 2843 and 2722, and the median numbers of unique transcripts were 11,906 and 11,090, respectively, for rostral and caudal samples. For both rostral and caudal samples, cells were retained for subsequent analysis if they displayed a number of genes between 1000 and 5300, UMI counts <30,500, and <10% mitochondrial counts. Quality filtering led to 5460 and 5242 cells for rostral and caudal samples, respectively, which were used for downstream analysis.

## Clustering analysis of scRNA-seq data

We performed normalization, dimensionality reduction, and cell clustering using Seurat package version 2.3.4[18]. The digital data matrices were normalized by a global method, whereby the expression

value of each gene was divided by the total expression in each cell and multiplied by a scale factor (10,000 by default). These values were then log-transformed with a pseudocount of 1. Highly variable genes across cells were selected using the FindVariableGenes function (parameters: x.low.cutoff = 0.0125, x.high.cutoff = 3, y.cutoff = 0.5). To identify cell clusters, principal component (PC) analysis was first performed using the top 34 PCs for the rostral sample and the top 31 PCs for the caudal sample, where the number of significant PCs was determined based on an elbow plot and jackstraw test. Cells were clustered using the Louvain community detection method in the FindClusters function (resolution = 0.05) and visualized using Uniform Manifold Approximation and Projection (UMAP). Motor neurons (MNs) and non-motor neurons were distinguished based on the expression of generic MN markers such as *Mnx1* and the cholinergic genes *Chat*, *Slc18a3*, and *Slc5a7*. Only clusters expressing MN markers were considered, retaining 5162 and 4699 cells for rostral and caudal samples, respectively. These MNs were analyzed using the same pre-processing procedures described above, but with different numbers of PCs and cluster resolutions.

To cluster MNs, cell clusters were obtained using the top 31 PCs with a resolution of 0.08 for the rostral sample and the top 24 PCs with a resolution of 0.3 for the caudal sample. Cell identities were assigned using known markers established in previous studies and markers identified in our study. Marker genes of each cell cluster were identified using the FindAllMarkers function. Genes were considered marker genes if: (i) the adjusted *p* values from the likelihood-ratio test was <0.05; (ii) the log fold-change was >0.25; and (iii) the percentage of cells expressing a given gene in the cluster was >25%.

To subcluster LMC neurons, cells were first grouped into "spatial quadrants" based on their expression pattern of *Hox* genes and LIM homeodomain factors. Cells in each "spatial quadrant" were then further clustered. In brief, for brachial LMC MNs, "spatial quadrants" were obtained by performing hierarchical clustering using *Hox* genes (*Hoxa5* and *Hoxc8*) and LIM homeodomain factors (*Lhx1* and *Isl1*). The robustness of assigned identities was confirmed by observing that the cells of a cluster within each "spatial quadrant" segregated from cells of a different cluster in UMAP space, which was established from PC analysis using highly variable genes as input. For the lumbar LMC MNs, "spatial quadrant" clusters were obtained using the FindClusters function with a resolution of 0.3. The robustness of the assigned identities was confirmed by predicting cell identities among lumbar LMC MNs from the assigned identities of brachial LMC MNs using a random forest model. This was achieved by training a random forest classifier based on the "spatial quadrant" identities of brachial LMC MNs using the ClassifyCells function in Seurat, which was subsequently applied to the lumbar LMC MNs.

To further subcluster LMC MNs in each "spatial quadrant" cluster, we adopted a multi-resolution ensemble method to determine the number of subclusters based on spectral graph theory[78]. Specifically, we first performed subclustering using the FindClusters function in Seurat with multiple resolutions ranging from 0.1 to 3 and an increment of 0.05. Next, we constructed a consensus matrix based on a set of clustering results from these multiple resolutions. Consensus matrix **A** was obtained by averaging the entries of a connectivity matrix **B** across all runs, where the entry $B_{ij}$ of the connectivity matrix **B** equals 1 if cells *i* and *j* are assigned to the same cluster for each run. Thus, each entry $A_{ij}$ for **A** varies from 0 to 1 and represents the probability of cells *i* and *j* being in the same cluster across multiple resolutions. The consensus matrix was further pruned by setting the elements to zero if they were <0.3 to ensure better robustness against noise. Finally, we estimated the number of clusters by computing the eigenvalues of the associated Laplacian matrix of the constructed consensus matrix. Based on perturbation theory and spectral graph theory, it has been theoretically proven that the number of clusters *N* equals the multiplicity of the eigenvalue 0 of the Laplacian matrix[78]. Therefore, in the ideal case of *N* completely disconnected clusters, the eigenvalue 0 has multiplicity *N*. More generally, the number of clusters *N* is usually given by the value of *N* that maximizes the eigenvalue gap (difference between consecutive eigenvalues), i.e., select the number *N* such that all eigenvalues $\lambda_1, \cdots, \lambda_N$ are very small, but $\lambda_{N+1}$ is relatively large. Once the number of clusters was determined, we ran FindClusters again to obtain the final clusters. In addition to this in silico inference, we also examined the heatmap of the top 10 marker genes of the identified clusters and ensured that the marker genes were biologically meaningful and that different clusters exhibited distinct expression patterns of those markers.

MMC MN subclusters were obtained using the top nine PCs with a resolution of 0.2.

## Similarity analysis between two groups of cells

Similarity between two groups of cells representing rostral or caudal MNs was determined using the calculateDistMat function (method = "trend") in the CIDER package[79]. In brief, differentially expressed signatures (DES) were first identified using limma-trend data for each group of cells against all other cells within the dataset, whereby DES were computed by fitting a linear regression model. The similarity was then measured according to the Pearson's correlation coefficient of DES between two groups.

To visually examine relatedness among LMC subclusters of brachial and lumbar segments, we merged all LMC neurons from the brachial and lumbar segments and then performed joint dimensional reduction via UMAP using the corrected embedding learned from the Harmony integration method[49]. To further quantify similarity among brachial and lumbar LMC subclusters, we assessed the degree of subcluster mixing across segments, quantified based on the overlap between one subcluster from brachial segments and another subcluster from lumbar segments in the joint UMAP space. Specifically, we first identified mutual nearest neighbors (MNNs) of each cell based on Euclidean distances in the UMAP space. Second, for each cell in one sample (e.g., brachial), we counted how many of its MNNs are cells in a certain subcluster from another sample (e.g., lumbar). Third, for each subcluster in the brachial sample, we computed its overlap ratio with another subcluster in the lumbar sample by calculating its average MNNs for cells from the lumbar sample, representing the probability of cells from two subclusters being MNNs across brachial and lumbar segments. Finally, to increase the robustness of the estimated similarity, we computed an average overlap ratio by using different numbers of neighbors (*k* = 10, 15, 20) to identify MNNs. The similarity matrix was visualized using the ComplexHeatmap R package[80]. To intuitively visualize the relationships of LMC subclusters between brachial and lumbar segments, we created an alluvial diagram using the ggalluvial R package (https://www.rdocumentation.org/packages/ggalluvial/versions/0.12.3). Lines of low similarity values (i.e., <0.09) were omitted to accentuate relatively substantial similarity and enhance readability. This threshold was heuristically searched within a range supported by prior knowledge, such as on Etv4[48], which should be shared among non-segment-specific clusters, as well as the relatedness of a given pair of clusters with other clusters in the UMAP space.

## Differential expression analysis and gene set enrichment analysis

Differentially expressed genes (DEGs) between segments or subclusters were identified using the FindMarkers function by performing likelihood-ratio tests. Genes with an adjusted *p* value <0.05 and log fold-change >0.25 were considered differentially expressed. Gene Ontology (GO) enrichment analysis of DEGs was performed using the clusterProfiler R package (v3.14.3, RRID: SCR_016884)[81], focusing on Biological Processes (BP) and Molecular Functions (MF). Redundant GO terms were removed using the simplify function with default parameters.

## Immunohistochemistry and in situ hybridization of mouse embryos

All E13.5 and E15.5 mouse embryos were fixed with 4% PFA for 2 h (for immunostaining) or 4 h (for in situ hybridization). To prepare samples for 20-μm cryosectioning, the samples were cryoprotected with 30% sucrose and embedded in OCT compound (Leica). For 300-μm vibratome sectioning, whole embryos were embedded in 4% low melting agarose and transversely sectioned. During the immunostaining step, sections were permeabilized and blocked with 10% FBS plus 0.3–0.5% Triton-X-100 for 1 h at room temperature or overnight at 4 °C. Antibodies were applied at respective titers and incubated at 4 °C for one or two nights for the 20- and 300-μm sections, respectively. Sections were washed frequently with wash buffer. For the 20-μm sections, samples were washed with 0.01% Triton X-100 in 1× PBS. For the 300-μm sections, an overnight wash with 0.5% Triton X-100 in 1× PBS (PBST) was performed. Secondary antibody was applied for 1 h at room temperature or overnight at 4 °C for the 20- and 300-μm sections, respectively. Finally, sections were washed with wash buffer and mounted. Primary antibodies used were: rabbit anti-Lhx3 (1:2000, Abcam Cat# ab14555, RRID:AB_301332); rabbit anti-Foxp1(1:20000, Abcam Cat# ab16645, RRID:AB_732428); goat anti-Foxp1(1:100 R&D systems Cat# AF4534, RRID:AB_2107102); rabbit anti-Hoxc8 (1:5000 Sigma-Aldrich Cat# HPA028911, RRID:AB_10602236); sheep anti-GFP (1:1000 AbD Serotec/ Bio-Rad Cat# 4745−1051, RRID:AB_619712); rabbit anti-RFP (1:500 Abcam Cat# ab62341, RRID: AB_945213); mouse anti-MNR2/MNX1/HB9 (1:50 DSHB Cat# 81.5C10, RRID:AB_2145209); goat anti-Chat (1:100 Millipore/Sigma Cat# ab144P, RRID:AB_2079751); mouse anti-COUP-TF2/NR2F2 (1:200 R&D systems Cat# PP-H7147-00, RRID:AB_2155627); mouse anti-COUP-TF1/NR2F1 (1:500 R&D systems Cat# PP-H8132-00, RRID:AB_2155494); rabbit anti-Evi1/Mecom (1:500 Cell Signaling Technology Cat# 2593, RRID:AB_2184098); goat anti-Isl1 (1:1000 Neuromics Cat# GT15051, RRID:AB_2126323); rabbit anti-Satb2 (1:1000 Abcam Cat# ab92446, RRID:AB_10563678); guinea pig anti-Satb2 (1:1000 Synaptic Systems Cat# 327004, RRID:AB_2620070); rat anti-Bcl11b/Ctip2 (1:1000 Abcam Cat# ab18465, RRID:AB_2064130); rabbit anti-Zfhx4 (1:200 Novus Cat# NBP1-82156, RRID:AB_11020060); rabbit anti-Calbindin D-28K/Calb1 (1:1000 Millipore/Sigma Cat# AB1778, RRID:AB_2068336); rabbit anti-Nfib (1:1000 Novus Cat# NBP1-81000, RRID:AB_11027763); rabbit anti-Grm5 (1:500 Millipore/Sigma Cat# AB5675, RRID:AB_2295173); sheep anti-Ebf2 (1:500 Novus Cat# AF7006, RRID:AB_10972102); rabbit anti-Nrp2 (1:200 Cell Signaling Technology Cat# 3366, RRID:AB_2155250). Guinea pig anti-Foxp1 (1:320000), anti-Isl1 (1:10000, cat# CU1277, RRID:AB_2631974), and anti-Lhx1 (1:20000, cat# CU453, RRID:AB_2827967), anti-Unc5c (1:50), rabbit anti-Nkx6.1 (1:1000) antibodies were gifts from Thomas Jessell and guinea pig anti-Mnx1 (1:1000) was a gift from Hynek Wichterle. Guinea pig anti-Hoxa5 (1:20000, RRID:AB_2744661) was made in-house. For secondary antibodies, donkey anti-Sheep IgG (H + L) Cross-Adsorbed, Alexa Fluor 488 (Thermo Fisher Scientific Cat# A-11015, RRID: AB_2534082); Donkey Anti-Goat IgG (H + L), Alexa Fluor 488 (Thermo Fisher Scientific Cat# A-11055, RRID: AB_2534102); 488-AffiniPure Donkey Anti-Rat IgG (H + L) (Jackson ImmunoResearch Lab, Cat# 712005153, RRID: AB_2340631); Cy3-AffiniPure Donkey Anti-Guinea Pig IgG (H + L) (Jackson ImmunoResearch Lab, Cat# 706165148, RRID: AB_2340460); Cy3-AffiniPure Donkey Anti-Mouse IgG (H + L) (Jackson ImmunoResearch Lab, Cat# 706165150, RRID: AB_2340813); Cy3-AffiniPure Donkey Anti-Rabbit IgG (H + L) (Jackson ImmunoResearch Lab, Cat# 711165152, RRID: AB_2307443); Cy5-AffiniPure Donkey AntiGuinea Pig IgG (H + L) (Jackson ImmunoResearch Lab, Cat# 706175148, RRID: AB_2340462); Cy5-AffiniPure Donkey Anti-Mouse IgG (H + L) (Jackson ImmunoResearch Lab, Cat# 715175150, RRID: AB_2340819); Cy5-AffiniPure Donkey Anti-Rabbit IgG (H + L) (Jackson ImmunoResearch Lab, Cat# 711175152, RRID: AB_2340607) were used at dilution titer of 1:1000.

For in situ hybridization, sections were dried, post-fixed with 4% PFA for 15 min at room temperature. Slides were pretreated with 3 μg/ ml Proteinase K for 5 min and acetylation buffer for 10 min. Pre-hybridization was performed for at least 2 h at room temperature. Riboprobes (150 ng) were heat-denatured at 80 °C for 5 min and hybridized to sections overnight at 58 °C. After washing and blocking, the slides were incubated with anti-digoxigenin-AP, Fab fragments overnight at 4 °C. After washing, the slides were color-developed with NBT/BICP solution. Sequences for riboprobe generation are indicated in Supplementary Table 1, and the template DNA was amplified either by cloning or polymerase chain reaction. All images were acquired using either Zeiss LSM780 or LSM980 confocal microscopes and an AxioImager Z1 upright fluorescence microscope, processed with Zen 2012 blue edition (Carl Zeiss). Images of axonal tracing are projections of z-stacks.

For mediolateral spatial delineation during LMC marker validation, adjacent slides were immunostained for spatial genes to determine the position of signals (rostral: *Hoxa5*; caudal: *Hoxc8*; medial: *Foxp1$^+$ Isl1$^+$*; lateral: *Foxp1$^+$ Isl1$^-$*). Boundaries were drawn and transferred to slides with marker staining.

## RNAScope

Neuropeptide transcripts were detected using RNAscope Multiplex Fluorescent Reagent Kit v2 (323100) and probes were detected with Akoya Biosciences Opal 520 (FP1487001KT), 570 (FP1488001KT) and 690 (FP1497001KT). The staining protocol followed the manufacturer's recommendations with minor modification. E13.5 embryos were fixed at 4% PFA for 30 h at 4 °C and washed with 1× PBS at 4 °C. Embedded samples were subjected to 14-μm cryosectioning. The fresh-frozen protocol was adapted by skipping the 15 min post-fix stage and proceeding directly with pretreatment and the RNAscope hybridization assay. Details of probes used in this study are presented in Supplementary Table 1.

## Quantification of MN subtype number

To quantify Nkx6.1$^+$ and Zfhx4$^+$ immunostained cells, sequential sections of 20 μm every 450 μm were categorized into four regions along the rostrocaudal axis of the brachial or lumbar spinal cord. Cell segmentation masks were first generated using Ilastik[82] for individual channels, and object center coordinates $(x,y)$ of expressing cells were outputted to ImageJ CellCounter (NIH) for manual inspection and quantification. Cells were considered as co-expressing if the $x,y$ coordinates from different channels were both unique within a diameter of 10 pixels.

For quantification of neuropeptide-expressing cells by RNAscope-based ISH, sequential sections of 14 μm every 500 μm were categorized into three regions along the rostrocaudal axis of the brachial or lumbar spinal cord. 3D cell segmentation masks were produced based on z-stack images of DAPI. Masks were imported into Imaris for surface detection. RNAscope signals were detected with spot detection and split into surfaces. Nuclei were counted as expressed when ≥3 spots within a surface were detected.

## Human embryonic spinal cord sample collection and immunostaining

The sample preparation and immunostaining procedures were performed similar to mouse embryos, except that spinal cords were fixed with fresh cold 4% PFA for 90 min and sectioned at 16 μm. After the immunostaining, sections were mounted in Fluoromount (Sigma-Aldrich or Cliniscience). All images were acquired with a DM6000 microscope (Leica) and a CoolSNAP EZ CDD camera.

## Analysis of the spatial distribution of MMC MN subpopulations

Coordinates $(x, y)$ were assigned based on the position of each MMC MN using the "spots" function in the imaging software Imaris 9.5.1 (Bitplane). Four additional coordinates were exported, i.e., midpoints for the dorsoventral and mediolateral spinal cord boundary. To

account for differences in spinal cord size and shape, sections were normalized to a standardized hemisection spinal cord (midline to lateral = 400 μm; dorsal to ventral edge = 800 μm). Density distributions for cells were plotted using the ggplot2 'geom_density' function.

## Statistics and reproducibility

Statistical analysis was performed by GraphPad Prism 6.0 (GraphPad Software). All statistical results are presented as mean ± SD (standard deviation) of three or more independent biological replicates or the median ± interquartile range (IQR), as indicated. Appropriate statistical tests are performed for each analysis and specified in the respective figure legend. All staining experiment were repeated on at least three independent biological replicates unless indicated otherwise and representative images were presented in the manuscript. No statistical method was used to predetermine sample size. The experiments were not randomized nor blinded because the quantification analysis and the assessment was performed in an objective and semi-automated manner.

## Reporting summary

Further information on research design is available in the Nature Portfolio Reporting Summary linked to this article.

## Data availability

The E13.5 mouse spinal MNs scRNA-seq raw data generated in this study have been deposited in the Gene Expression Omnibus repository and are accessible through accession code GSE183759. The public datasets of adult spinal MNs used in this study are available at http://spinalcordatlas.org. All data supporting the findings of this study are available within the article and Supplementary Information. Source data are provided with this paper.

## Code availability

The computational codes and metadata are publicly available on GitHub (https://github.com/sqjin/scRNA-seq-motor-neuron) and Zenodo repositories[83].

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

## Acknowledgements

We thank Hynek Wichterle, Tom Jessell, and Susan Morton (Columbia University) for many gifts of antibodies used in this study. We thank Virginie Rouiller-Fabre and Nour Nicolas (CEA, Fontenay aux Roses, France) for providing human tissues. We acknowledge the FACS, Transgenics, Genomics, Bioinformatics and Imaging cores of IMB Academia Sinica for their substantial technical assistance, and Dr. John O'Brien for further editing the manuscript. We thank Dr. Jr-Kai Yu, Dr. Yi-Hsien Su, Dr. Shen-Ju Chou and Dr. Suewei Lin (Academia Sinica) for valuable inputs and members of the JAC lab for discussion and proofreading. J.-A.C. is supported by Academia Sinica (CDA-107-L05 and AS-GC-109–03), MOST (110-2326-B-001-009, 109-2314-B-001-010-MY3, 108–2311-B-001–011) and NHRI (NHRI-EX110-10831NI) grants. Q.N. is supported by NSF (DMS1763272) and Simons Foundation grants (594598, Q.N.). Schematic diagrams in Figs. 1b and 9 were created with BioRender.com.

## Author contributions

J.-A.C. and E.S.L. conceptualized and designed the project.; E.S.L. performed most of the genomics and experimental work, assisted by W.-S.L.; M.C. performed the immunofluorescence experiments in the human embryo; S.J., E.S.L., and Y.-C.C. conducted the computational analyses; J.-A.C. and Q.N. provided supervision and funding; J.-A.C. and E.S.L. wrote the original draft and the manuscript was revised with help from S.J., Y.-C.C., and S.N.

## Competing interests

The authors declare no competing interests.
