## [Peer Review File · Nature Communications]

Single-cell transcriptomic analysis reveals diversity within
mammalian spinal motor neuronsREVIEWER COMMENTS

Reviewer #1 (Remarks to the Author):

Here the authors use single-cell RNA sequencing to describe the diversity of spinal motor neurons. They sorted motor neurons from the spinal cord using an *Mnx1*-GFP transgenic animal. Their analysis finds two novel MN subtypes, as well as novel heterogeneity within the LMC and MMC subtypes. Their findings also suggest that these subpopulations are present in multiple species. Although the manuscript's conclusions seem largely sound, they might be strengthened by addressing the questions below:

Major:

1) I did not see a doublet-removal procedure in the methods, and I think it would be useful to run something like DoubletFinder to help determine if any of these clusters might contain doublets.

2) Is it correct that only one caudal sample and one rostral sample were processed? Additional replicates are not likely to negate the novel heterogeneity the authors describe, but lack of replicates might affect comparisons made across samples (Fig 6). For example, can you rule out that batch effects underlie the up/down-regulated pathways across brachial and lumbar samples (6b)? Or that some of the segment-specific clusters are only specific due to sampling differences?

3) Also regarding Figure 6:

- What is the rationale for using Harmony? If Harmony is not used, are all clusters segment-specific? And might this conclusion, in some sense, be more accurate (i.e. could it be biological reality that all spinal cord cell types are segment specific?) In Supp Fig 7a, some genes clearly distinguish brachial and lumbar cells that are considered the same "group." This observation leads me to wonder how a threshold is chosen for considering a cluster segment-specific (and more broadly, if any threshold can be justified). Would it make more sense to skip Harmony and call all clusters segment-specific? Cross-segment comparisons would still be useful.

- Why not calculate mutual nearest neighbors on the PCA instead of the UMAP projection? The UMAP is a 2D representation where distances between cells are not necessarily interpretable.

- It would be nice to see what the rm/cm clusters look like on the UMAP in 6d/e.

4) My understanding is that cluster markers were determined mainly by fold change in expression. As a result, some of the markers are not very specific, and I wonder if making use of the *pct1/pct2* value in the differential expression table could help find markers that are easier to use for validation experiments (i.e. "on" in one cluster and "off" in another). Two examples:

- Fig 2E: I find it difficult to see that *Grm5* and *Nfib* co-localize only in the sacral stainings and not in the other sections. Is this clearer in magnification? Or would it be possible to use a more specific marker than *Grm5*?

- Fig3B/C, Line 191: The phrase "Lsamp-expressing MMC neurons" is a little confusing, because *Lsamp* is expressed in both MMCs and LMCs (expression is higher in MMCs, but still >60% of LMCs express *Lsamp*).

Minor:

Fig 1:

- A: The schematic might be clearer if MMC and LMC are labeled everywhere relevant, also caudally.

- D. Can you comment on the cluster labeled "unknown?"

- E, F: Is it possible to elaborate on how these bar plots are made? Are the bins the same for every gene? Why not use a dot plot to be consistent with other figures?

Fig 2:

- A, Line 158: What is the rationale for looking at *Chat*? Does it provide more information than *Mnx1* alone?

- C: At P7, it's hard to appreciate the overlap between Calb1 and Chat in the image magnifications without zooming a lot. Is it possible for an overlay also to be magnified?

Fig 4:

- C: On the UMAP some clusters are labeled caudal despite expressing both Hoxa5 and Hoxc8 (contrary to the staining in A). Is this expected?

- D: It would be nice to see a dot plot for these genes instead. I don't think all are listed in 4C.

Fig 5

- A: How is the Gini index calculated? Is it possible that the index is affected by either: (1) the number of genes belonging to each gene set, or (2) the average expression levels in a gene set? i.e. Are either of these values correlated with the index?

- G: What are the abbreviations?

Fig 7

- C, Supp Fig 8c: From Fig7b, I would have expected that Nr2f2 and Bcl11b are co-expressed less often than Satb2 and Bcl11b. Could this co-expression actually be a fourth cluster? Only a subset of the Bcl11b cluster seems to be Nr2f2+. Maybe these cells can be distinguished by another marker?

Supplementary Figure 1D: Why is Mnx1 absent from so many cells if an Mnx1-GFP line was used? Is Mnx1 maintained only at the protein level?

Reviewer #2 (Remarks to the Author):

Liau et al. characterize the diversity of motor neuron subtypes in the embryonic spinal cord. To do so, they isolate motor neurons from E13.5 Mnx1-GFP embryos and perform single cell RNA sequencing using the 10X approach. Using bioinformatic analysis the authors first reveal previously unrecognized motor neuron subtypes and provide experimental evidence for their existence in the embryo. They subsequently focus on lateral motor column (LMC) neurons. By using clustering approaches, the authors identify combinatorial codes of transcription factors (TFs) and neuropeptides that define distinct LMC subtypes and demonstrate that they can approximate where these subtypes are located based on the expression of marker genes known to partition LMC neurons into medial-lateral and rostral-caudal identities. Lastly, the authors demonstrate that medial motor column (MMC) neurons can be further partitioned based on the expression of the TFs Satb2, Bcl11b and Nr2f2, provide a first indication that this division is also functionally significant as these subtypes display distinct axonal projection patterns and demonstrate that this subdivision is evolutionary conserved between mouse, chicken, and human.

Overall, this is a well-executed study and a well written manuscript that I enjoyed reading. The single cell transcriptome data will undoubtedly help to further decipher how motor neuron diversity is established during development and will be an important resource for the scientific community in the future. Furthermore, the finding of a combinatorial code of neuropeptides in LMC neurons has the potential to be highly significant to understand how motor neuron diversity is established. The bioinformatic analysis is solid and observations made in the single cell transcriptome data are thoroughly validated experimentally. I have few comments that I am sure the authors will be able to address during revision.

1) Subclustering of the LMC data (lines 241 – 244): The authors state that they can "uncover previously uncharacterized motor pools that innervate individual muscle groups". This statement is based on the fact that some of the clusters they identified indeed express known markers for distinct motor pools. However, this does not mean that the other clusters also correspond to specific motor pools or even groups of motor pools. In fact, some of these clusters must comprise neurons from multiple motor pools

given that, as the authors state themselves in the discussion, the number of LMC subtypes they uncover does not match ~60 expected motor pools.

2) Related to the previous point, the authors show for many quantifications in the paper mean expression levels and % expressed cells as is widely-used standard in the field. From these quantifications it is clear that in most cases not all cells in a cluster express a specific marker. This may obviously be due to drop-out events in the single cell sequencing but could also reflect true biological heterogeneity. For example, the authors chose *Zfmx4* for their proof of principle validation experiment, which is specifically expressed in rl1 and rl2 subclusters, however, in both subclusters less than 50 % of cells have reads for this gene. Is this also true when looking at the protein level?

3) Line 263 – 266: I am not sure which message the authors want to convey here and why it is “remarkable that the *Etv4+* subcluster expressed *Sst*, whereas the *Pou3f1+* subcluster displayed *Penk* expression.” Do the authors want to indicate that the subcluster-specific TFs might regulate the expression of the neuropeptides? Related to this point, the authors describe combinatorial codes of TFs and neuropeptides in LMC neurons. How well do these 2 codes match and is there a close correlation between the expression of specific TFs and neuropeptides, which may be indicative of specific TFs regulating the expression of specific neuropeptides?

4) Because the body axis is sequentially generated from anterior to posterior, the lumbar part of the spinal cord is typically developmentally delayed compared to the brachial region. Consistently, GO term analysis revealed that lumbar LMC neurons were enriched for terms related to MN cell fate specification and development, which may indicate an earlier maturation stage. This raises the question to which extent the results of the differential gene expression analysis between brachial and lumbar LMC neurons may be biased by neurons in these regions being potentially at different stages of their maturation process instead of revealing true differences due to axial identity? One possibility to test this hypothesis could be to compare the E13.5 lumbar LMC sequencing data to single cell transcriptome data from earlier stages, e.g. from E12.5 motor neurons from Li et al. *Mol Syst Biol.* (2021) 17:e9945.

Reviewer #3 (Remarks to the Author):

This detailed single cell transcriptomic description of embryonic motor neurons by Liao et al. is compelling evidence for the molecular diversity of motor neuron subtypes whose existence has long been hypothesized but for which specific markers were for the most part unknown. In this study, the authors perform single cell sequencing of *Mnx1*-GFP positive motor neurons (MNs) of 2 regions of the developing spinal cord at embryonic day E13.5 (brachial and lumbar). They discover new MN subtypes and define new markers for known and novel MN subtypes. They reveal combinations of transcription factors that define motor neuron subclusters likely corresponding to motor pools. The authors show these clusters are also linked to a neuropeptide code which they hypothesize might play a functional role in the formation of specific connections between motor pools and their muscle targets. Finally, they convincingly demonstrate the existence of 3 subtypes of MMC neurons in the mouse and that these exist in other species (chick and human).

This work merits publication as an important resource for the community of researchers interested in motor neuron biology. However, several important issues should first be addressed.

The single cell RNA sequencing (scRNAseq) data presented here is of high quality, and the interpretation is solid and reasonable. However, the validation of the scRNAseq data in tissue is often quite limited, to the point where some claims are validated by a single example image. The authors should more rigorously validate their scRNAseq data with quantifications of their in situ hybridization and immunostaining data to help support their claims. Furthermore, demonstrating co-expression of combinatorial markers that define novel cell types rather than showing the presence of markers in a similar region

in different sections would be a more appropriate and powerful validation of the scRNAseq data.

Another issue is that the authors may be inflating the true number of MN subclusters by over-subclustering their data, in particular in the reported diversity of LMC subtypes.

“Within limb MNs, we reveal transcriptional codes for 26 subclusters corresponding to putative motor pools and demonstrate unexpectedly diverse expression of neuropeptides in the LMC MNs.”

Unless I missed it, there was no clear figure representation of the 26 clusters. If the 26 arise from adding up the brachial and lumbar clusters, then this number is not convincing unless the authors integrate brachial and lumbar and show that clusters shared between the 2 regions are distinct from one another. It is not clear this will be true since some of these appear to share identity, for example, in Figure 6a, brachial cm2 = lumbar cm1.

Finally, relating to the discovery of 3 distinct subtypes of axial MNs:

The data presented in Figure 7 was some of the strongest data in the manuscript, though I was not entirely convinced by the validation for the “different axon guidance signatures” (Fig 7f-h). If there is a better way to show one branch projecting to its target, or additional examples, the support for this claim would be substantially stronger.

Below is a detailed list of comments to address, in the order in which the items appear in the manuscript:

The authors should call attention to the fact that their dataset is missing the thoracic T4-T13 region; this is important background info to appreciate. It is indirectly shown in Figure 1b and reported in the methods, but some mention in the main text would be helpful.

Since the authors used a Mnx1-GFP line to isolate the neurons, can they comment on the relatively low expression levels of cholinergic and spinal MN markers in their dataset (Suppl Fig 2d) – eg Chat and Mnx1 itself.

Figure 1c-d: Why didn't the authors first integrate the data to highlight the differences between the rostral and caudal spinal cord levels? Starting with a view of the data integrated as a whole, and later separated out, would be a more intuitive way to present the data. If there is a reason why the authors did not do it, they should provide an explanation, but it would be preferable to provide this integrated view.

“First, we focused on previously unreported and specifically expressed TFs (i.e., Zfhx3, Zfhx4, Foxp4 and Casz1), since motor pool-specific TFs are usually the major determinants for establishing motor pool identity and properties. As a proof of principle validation experiment, we chose Zfhx4” – Can the authors add a quantification for Zfhx4 (one example image is not sufficient validation). It would be preferable to see at least one additional TF validated.

Figure 4h Line 238-239 – “We corroborated its sole expression in a subpopulation of LMC MNs in the rostral lateral region of the brachial segment (Fig. 4h)”. The authors should quantify the prevalence of their markers in the different quadrants, or at least add a quantification illustrated by this example (4h).

Combinatorial expression of neuropeptides and correlation of TFs in the given LMC subcluster is very interesting.

Please clarify: lines 263-266 “Remarkably, we also detected that the Etv4+ 263 subcluster (cm1)

expressed Sst, whereas the Pou3f1+ 264 subcluster (cm3) displayed Penk expression (Fig. 5e, 5f), implying that these motor pool TFs might co-regulate with neuropeptides

to provide subtype-specific physiological roles.”

Are the authors suggesting the TF combinations determine neuropeptide expression? Or just that motor TF's in combination with neuropeptides provide subtype specific physiological roles?

It is unclear why the authors specifically call out neuropeptide expression as unique. From their figure 5a, it's TFs, ligands/receptors and neuropeptides all stand out as important markers. Every category of cell type that is distinct will have some TFs/ligands/neuropeptides that are unique to it.

The authors should consider exploring if any of this distinction is retained in the adult by comparing to data from Blum et al., 2021 and Alkaslasi et al., 2021, and whether (or how much of) this code is erased after developmental connections are complete. A parallel question is whether there is any correspondence of the LMC or MMC clusters detected here with the adult skeletal MN clusters described in Blum et al or Alkaslasi et al.

Figure 5c: the dotted region showing distinction between m and l is in the wrong position.

And a related note: in the methods, the authors should provide an explanation for how the dotted lines separating lateral vs medial LMC are defined.

Figure 5e: Pou3f1 is red in the label but magenta in the image.

Figure 5f: the idea for a combinatorial code of markers defining each motor pool is compelling. To make this data fully convincing, the authors should perform multiplexed in situ hybridization for some of these combinations of transcription factors + neuropeptides to demonstrate their existence. For example in Fig 5e, it would be better to see Penk and Pou3f1 expressed in the same cells, not separate staining in neighboring sections.

Figure 5g - please define the arrows.

Fig 5g is very nice evidence of the specificity of Npy to just a subset of brachial nerves. What do Npy-Cre projections look like in the lumbar region? (re. Fig 6c).

Figure 6 nomenclature is confusing: there is no definition of "crl" shown in 6f and 6g. Also in the text: line 217 - add definition of "crl" as show in Fig 6f and 6g

Lines 285-287 - "In contrast, neuropeptide-related terms were more enriched in the brachial LMC MNs (Fig. 6b), and we confirmed higher expression of Grp and Npy neuropeptides in the brachial LMC MNs by means of in situ hybridization (Fig. 6c)." To support this claim, the authors should add a quantification for data represented in Fig. 6c.

Lines 309-312: "but 5 and 2 LMC subclusters were specific to the brachial or lumbar segment, respectively. For example, the Grp+ cl3 and rl5 subclusters are present in the brachial segment alone, whereas the Nkx6.1+ 311 crl6 subcluster predominantly occurs in the lumbar segment (Fig. 6g, 6i; Supplementary Fig. 7b)." The authors should show this specificity in tissue.

Is each point one section in 7d? If so, please could the authors show the data by animal.

Figure 7g: the yellow and orange arrows are much too small to see. Please provide higher magnification.

Lines 347-350: "Collectively, our results indicate that, similar to LMC MNs, MMC subtypes exhibit preferential mediolateral distributions and they selectively express TFs, receptors and axon guidance molecules that might facilitate high fidelity axonal innervations to the

correct epaxial muscles.”

This interpretation sounds reasonable, but the data shown in Fig 7h are not very compelling. From the figure, it looks like *Unc5c* is also present in the *Nrp2+* branch. Also, the pink line tracing *Nrp2* obscures the actual signal, which is not ideal.

Figure 8c: This figure would be stronger by showing *Nr2f2* + *Satb2* together in same image. If there is a reason why this was not shown please explain why this is not feasible.

Figure 8f-g: higher magnification for these images is required to see the cells indicated by the arrows.

We sincerely thank the reviewers for appreciating the importance of our study and for their insightful comments. Our point-by-point responses to the reviewers' comments and descriptions of the new experiments and computational analyses we have now conducted are provided in this letter. The original reviewers' comments are in **black**, followed by our responses in **blue**. We highlight changes made to the text and figures in **yellow**, whereas **cyan** indicates the paragraph line numbers in the revised text. All changes are highlighted in the revised text. We have made changes to the figure numbering to address the reviewers' comments. To avoid confusion about the numbering, we provide a conversion table here for the reviewers' and editor's reference.

Table R1. Changes to figure numbering and notes for revision.

Old Numbering	New Numbering	Content	Note for changes during revision	In response to
Figure 1	Figure 1	Clustering and annotation of MN subtypes	Added labels to panel a, changed panels e and f.	Reviewer 1
Figure 2	Figure 2	Identifying new subtypes	Added enlarged image to panel c and new image to panel e, moved original panel e to S3b.	Reviewer 1
Figure 3	Figure 3	New markers validation	Unchanged	
Figure 4	Figure 4	LMC MNs subclustering	Moved original panel g to S6a, added panel g and h, dot-plot to panel c.	Reviewer 1, 3
Figure 5	Figure 5	Examining differential neuropeptide expressions	Changed panel e, added new panel g and h (right).	Reviewer 2, 3
Figure 6	Figure 6	Differential gene expression between LMC in fore- and hindlimbs	Moved panels d - h to Fig. 7. Added new panels c-g.	Reviewer 1, 2, 3
	Figure 7	LMC cluster comparisons between fore- and hindlimbs	Added new images to panel c, h and quantification to i.	Reviewer 1, 2, 3
Figure 7	Figure 8	MMC diversity	Changed panels d, g and i, added quantification to h.	Reviewer 3
Figure 8	Figure 9	MMC subtype conservation	Changed panels d (overlay), enlarged figures e-h.	Reviewer 3
Figure S1	Figure S1	Sample preparation criteria	Added panel d.	Reviewer 1, 3
Figure S2	Figure S2	QC filtering of single-cell sample	Unchanged	
Figure S3	Figure S3	Novel markers and subtypes identification	Added new panels a and b.	Reviewer 1
	Figure S4	Differentially expressing genes within and between subtypes	Added new panel c.	Reviewer 3
Figure S4	Figure S5	Brachial LMC subclustering	Unchanged	
Figure S5	Figure S6	TFs and neuropeptide genes validation in LMC.	Added new panels a and b.	Reviewer 3
	Figure S7	Cluster comparison between E13.5 and adult MNs.	New Figure	Reviewer 3
	Figure S8	Gene expression comparison between E13.5 and adult MNs.	New Figure	Reviewer 3
	Figure S9	Consistent differential neuropeptide expression between LMC in fore- and hindlimbs across development.	New Figure	Reviewer 1, 2, 3

Figure S6	Figure S10	Lumbar LMC subclustering	Moved panel f to S11b.	
Figure S7	Figure S11	Cluster comparison between brachial and lumbar LMC	Added new panel a. Panel b originated from S10f.	Reviewer 1
Figure S8	Figure S12	MMC MN diversity	Added new panel b.	Reviewer 1
	Figure S13	Integration analysis of all cells in this study	New Figure	Reviewer 3

In addition to the new 9 main and 13 supplementary figures shown in the manuscript, we also attach 4 specific figures that address the reviewers' comments, listed below. We would be happy to incorporate these into the revised manuscript if it is felt that is warranted.

Additional Revision Figures:

- R1:** Doublet identification using Doublet Finder in both samples of spinal MNs collected.
- R2:** Examples of housekeeping genes indicating comparable expression levels across both samples.
- R3:** Dot-plot of known marker gene expressions in *Drosophila* optic lobe neuronal types. Dataset from Ozel et al., 2021.
- R4:** Analysis of transcription factor (TF)-neuropeptide co-regulation. **(a)** An *Etv4* inducible line was generated, differentiated from mouse embryonic stem cells into postmitotic MNs, and then harvested to assess **(b)** changes in neuropeptide expression by means of quantitative RT-PCR. **(c)** Heatmap showing correlated expression between TF (y-axis) and neuropeptide (x-axis) genes.

Reviewer #1

Here the authors use single-cell RNA sequencing to describe the diversity of spinal motor neurons. They sorted motor neurons from the spinal cord using an *Mnx1*-GFP transgenic animal. Their analysis finds two novel MN subtypes, as well as novel heterogeneity within the LMC and MMC subtypes. Their findings also suggest that these subpopulations are present in multiple species. Although the manuscript's conclusions seem largely sound, they might be strengthened by addressing the questions below:

We very much appreciate this reviewer's encouragement. Below, we address all of his/her comments in detail.

Major:

1) I did not see a doublet-removal procedure in the methods, and I think it would be useful to run something like DoubletFinder to help determine if any of these clusters might contain doublets.

Response:

In light of the reviewer's suggestion, we have now performed a doublet analysis for our rostral and caudal scRNA-seq data using DoubletFinder, as recommended. For each individual sample, we ran DoubletFinder with default parameters using the Seurat object as an input. Through predictive singlet and doublet analyses visualized in UMAP space, we observed only very few doublets, which were distributed in random groups (Fig. R1). Accordingly, the frequency of doublets in our final dataset is very low. Thus, we think these doublets are unlikely to affect our conclusions, and so we have retained our original analysis in the revised manuscript.

R1: Doublet identification using Doublet Finder in both samples of spinal MNs collected.

2) Is it correct that only one caudal sample and one rostral sample were processed? Additional replicates are not likely to negate the novel heterogeneity the authors describe, but lack of replicates might affect comparisons made across samples (Fig 6). For example, can you rule out that batch effects underlie the up/down-regulated pathways across brachial and lumbar samples (6b)? Or that some of the segment-specific clusters are only specific due to sampling differences?

Response:

We understand the reviewer's concern and agree that we cannot rule out batch effects when performing differential expression analysis on rostral and caudal samples. To assess how much variability is associated with batches, we first used housekeeping gene expression that is relatively

invariant across batches to explore differential expression across rostral and caudal LMC samples. Among the 25 stable mouse housekeeping genes we assessed (defined in a previous study¹), we found that none exhibited a significant difference in expression across brachial and lumbar LMC samples (Fig. R2). Among 2716 mouse housekeeping genes, 94% remained comparable without showing a significant difference in their expression across rostral and caudal LMC samples. These results indicate that the differentially enriched genes or pathways we identified across rostral and caudal LMC samples likely reflect biological differences. Additionally, we reasoned that the batch effect could result in false positive findings and be absent *in vivo*. To examine that possibility, we

have made tremendous efforts to strengthen our conclusions by performing additional experiments for this revision (new Figures, including Figs. 6c-g, 7h-i, and Supplementary Fig. 9a, b).

We confirmed some brachial-enriched (i.e., *Grp* and *Npy*) and lumbar-enriched (i.e., *Igf*) genes identified from single-cell analysis using a fluorescence *in situ*

hybridization (RNAscope) approach and examined their expression along the brachial and lumbar spinal cord sections (new Fig. 6c-g and Supplementary Fig. 9a, b). Consistent with our scRNAseq analyses, *Grp* was expressed in LMC subsets within the brachial spinal cord, but it was absent from the lumbar segments. Likewise, *Npy* was significantly enriched in the most rostral brachial spinal cord sections. In contrast, *Igf1* exhibited a higher ratio of expressing cells in the lumbar segments. **These new results corroborate that the segment-specific neuropeptide-expressed pathways we uncovered are not due to sampling artifacts.**

Finally, we also used *Nkx6.1* as a proof-of-principle to verify that the identified segment-specific LMC subcluster (i.e., *crl6*) could not be attributable to sampling differences. Concordant with our scRNAseq-based annotations (Fig. 7g), we observed a tiny population of *Nkx6.1*⁺ *Isl1*⁻ cells specifically located in the caudal region (~region 4) of the lumbar segments. However, no such population occurred in the brachial part (new Fig. 7h-i). Thus, our new RNAscope *in situ* hybridization and immunostaining experiments, and respective quantifications, strengthen that the up/down-regulated pathways and the differential clusters across brachial and lumbar samples we report in our study are not due to sampling differences.

In our revised manuscript, we have revised the section to (lines 308-316):

“To validate that the observed differences between limbs are bona fide biological variations rather than batch sampling artifacts, we performed RNAscope multiplexed fluorescence *in situ* hybridization and quantified cell numbers to confirm higher expression of *Grp* and *Npy* neuropeptides in the brachial LMC MNs (Fig. 6c and 6d; Supplementary Fig. 9a). Additionally, we identified important neuropeptide genes such as *Igf1*², known to exert a neuroprotective function, as being differentially enriched in *Hoxd10*⁺ medial LMC MNs in the lumbar segment (Fig. 6c and 6f). Similarly, we observed that these neuropeptide-expressing populations are

dynamically distributed across the rostrocaudal axis of the brachial or lumbar spinal cord (Fig. 6d-6g; Supplementary Fig. 9a, 9b), corroborating their aforementioned cluster-specific expression (Fig. 5b).”

(lines 342 – 349)

“Nineteen subclusters, including *Etv4*⁺, were found to have correspondences between the brachial and lumbar segments (Fig. 7f; Supplementary Fig. 11c), but five and two LMC subclusters were specific to the brachial or lumbar segment, respectively. For example, the *Grp*⁺ cl3 and rl5 subclusters are present in the brachial segment alone (Fig. 7g; Supplementary Fig. 11d). Although *Nkx6.1* is known to be expressed in certain *Isl1*⁺ LMC MNs in brachial and lumbar segments³, we further identified a *Nkx6.1*⁺ *Isl1*⁻ subpopulation, presumably the *crl6* cluster, which is only located in the caudal part of the lumbar segment (Fig. 7g-7i).”

3) Also regarding Figure 6:

- What is the rationale for using Harmony? If Harmony is not used, are all clusters segment-specific? And might this conclusion, in some sense, be more accurate (i.e. could it be biological reality that all spinal cord cell types are segment specific?) In Supp Fig 7a, some genes clearly distinguish brachial and lumbar cells that are considered the same “group.” This observation leads me to wonder how a threshold is chosen for considering a cluster segment-specific (and more broadly, if any threshold can be justified). Would it make more sense to skip Harmony and call all clusters segment-specific? Cross-segment comparisons would still be useful.

Response:

We thank the reviewer for raising this question. Based on prior knowledge of the identified spinal motor neuron (MN) subtypes, it has already been documented⁴⁻⁶ that brachial and lumbar segments share spinal MN subtypes. However, the field still lacks an unbiased measure to compare molecular subtypes in different segments. Integration methods such as Harmony⁷ have been widely used to characterize shared and specific cell subpopulations across different experiments and biological conditions due to their ability to simultaneously account for multiple experimental and biological factors. Thus, by using Harmony, we expected to identify both molecularly convergent and divergent subclusters between brachial and lumbar cells.

As a previous study has shown that *Etv4*⁺ motor pools are present in both brachial/lumbar segments⁸, we used this as a criterion to set a threshold in Harmony to reflect prior knowledge. Although methods such as Harmony enable identification of shared brachial and lumbar LMC subclusters, we agree with the reviewer that there might be a few genes that could still distinguish among these shared subclusters, perhaps reflecting the biological variation across segments. In our pioneering study, we attempted to compare cluster similarity at the transcriptome level, but we also recognize that functionally distinct clusters may exist in different segments defined by other extrinsic factors, such as innervation target preference or pre-motor circuit involvement. Further experimental validations, including marker staining with axonal tracing, to complement computational analyses are warranted in the future.

In the revised manuscript, to highlight how we defined our threshold setting based on previous *in vivo* knowledge and *in silico* relatedness, we now specify how we set the threshold in Harmony in the Methods section and have added the following statement in lines 297-301:

“LMC MNs are present in both the forelimbs and hindlimbs⁹, and some motor pools, such as *Etv4* subtype⁸, are known to be present in both brachial and lumbar segments, indicating that there might be more convergent or divergent subclusters between the two segments. Accordingly, we performed an unbiased cross-segmental molecular comparison of these LMC MN subclusters with Harmony⁷ to examine this hypothesis.”

Additionally, we now also point out the limitation of our approach in the Discussion section (lines 491-497):

“Consistent with this notion, we also report here that not all LMC subclusters are conserved between brachial and lumbar segments. In our pioneering study, while we attempted to compare cluster similarity at the transcriptome level, we also recognize that there might be functionally distinct clusters existing in different segments that are defined by other extrinsic factors, such as innervation target preference or pre-motor circuit involvement. Further experimental validations, including marker staining with axonal tracing, to complement computational analyses, are warranted.”

- Why not calculate mutual nearest neighbors on the PCA instead of the UMAP projection? The UMAP is a 2D representation where distances between cells are not necessarily interpretable.

Response:

We acknowledge the debate surrounding whether distances between cells in the UMAP space are biologically meaningful, but we also note that the new version of Monocle (i.e., Monocle 3) infers pseudotemporal trajectories based on the UMAP coordinates of cells. One previous benchmarking study has shown that UMAP can preserve both local and global structure from single-cell data and it highlighted the use of UMAP for improved visualization and interpretation of single-cell data¹⁰. Consequently, we calculated the mutual nearest neighbors in UMAP space.

Based on the reviewer’s suggestion, we have now calculated the mutual nearest neighbors in Harmony space (i.e., the corrected PCA space using the Harmony algorithm) (revised in line 335). We found that the calculated overlap ratios between brachial and lumbar LMC subclusters were highly consistent with those using the UMAP projection (Supplementary Fig. 11a). Quantitatively, the Pearson correlation between these two approaches was 0.99. These results support that the distances we calculated for our dataset in UMAP space are biologically interpretable.

- It would be nice to see what the rm/cm clusters look like on the UMAP in 6d/e.

Response:

In the revised manuscript, we have now added a UMAP distribution of rostral and caudal clusters as new **Fig. 7c**, described in **lines 336-342**:

“We observed two principles: (1) medial subclusters are similar between brachial and lumbar segments, as is the case for lateral subclusters in brachial/lumbar LMC MNs, suggesting that both segments share the molecular basis of mediolateral identity, whereas the similarity between rostrocaudal subclusters between limbs is less evident (**Fig. 7b-7e; Supplementary Fig. 11b**); and (2) though the overall molecular profile for mediolateral identity is similar for brachial/lumbar LMCs, a detailed comparison of each subcluster revealed a divergent tendency.”

4) My understanding is that cluster markers were determined mainly by fold change in expression. As a result, some of the markers are not very specific, and I wonder if making use of the pct1/pct2 value in the differential expression table could help find markers that are easier to use for validation experiments (i.e. “on” in one cluster and “off” in another). Two examples: - Fig 2E: I find it difficult to see that *Grm5* and *Nfib* co-localize only in the sacral stainings and not in the other sections. Is this clearer in magnification? Or would it be possible to use a more specific marker than *Grm5*?

Response:

We thank the reviewer for raising this issue. We have examined the difference between pct1 and pct2 for markers between our major clusters and uncovered the top five genes with the greatest differences for the caudal *Grm5*⁺*Nfib*⁺ cluster, which include our reported markers *Nfib* and *Grm5*. Furthermore, we found that none of these markers was exclusively specific to the annotated *Grm5*⁺*Nfib*⁺ cluster visualized in UMAP. Therefore, to better illustrate *Nfib* and *Grm5* co-expression in the caudal sample, we have now additionally performed RNAscope fluorescence in situ hybridization and corroborated exclusive co-expression of *Grm5*⁺*Nfib*⁺ in the sacral segment, but not in the brachial segment (new **Fig.2e**). Following the reviewer’s suggestion, we have further verified that *B3glt*, another top marker, also shared a similar expression profile to *Grm5*⁺*Nfib*⁺ in the sacral segment (new **Fig. 2e**, revised in **lines 166-168**). We now include the differences between pct1 and pct2 in Supplementary Table 1.

- Fig3B/C, Line 191: The phrase “*Lsamp*-expressing MMC neurons” is a little confusing, because *Lsamp* is expressed in both MMCs and LMCs (expression is higher in MMCs, but still >60% of LMCs express *Lsamp*).

Response:

We have now revised it to “*Lsamp*^{high} MMC neurons” to reflect its high expression in MMCs (**line 194**).

Minor:

Fig 1:

- A: The schematic might be clearer if MMC and LMC are labeled everywhere relevant, also

caudally.

- D. Can you comment on the cluster labeled “unknown?”
- E, F: Is it possible to elaborate on how these bar plots are made? Are the bins the same for every gene? Why not use a dot plot to be consistent with other figures?

Response:

A: We have added the requested annotation to our revised **Fig. 1a**.

D: As yet, we cannot annotate this cluster to any known subtype or as a novel MN subtype. Although this cluster exhibited differentially expressed markers, such as *Rxrg*, *Cntnap2* and *Onecut2* (new **Supplementary Fig. 3a**, pointed out in **line 150-151**), these markers are not specific in MNs, making *in vivo* validation incredibly challenging. Although this cluster of cells of “unknown” type shares some transcriptomic similarities to the MMC and Calb1⁺ clusters (such as *Mecom* expression), many genes are still differentially expressed between these clusters. For example, 509 genes were differentially expressed between the Calb1⁺ and the unknown cluster, including *Calb1*, *Fth1* and *Chchd2*, which is consistent with the *in vivo* observation that Calb1⁺ MNs are only observed in the brachial and upper thoracic segment. Thus, we have annotated the cluster as “unknown” at this stage. It is not an uncommon scenario in scRNAseq-based cell atlas annotation studies to have unassignable clusters, given that it might not be possible to define some types of cells simply at the transcriptome level.

E, F: We thank the reviewer for pointing out this ambiguity. For each gene in an individual cell cluster, we first ordered the cells based on the expression levels and then averaged the expression levels of cells located in each bin by evenly dividing the cells into 10 bins. Bar height corresponds to the average gene expression in each of the 10 bins in an individual cell cluster. We normalized the bar height across all cell clusters based on the maximum bar height. Thus, the number of bins is the same for each gene within the same cluster.

We agree with the reviewer that it would be more intuitive to use a dot-plot to represent the expression patterns of the selected markers, and we have now revised our figures accordingly (**new Fig. 1e and 1f**).

Fig 2:

- A, Line 158: What is the rationale for looking at Chat? Does it provide more information than Mnx1 alone?
- C: At P7, it’s hard to appreciate the overlap between Calb1 and Chat in the image magnifications without zooming a lot. Is it possible for an overlay also to be magnified?

Response:

A: Although Mnx1 is a marker specific to MNs in the spinal cord, it has two limitations. First, there is a population of spinal glutamatergic interneurons that co-express Mnx1 during development. As spinal motor neurons are all cholinergic, using Chat as another marker helped us exclude interneurons. Second, *Mnx1* was sparsely detected in our single-cell dataset, likely due to

a drop-out issue commonly reported for scRNAseq data. Thus, we defined MNs using a series of MN markers, i.e., cholinergic (*Slc18a3* and *Chat*) and spinal MN markers (*Mnx1* or *Isl1*).

In our revised manuscript, we have specifically addressed this query by revising our text to (lines 130-135):

“Although *Mnx1* labels MNs and a population of spinal glutamatergic interneurons (IN) specifically in the spinal cord¹¹, we detected sparse *Mnx1* expression in our single-cell dataset, likely due to scRNA-seq sensitivity limitation and consequently regarded as false-negative cells. As spinal MNs are all cholinergic, we thus defined MNs according to their expression of both cholinergic (*Slc18a3*, *Chat*) and spinal MN markers (*Mnx1*), with spinal MNs accounting for ~91% of the total cells (rostral: 5162; caudal: 4699 cells) upon quality filtering (Supplementary Fig. 2).”

C: We have now added a magnified overlay image to the revised manuscript (new Fig. 2c).

Fig 4:

- C: On the UMAP some clusters are labeled caudal despite expressing both *Hoxa5* and *Hoxc8* (contrary to the staining in A). Is this expected?
- D: It would be nice to see a dot plot for these genes instead. I don't think all are listed in 4C.

Response:

C: The reviewer has made a sharp observation, which has indeed been reflected in our previous observations from several studies, including an E12.5 brachial scRNAseq dataset^{12,13}. In our previous study, we observed that lineage segregation at the tissue boundary is manifested by proteins rather than mRNAs. One of our studies demonstrated that a miRNA-Hox circuit is important in establishing robust boundary formation at the post-transcriptional level¹³. We have incorporated this explanation in our Discussion section (lines 561-568).

D: We have now included a dot-plot of spatial genes in Fig. 4c.

Fig 5

- A: How is the Gini index calculated? Is it possible that the index is affected by either: (1) the number of genes belonging to each gene set, or (2) the average expression levels in a gene set? i.e. Are either of these values correlated with the index?

Response:

We calculated the Gini index according to its standard definition. The Gini index is a measure of inequality, defined as the mean of absolute differences between all pairs of individuals for some measure. The minimum value is 0 when all measurements are equal and the theoretical maximum is 1 for an infinitely large set of observations where all measurements but one has a value of 0. If the x values are first placed in ascending order, then the Gini index is defined as:

$$G = \frac{2}{n^2\bar{x}} \sum_{i=1}^n i(x_i - \bar{x}).$$

Computing the correlation between the number of genes within each gene set and the median Gini index of genes within each gene set, we did not observe a statistically significant correlation (corr = -0.12, p-value = 0.82). Computing the correlation between the average expression levels within each gene set and the median Gini index of genes within each gene set, again we did not observe a statistically significant correlation (corr = -0.09, p-value = 0.87). We have now added these observations to the Results section of the revised manuscript (lines 265-267):

“Using the Gini index to assess variable gene expression among LMC subclusters, we observed that neuropeptides are as variable as cell adhesion molecules and significantly more variable than housekeeping genes (Fig. 5a; the Gini index was not correlated with either the number of genes within a gene set (correlation = -0.12, p-value = 0.82) or with the average expression levels within a gene set (correlation = -0.09, p-value = 0.87)).”

- G: What are the abbreviations?

Response:

Apologies for the omission. The abbreviations stand for brachial nerves. We thank the reviewer for reminding us to include their full names in the figure legend.

Fig 7

- C, Supp Fig 8c: From Fig7b, I would have expected that Nr2f2 and Bcl11b are co-expressed less often than Satb2 and Bcl11b. Could this co-expression actually be a fourth cluster? Only a subset of the Bcl11b cluster seems to be Nr2f2+. Maybe these cells can be distinguished by another marker?

Response:

This is an interesting question. To investigate this issue further, we performed differential expression analysis on these five subclusters of MMCs. However, no specific marker genes were identified for the Bcl11b⁺Satb2⁺ nor Bcl11b⁺Nr2f2⁺ clusters, respectively (new Supplementary Fig. 12b, noted in text (lines 364-365)). Therefore, we do not consider these double-positive cells as independent subclusters.

Supplementary Figure 1D: Why is Mnx1 absent from so many cells if an Mnx1-GFP line was used? Is Mnx1 maintained only at the protein level?

Response:

We thank the reviewer for raising this query. We think that there might be two possibilities. First, it may be attributable to potential scRNAseq detection limitation (known as drop-out issue¹⁴). We

have examined GFP expression in our single-cell dataset, which was detectable in most cells, except for the non-neurons and interneurons (new Supplementary Fig. 1d, noted in text (lines 126-128)). However, in comparison to GFP, both *Mnx1* and *Chat* exhibited only sparse expression, supporting our notion of a drop-out issue. Second, it may be due to long GFP protein persistence. This scenario is reflected in several previous studies showing that while *Mnx1*-GFP^{high} cells are largely represented as “presumptive MNs”, GFP persistence does not fully recapitulate endogenous *Mnx1* expression in embryos¹⁵. In fact, some MN subtypes, such as LMCm, appear as *Mnx1*^{off} at E13.5. Accordingly, by using *Mnx1*::GFP embryos to sort out the GFP^{high} cells, we could identify all MNs that transiently express generic MN markers, such as *Mnx1*/*Isl1*/*Isl2*/*Lhx3*.

Reviewer #2 (Remarks to the Author):

Liau et al. characterize the diversity of motor neuron subtypes in the embryonic spinal cord. To do so, they isolate motor neurons from E13.5 Mnx1-GFP embryos and perform single cell RNA sequencing using the 10X approach. Using bioinformatic analysis the authors first reveal previously unrecognized motor neuron subtypes and provide experimental evidence for their existence in the embryo. They subsequently focus on lateral motor column (LMC) neurons. By using clustering approaches, the authors identify combinatorial codes of transcription factors (TFs) and neuropeptides that define distinct LMC subtypes and demonstrate that they can approximate where these subtypes are located based on the expression of marker genes known to partition LMC neurons into medial-lateral and rostral-caudal identities. Lastly, the authors demonstrate that medial motor column (MMC) neurons can be further partitioned based on the expression of the TFs Satb2, Bcl11b and Nr2f2, provide a first indication that this division is also functionally significant as these subtypes display distinct axonal projection patterns and demonstrate that this subdivision is evolutionary conserved between mouse, chicken, and human.

Overall, this is a well-executed study and a well written manuscript that I enjoyed reading. The single cell transcriptome data will undoubtedly help to further decipher how motor neuron diversity is established during development and will be an important resource for the scientific community in the future. Furthermore, the finding of a combinatorial code of neuropeptides in LMC neurons has the potential to be highly significant to understand how motor neuron diversity is established. The bioinformatic analysis is solid and observations made in the single cell transcriptome data are thoroughly validated experimentally. I have few comments that I am sure the authors will be able to address during revision.

We very much appreciate this reviewer's encouragement. We address all of his/her comments below in detail.

1) Subclustering of the LMC data (lines 241 – 244): The authors state that they can “uncover previously uncharacterized motor pools that innervate individual muscle groups”. This statement is based on the fact that some of the clusters they identified indeed express known markers for distinct motor pools. However, this does not mean that the other clusters also correspond to specific motor pools or even groups of motor pools. In fact, some of these clusters must comprise neurons from multiple motor pools given that, as the authors state themselves in the discussion, the number of LMC subtypes they uncover does not match ~60 expected motor pools.

Response:

We thank the reviewer for raising this topic. Indeed, expected motor pools outnumber the clusters identified in our dataset, and we do not rule out the possibility that the clusters may comprise multiple motor pools. We have now corrected our statement in the revised manuscript to read (lines 102-104):

“Within LMC MNs, we reveal transcriptional codes for 16 and 10 subclusters from brachial and lumbar limbs, respectively, which may correspond to putative motor pool groups. Moreover, we demonstrate unexpectedly diverse expression of neuropeptides in the LMC MNs.”

and (lines 250-254)

“Thus, profiling of our enriched MN population allowed us to recover LMC MN subtypes at an unprecedented resolution, uncovering previously uncharacterized subtypes that might innervate individual muscle groups and suggesting that the cluster-specific TFs they express might be important to establish their presumptive motor pool identity.”

Moreover, we elaborate further in our revised Discussion (lines 499-513):

“Although we have expanded the number of molecularly defined LMC subtypes, the number does not match the expected ~60 known LMC motor pools. There are a few possible reasons for this outcome. First, though motor pools are expected to be molecularly defined at the transcriptome level, the subtypes we identified could also correspond to one or multiple motor pools that manifest subtle transcriptomic differences. Second, motor pools at the embryonic stage might not be reflected by the transcriptome alone. An increasing number of studies have demonstrated that single-cell multi-omics analyses, such as joint single-cell transcriptomic and epigenomic profiling, can increase the ability to dissect cellular heterogeneity^{16,17}. Third, some neuronal identities may only be elicited by signaling or activity during spinal cord wiring at the postnatal stage¹⁸. Accordingly, a longitudinal examination of spinal MN identity integrated with later time-points may be enlightening¹⁸. Fourth, although we determined the number of subclusters using a robust ensemble approach based on mathematical spectral graph theory, single-cell transcriptomic data is inherently noisy and sparse in nature, which engenders additional challenges for estimating optimal numbers of clusters from such datasets. Further computational subclustering analysis with *in vivo* functional validation would likely enhance our understanding of LMC diversity.”

2) Related to the previous point, the authors show for many quantifications in the paper mean expression levels and % expressed cells as is widely-used standard in the field. From these quantifications it is clear that in most cases not all cells in a cluster express a specific marker. This may obviously be due to drop-out events in the single cell sequencing but could also reflect true biological heterogeneity. For example, the authors chose *Zfhx4* for their proof of principle validation experiment, which is specifically expressed in r11 and r12 subclusters, however, in both subclusters less than 50 % of cells have reads for this gene. Is this also true when looking at the protein level?

Response:

We thank the reviewer for noting that some marker genes were only sparsely detected in our dataset, prompting the query as to whether it represents biological heterogeneity. It is known in the previous studies that sometimes the marker genes are not uniformly expressed in identified subcluster partly due to drop-out. For example, in the fly visual system, despite *Dac* being a lamina neuron marker that is always detected by means of immunofluorescence, only 15-30% of those cells have *Dac* transcripts detected by scRNA-seq¹⁹, and the detection rate can vary widely for well-characterized markers for defined cell types (Fig. R3). Similarly, drop-out issue likely occurred in our study, as postulated for *Mnx1* in our study (see Supplementary Fig. 2d and 2f).

As the reviewer suggested, *Zfhx4* might provide a good paradigm to check this issue, given that it is expressed in about 50% of the cells in r11 or r12 subclusters from single-cell data. In our revised manuscript, we have now added quantification data of *Zfhx4*⁺ *Isl1*⁻ protein expression across the rostrocaudal axis of the spinal cord. We observed that ~15% of the *Zfhx4*⁺ *Isl1*⁻ LMC cell population occurs in the most rostral segment (new Fig. 4g and 4h), in support to selective expression of this TFs in r11/r12 from scRNA-seq analysis. We further quantified their proportions in *Foxp1*⁺ *Isl1*⁻ rostral lateral only cells from two sections for each embryo with n=3 and identified that *Zfhx4*⁺ *Isl1*⁻ cells accounted for ~12% in rostral lateral LMCs. By quantifying *Zfhx4*⁺ cells in rLMC1 from single-cell data, ~26% of rLMC1 cells expressed *Zfhx4* (normalized expression > 0.5). If drop-out has occurred, we would have expected far less percentage of *Zfhx4*-expressing cells in

single-cell data, compared to protein levels. One possibility of the lower percentage of protein levels was the inaccurate quantification of protein levels as there is no specific r11 and r12 subcluster marker that can be used as a denominator to quantify further.

Therefore, we think it is a challenge to clarify whether *Zfhx4* is undetected due to limitations in scRNA-seq, or if *Zfhx4* truly labels only partial of the subcluster with true biological heterogeneity that lies within the r11/r12 subclusters and corresponds to functional distinction.

In our manuscript, we hope to emphasize that in LMC MNs, we identified more transcriptional heterogeneity than previously reported motor pool TFs. We identified *Zfhx4* that labels a subset of LMC MNs from single-cell data that might represent the LMC subtype and further corroborate their expression *in vivo*. We also remind the readers about the caveat in our study in the Discussion as below (lines 499-513):

“Although we have expanded the number of molecularly defined LMC subtypes, the number does not match the expected ~60 known LMC motor pools. There are a few possible reasons for this outcome. First, though motor pools are expected to be molecularly defined at the transcriptome level, the subtypes we identified could also correspond to one or multiple motor pools that manifest subtle transcriptomic differences. Second, motor pools at the embryonic stage might not be reflected by the transcriptome alone. An increasing number of studies have demonstrated that single-cell multi-omics analyses, such as joint single-cell transcriptomic and epigenomic profiling, can increase the ability to dissect cellular heterogeneity^{16,17}. Third, some neuronal identities may only be elicited by signaling or activity during spinal cord wiring at the postnatal stage¹⁸. Accordingly, a longitudinal examination of spinal MN identity integrated with later time-points may be enlightening¹⁸. Fourth, although we determined the number of subclusters using a robust ensemble approach based on mathematical spectral graph theory, single-cell transcriptomic data is inherently noisy and sparse in nature, which engenders additional challenges for estimating optimal numbers of clusters from such datasets. Further computational subclustering analysis with *in vivo* functional validation would likely enhance our understanding of LMC diversity.”

3) Line 263 – 266: I am not sure which message the authors want to convey here and why it is “remarkable that the *Etv4*⁺ subcluster expressed *Sst*, whereas the *Pou3f1*⁺ subcluster displayed *Penk* expression.” Do the authors want to indicate that the subcluster-specific TFs might regulate the expression of the neuropeptides? Related to this point, the authors describe combinatorial codes of TFs and neuropeptides in LMC neurons. How well do these 2 codes match and is there a close correlation between the expression of specific TFs and neuropeptides, which may be indicative of specific TFs regulating the expression of specific neuropeptides?

Response:

In order to examine if specific TFs directly regulate specific neuropeptides, we generated an inducible Tet-ON *Etv4* ESC line as an example to test our hypothesis, selecting the *Etv4*⁺ motor pool as it is one of the best characterized MN subtypes^{6,8}. We differentiated this line into spinal MNs and added doxycycline to induce *Etv4* overexpression (Fig. R4). However, we observed no consistent induction for our selected neuropeptides, which showed positive correlations with *Etv4*,

including *Sst*, *Grp*, *Npy*, *Gal*, *Calcb*, and *Vgf*, despite slight inductions with *Cbln4*. This outcome

indicates that *Etv4* itself might not be able to induce directly presumptive specific neuropeptides in this motor pool. However, our experimental design only encompasses a single TF induction, so we cannot rule out that other segmental TFs (e.g., *Hox* genes such as *Hox5/6/8*) are needed to co-regulate neuropeptides, with combinatorial TF codes perhaps being required to delineate subclusters. In addition, we did not investigate context-dependent regulation as our *in vitro* model

system only mimics a limited region of the spinal cord axis. Multi-factor co-regulation would require further experimental validations, such as using ChIP-seq.

Nevertheless, we have investigated the correlations between subcluster-specific TFs and neuropeptides. We computed Pearson's correlation between these genes based on their gene expression levels. We then compared the calculated correlations against the correlations between any pair of TFs and neuropeptides, which did not display a cluster-enriched pattern. We observed a significantly stronger correlation between the subcluster-specific TFs and neuropeptides compared to the correlation between randomly selected TFs and neuropeptides (new Fig. 5g). This stronger correlation between the expression of subcluster-specific TFs and neuropeptides from our analysis implies that these genes might exert subtype-specific physiological functions, warranting further experimental validation.

In our revised manuscript, we have now modified our sentence about TF and neuropeptide co-regulation to (lines 277-285):

“Furthermore, we observed a significantly stronger correlation between the subcluster-specific TFs and neuropeptides compared to the correlation between randomly selected TFs and neuropeptides (Fig. 5g), implying that motor pool TFs in combination with neuropeptides may exert subtype-specific physiological roles that warrant future experimental validation. Strikingly, upon comparing our embryonic MN dataset to previously published adult MN scRNAseq results^{20,21}, we found that though the embryonic LMC subtype TF/neuropeptide code is largely distinct from that of adult MN subtypes, adult skeletal MN subtypes still manifest a different set of neuropeptide combinations, implying that the neuropeptides might function differentially in both embryonic and adult MNs (Supplementary Fig. 7, 8).”

4) Because the body axis is sequentially generated from anterior to posterior, the lumbar part of the spinal cord is typically developmentally delayed compared to the brachial region. Consistently, GO term analysis revealed that lumbar LMC neurons were enriched for terms related to MN cell fate specification and development, which may indicate an earlier maturation stage. This raises the question to which extent the results of the differential gene expression analysis between brachial and lumbar LMC neurons may be biased by neurons in these regions being potentially at different stages of their maturation process instead of revealing true differences due to axial identity? One possibility to test this hypothesis could be to compare the E13.5 lumbar LMC sequencing data to single cell transcriptome data from earlier stages, e.g. from E12.5 motor neurons from Li et al. Mol Syst Biol. (2021) 17:e9945.

Response:

In light of the reviewer's suggestion, we compared E12.5 brachial and E13.5 lumbar motor neuron datasets and found that the neuropeptide signaling pathway (GO:0007218) was still more enriched in the E12.5 brachial compared to the E13.5 lumbar datasets (new Supplementary Fig. 9c and 9d).

To strengthen this conclusion, we first performed additional experiments to confirm some identified brachial-enriched (e.g., *Grp* and *Npy*) and lumbar-enriched (i.e., *Igf1*) genes from single-cell analysis along the E13.5 brachial and lumbar spinal cord sections using fluorescence in situ hybridization (RNAscope), which allowed us to conduct quantification analysis (new Fig. 6c-g and Supplementary Fig. 9a, b). Consistent with our scRNAseq analyses, *Npy* and *Grp* were enriched in the brachial relative to lumbar spinal cord, whereas we detected a higher ratio of *Igf1*-expressing cells in lumbar segments at E13.5. Furthermore, at E14.5, brachial-enriched *Npy* and *Grp* exhibited consistently higher expression in the brachial relative to lumbar segment (new Supplementary Fig. 9e and 9f). Together, these new results corroborate that the differential expression of neuropeptide genes between brachial and lumbar segments is unlikely to be biased by the neurons being at different developmental stages.

We have added these new validation experiments to our revised manuscript and changed the respective text accordingly in the Results section (lines 317 – 323):

“The differential neuropeptide expression in either brachial or lumbar LMCs was not a reflection of a developmental timing delay, as when we compared E12.5 brachial and E13.5 lumbar LMC MN datasets, we found that the neuropeptide signaling pathway (GO:0007218) is still more enriched in the E12.5 brachial¹³ compared to E13.5 lumbar segment, representing a similar outcome to our findings for segmental gene expression at the same stage (E13.5) (Supplementary Fig. 9c and 9d). Furthermore, expressions of brachial-enriched *Npy* and *Grp* were consistent from E13.5 to E14.5 (Supplementary Fig. 9e and 9f).”

Reviewer #3 (Remarks to the Author):

This detailed single cell transcriptomic description of embryonic motor neurons by Liao et al. is compelling evidence for the molecular diversity of motor neuron subtypes whose existence has long been hypothesized but for which specific markers were for the most part unknown. In this study, the authors perform single cell sequencing of Mnx1-GFP positive motor neurons (MNs) of 2 regions of the developing spinal cord at embryonic day E13.5 (brachial and lumbar). They discover new MN subtypes and define new markers for known and novel MN subtypes. They reveal combinations of transcription factors that define motor neuron subclusters likely corresponding to motor pools. The authors show these clusters are also linked to a neuropeptide code which they hypothesize might play a functional role in the formation of specific connections between motor pools and their muscle targets. Finally, they convincingly demonstrate the existence of 3 subtypes of MMC neurons in the mouse and that these exist in other species (chick and human).

This work merits publication as an important resource for the community of researchers interested in motor neuron biology. However, several important issues should first be addressed.

We very much appreciate this reviewer's encouragement. We address his/her comments below in detail.

The single cell RNA sequencing (scRNAseq) data presented here is of high quality, and the interpretation is solid and reasonable. However, the validation of the scRNASeq data in tissue is often quite limited, to the point where some claims are validated by a single example image. The authors should more rigorously validate their scRNAseq data with quantifications of their in situ hybridization and immunostaining data to help support their claims. Furthermore, demonstrating co-expression of combinatorial markers that define novel cell types rather than showing the presence of markers in a similar region in different sections would be a more appropriate and powerful validation of the scRNAseq data.

Response:

We appreciate the reviewer's suggestions. For the revised manuscript, we have now performed several new RNAscope and immunostaining experiments, enabling us to simultaneously assess different combinations of newly uncovered markers with known genes on the same sections, together with detailed quantifications:

- Neuropeptide differential expression between brachial and lumbar limbs (new Fig. 6c-g, and Supplementary Fig. 9a, b)
- Lumbar-specific LMC subcluster (new Fig. 7h-7i)
- Quantification of cluster-specific TFs (new Fig. 4g, 4h)
- Multiplexed fluorescence ISH of TFs and neuropeptides (new Fig. 5e)
- Differential receptor expression in MMC subtypes (new Fig. 8g, h, i)

Another issue is that the authors may be inflating the true number of MN subclusters by over-

subclustering their data, in particular in the reported diversity of LMC subtypes.

“Within limb MNs, we reveal transcriptional codes for 26 subclusters corresponding to putative motor pools and demonstrate unexpectedly diverse expression of neuropeptides in the LMC MNs.”

Unless I missed it, there was no clear figure representation of the 26 clusters. If the 26 arise from adding up the brachial and lumbar clusters, then this number is not convincing unless the authors integrate brachial and lumbar and show that clusters shared between the 2 regions are distinct from one another. It is not clear this will be true since some of these appear to share identity, for example, in Figure 6a, brachial cm2 = lumbar cm1.

Response:

We appreciate this observation. There are indeed molecularly similar populations present in different segments. We annotated the brachial and lumbar clusters separately, considering they innervate the forelimbs and hindlimbs respectively, which is consistent with how motor pools are functionally defined. However, this functional distinction could result from the physical separation of their targets and may not be encoded molecularly. To take both molecular and functional distinctions into consideration when annotating LMC MN subclusters, we have now corrected the statement in the revised manuscript to:

(lines 102-104)

“Within LMC MNs, we reveal transcriptional codes for 16 and 10 subclusters from brachial and lumbar limbs, respectively, which may correspond to putative motor pool groups. Moreover, we demonstrate unexpectedly diverse expression of neuropeptides in the LMC MNs.”

and

(lines 425-428)

“Furthermore, we leveraged the large number of MNs in our dataset to permit a more detailed and unbiased analysis of heterogeneity within LMC and MMC MNs, resulting in 16 and 10 LMC subclusters from brachial and lumbar segments, respectively, and 3 MMC subclusters.”

Also, we expand on this topic a bit more in the Discussion:

(lines 491-497)

“Consistent with this notion, we also report here that not all LMC subclusters are conserved between brachial and lumbar segments. In our pioneering study, while we attempted to compare cluster similarity at the transcriptome level, we also recognize that there might be functionally distinct clusters existing in different segments that are defined by other extrinsic factors, such as innervation target preference or pre-motor circuit involvement. Further experimental validations, including marker staining with axonal tracing, to complement computational analyses, are warranted.”

and (lines 499-513)

“Although we have expanded the number of molecularly defined LMC subtypes, the number does not match the expected ~60 known LMC motor pools. There are a few possible reasons for this outcome. First, though motor pools are expected to be molecularly defined at the transcriptome level, the subtypes we identified could also correspond to one or multiple motor pools that manifest subtle transcriptomic differences. Second, motor pools at the embryonic stage might not be reflected by the transcriptome alone. An increasing number of studies have demonstrated that single-cell multi-omics analyses, such as joint single-cell transcriptomic and epigenomic profiling, can increase the ability to dissect cellular heterogeneity^{16,17}. Third, some neuronal identities may only be elicited by signaling or activity during spinal cord wiring at the postnatal stage¹⁸. Accordingly, a longitudinal examination of spinal MN identity integrated with later time-points may be enlightening¹⁸. Fourth, although we determined the number of subclusters using a robust ensemble approach based on mathematical spectral graph theory, single-cell transcriptomic data is inherently noisy and sparse in nature, which engenders additional challenges for estimating optimal numbers of clusters from such datasets. Further computational subclustering analysis with *in vivo* functional validation would likely enhance our understanding of LMC diversity.”

Finally, relating to the discovery of 3 distinct subtypes of axial MNs:

The data presented in Figure 7 was some of the strongest data in the manuscript, though I was not entirely convinced by the validation for the “different axon guidance signatures” (Fig 7f-h). If there is a better way to show one branch projecting to its target, or additional examples, the support for this claim would be substantially stronger.

Response:

We appreciate this suggestion to strengthen our conclusion through additional experiments. We acknowledge that retrograde labeling of different axial muscles would represent the gold standard in showing potential muscle target differences of MMC subtypes. However, it is extremely challenging to perform axial muscle subtype retrograde labeling at this embryonic stage. We have attempted to do so, but with very little success (from ~8 batches of experiments on E14.5 embryos, we obtained <3 embryos displaying successful and specific labeling). In our revised Discussion, we now caution readers that further experiments are needed to explore the link between axial MN subtypes and muscle subtype connections (lines 523-527):

“Future studies to confirm functional differences between these subtypes via retrograde labeling in embryonic muscles will be necessary, albeit technically challenging. In addition, the identified MMC subtype markers of *Nr2f2*, *Satb2* and *Bcl11b* facilitate region patterning in the brain^{22,23}. Whether they exert similar regulatory roles in MMC MNs warrants further investigation.”

Additionally, we also now provide a higher resolution examination and quantification of *Nrp2* transcripts in *Satb2*⁺ and *Nr2f2*⁺ cells using RNAscope multiplexed fluorescence *in situ* hybridization (new Fig. 8g, 8h, revised text in line 375-377) on MMC MNs, as well as higher magnification images of axonal staining with *Unc5c/Nrp2/Mnx1*-RFP (new Fig. 8i). We believe that these new data have strengthened the evidence that axial MN subtypes express different sets of axon guidance molecules.

Below is a detailed list of comments to address, in the order in which the items appear in the manuscript:

The authors should call attention to the fact that their dataset is missing the thoracic T4-T13 region; this is important background info to appreciate. It is indirectly shown in Figure 1b and reported in the methods, but some mention in the main text would be helpful.

Response:

Agreed, we have now added the following statement in our Results section (lines 205-214):

“The identification of major columnar MNs within our MN dataset prompted us to uncover subtypes within these motor columns. Characterization of MN subtypes in the adult stage has revealed previously unreported heterogeneity in PGC MNs^{20,21} and the embryonic origin of these subtypes remains unclear. Although we sampled spinal cord tissue mainly at the limb segments, we had included small populations of PGC and HMC MNs within our collected samples. We observed that PGC MNs also exhibit subtypes (Fig. 1c; Supplementary Fig. 4c), concordant with observations from a previous study²⁴, however we did not endeavor to characterize these motor columns further. In this study, we hope to focus on analyzing LMC and MMC MN subtypes. We used LMCs as a proof-of-principle type to test if transcriptome-based subclusters match certain known motor pools, as marker genes for some LMC motor pools have been revealed²⁵.”

Since the authors used a *Mnx1*-GFP line to isolate the neurons, can they comment on the relatively low expression levels of cholinergic and spinal MN markers in their dataset (Suppl Fig 2d) – eg *Chat* and *Mnx1* itself.

Response:

We thank the reviewer for raising this query. We think that there might be two possibilities. First, it may be attributable to potential scRNAseq detection limitation (known as drop-out issue¹⁴). We have examined GFP expression in our single-cell dataset, which was detectable in most cells, except for the non-neurons and interneurons (new Supplementary Fig. 1d). However, in comparison to GFP, both *Mnx1* and *Chat* exhibited only sparse expression, supporting our notion of a drop-out issue. Second, it may be due to long GFP protein persistence. This scenario is reflected in several previous studies showing that while *Mnx1*-GFP^{high} cells are largely represented as “presumptive MNs”, GFP persistence does not fully recapitulate endogenous *Mnx1* expression in embryos¹⁵. In fact, some MN subtypes, such as LMCm, appear as *Mnx1*^{off} at E13.5. Accordingly, by using *Mnx1*::GFP embryos to sort out the GFP^{high} cells, we could identify all MNs that transiently express generic MN markers, such as *Mnx1*/*Isl1*/*Isl2*/*Lhx3*.

In our revised manuscript, we have specifically addressed this query by revising our text to (lines 130-135):

Although *Mnx1* labels MNs and a population of spinal glutamatergic interneurons (IN) specifically in the spinal cord¹¹, we detected sparse *Mnx1* expression in our single-cell dataset, likely due to

scRNA-seq sensitivity limitation and consequently regarded as false-negative cells. As spinal MNs are all cholinergic, we thus defined MNs according to their expression of both cholinergic (*Slc18a3*, *Chat*) and spinal MN markers (*Mnx1*), with spinal MNs accounting for ~91% of the total cells (rostral: 5162; caudal: 4699 cells) upon quality filtering (**Supplementary Fig. 2**).

Figure 1c-d: Why didn't the authors first integrate the data to highlight the differences between the rostral and caudal spinal cord levels? Starting with a view of the data integrated as a whole, and later separated out, would be a more intuitive way to present the data. If there is a reason why the authors did not do it, they should provide an explanation, but it would be preferable to provide this integrated view.

Response:

We thank the reviewer for these insightful comments. Although we agree with the reviewer that starting with an integration analysis would represent an intuitive way to present our data, as explained below, reliable integration between biological conditions remains challenging. Previously, we indeed have performed the suggested analysis of combined brachial and lumbar samples using the widely deployed Seurat V3 and Harmony R packages (**new Supplementary Fig. 13**). Similar to other studies reporting erosion of transcriptomic differences as an issue^{26,27}, including our previous study²⁸ from Dr. Qing Nie's lab (one of the corresponding authors in this manuscript), we noted that Seurat V3 integration intermingled some clusters with other clusters in the integrated assay. For example, a small subset of *Tac1*-expressing cells was not differentiated as a single cluster (**Arrowhead, Supplementary Fig. 13**), which we subsequently identified as HMC MNs based on known marker expression (*Isl1*⁺*Lhx3*-*Foxp1*) and validated through *Tac1* *in situ* hybridization. A group of cells expressing the LMC markers *Foxp1* and *Aldh1a2* were separated and assigned to another cluster (cluster 1, **Arrow in Supplementary Fig. 13**). These unexpected results are likely integration artifacts, as *Foxp1* and *Aldh1a2* co-expression is known to specifically label LMCs *in vivo*. Other previous benchmarking studies have also shown that most methods can eliminate *bona fide* transcriptomic differences as a side-effect when removing batch effects²⁹, and we observed similar complications in our integrative analysis.

Although major neuronal types can be identified using various integrated methods, integration while correcting for batch effects and preserving segmental differences is challenging in our case as transcriptomic differences are expected to be less prominent than the major neuronal types observed in other studies. As integration tools have been shown to erase biologically-specific differences to various extents²⁹, we reasoned that analyzing the brachial vs lumbar datasets separately would preserve more molecular distinctiveness, with *in vivo* validations accounting for batch-associated artifacts.

To explain this issue, we have added sentences to explain this in the Discussion (**lines 571-585**):

“Furthermore, while we have also attempted to cluster the collected MNs after integrating rostral and caudal datasets (**Supplementary Fig. 13**), we noticed several integration complications that disagreed with *in vivo* observations reported by previous and our studies. For example, a small subset of *Tac1*-expressing cells was not differentiated as a single cluster (**Supplementary Fig. 13**). Some cells expressing the LMC markers (*Foxp1* and *Aldh1a2*) were incorrectly assigned to another cluster which does not express these marker genes (**Supplementary Fig. 13**). Yet in our separate

analysis, we identified the *Tac1*-expressing cluster of cells as the HMC MNs based on known marker expression (*Isl1⁺Lhx3⁻Foxp1⁻*) and validation through *Tac1 in situ* hybridization. Given that many previous elegant studies have provided rich knowledge on Hox-mediated segmental MN types²⁵, which could be served as a reference for us to annotate known segmentally-restricted types with confidence, thus, in our study, we presented separate analyses for rostral and caudal samples to prevent overlooking subtle sample type-intrinsic differences. This also highlights that reliable integration between biological conditions remains challenging²⁹, and future development with more advanced algorithms to overcome this issue is anticipated.”

“First, we focused on previously unreported and specifically expressed TFs (i.e., *Zfhx3*, *Zfhx4*, *Foxp4* and *Casz1*), since motor pool-specific TFs are usually the major determinants for establishing motor pool identity and properties. As a proof of principle validation experiment, we chose *Zfhx4*” – Can the authors add a quantification for *Zfhx4* (one example image is not sufficient validation). It would be preferable to see at least one additional TF validated.

Figure 4h Line 238-239 – “We corroborated its sole expression in a subpopulation of LMC MNs in the rostral lateral region of the brachial segment (Fig. 4h)”. The authors should quantify the prevalence of their markers in the different quadrants, or at least add a quantification illustrated by this example (4h).

Response:

In our revised manuscript, we have now added quantification data for the distribution of *Zfhx4⁺Isl1⁻* cells across the rostrocaudal axis of the spinal cord. We observed that ~15% of the *Zfhx4⁺Isl1⁻* LMC cell population occurs in the most rostral segment (new Fig. 4g and 4h), in support to selective expression of this TFs in r11/r12 from single-cell result.

Additionally, as an additional proof-of-principle experiment to show LMC-TFs, we chose the TF *Nr2f1* and verified its expression in subsets of LMC MNs by immunostaining (new Supplementary Fig. 6b), with the results supporting the notion that TFs are usually differentially expressed in motor pool subtypes.

We have revised the respective statements in our manuscript as follows (lines 245-250):

“As a proof-of-principle validation experiment, we chose *Zfhx4* that is selectively expressed in the r11 and r12 subclusters (Fig. 4f). We corroborated its expression in ~15% of LMC MNs in the rostral lateral region of the brachial segment (Fig. 4g and 4h), resembling the protein expression pattern of *Zfh2* (a *Drosophila* ortholog of *Zfhx4*) in the late ventral nerve cord neurons of the fly³⁰. We also further observed *Nr2f1* expression in subsets of LMC MNs (Supplementary Fig. 6b).”

Combinatorial expression of neuropeptides and correlation of TFs in the given LMC subcluster is very interesting.

Please clarify: lines 263-266 “Remarkably, we also detected that the *Etv4⁺* 263 subcluster (cm1) expressed *Sst*, whereas the *Pou3f1⁺* 264 subcluster (cm3) displayed *Penk* expression (Fig.

5e, 5f), implying that these motor pool TFs might co-regulate with neuropeptides to provide subtype-specific physiological roles.”

Are the authors suggesting the TF combinations determine neuropeptide expression? Or just that motor TF's in combination with neuropeptides provide subtype specific physiological roles?

Response:

In order to examine if specific TFs directly regulate specific neuropeptides, we generated an inducible Tet-ON Etv4 ESC line as an example to test our hypothesis, selecting the Etv4⁺ motor pool as it is one of the best characterized MN subtypes^{6,8}. We differentiated this line into spinal MNs and added doxycycline to induce Etv4 overexpression (Fig. R4). However, we observed no consistent induction for our selected neuropeptides, which showed positive correlations with *Etv4*, including *Sst*, *Grp*, *Npy*, *Gal*, *Calcb*, and *Vgf*, despite slight inductions with *Cbln4*. This outcome indicates that Etv4 itself might not be able to induce directly presumptive specific neuropeptides in this motor pool. However, our experimental design only encompasses a single TF induction, so we cannot rule out that other segmental TFs (e.g., *Hox* genes such as *Hox5/6/8*) are needed to co-regulate neuropeptides, with combinatorial TF codes perhaps being required to delineate subclusters. In addition, we did not investigate context-dependent regulation as our *in vitro* model system only mimics a limited region of the spinal cord axis. Multi-factor co-regulation would require further experimental validations, such as using ChIP-seq.

Nevertheless, we have investigated the correlations between subcluster-specific TFs and neuropeptides. We computed Pearson's correlation between these genes based on their gene expression levels. We then compared the calculated correlations against the correlations between any pair of TFs and neuropeptides, which did not display a cluster-enriched pattern. We observed a significantly stronger correlation between the subcluster-specific TFs and neuropeptides compared to the correlation between randomly selected TFs and neuropeptides (new Fig. 5g). This stronger correlation between the expression of specific TFs and neuropeptides from our analysis

implies that these genes might exert subtype-specific physiological functions, warranting further

experimental validation.

In our revised manuscript, we have now modified our sentence about TF and neuropeptide co-regulation to (lines 277-285):

“Furthermore, we observed a significantly stronger correlation between the subcluster-specific TFs and neuropeptides compared to the correlation between randomly selected TFs and

neuropeptides (Fig. 5g), implying that motor pool TFs in combination with neuropeptides may exert subtype-specific physiological roles that warrant future experimental validation. Strikingly, upon comparing our embryonic MN dataset to previously published adult MN scRNAseq results^{20,21}, we found that though the embryonic motor pool TF/neuropeptide code is largely distinct from that of adult MN subtypes, adult skeletal MN subtypes still manifest a different set of neuropeptide combinations, implying that the neuropeptides might function differentially in both embryonic and adult MNs (Supplementary Fig. 7, 8).”

It is unclear why the authors specifically call out neuropeptide expression as unique. From their figure 5a, it's TFs, ligands/receptors and neuropeptides all stand out as important markers. Every category of cell type that is distinct will have some TFs/ligands/neuropeptides that are unique to it.

Response:

We apologize that we did not explain this rationale clearly in the original manuscript. A battery of MN subtype-specific TFs/ligands/receptors/cell adhesion molecules have been very well documented in previous studies^{3,6,8,25,31-37}. Though neuropeptide diversity has been reported for some other neuronal types in the neocortex or for sensory neurons^{38,39}, their expression and functional implications in MNs remain obscure, which prompted this study. Given that we found that neuropeptides are diversely expressed in a cluster-specific manner among LMC subclusters, as reported previously for TFs, ligands and receptors, we speculated that the respective genes could work in a combinatorial fashion to exert at least some of the physiological functions of MN subtypes.

We pointed out the importance of TFs/ligands/receptors and neuropeptides in our revised manuscript lines 232-237:

“To understand which gene types are diversely expressed in the LMC MNs, we performed ontology (GO) analysis on the top-ranked variable genes among these neurons. We observed that many of these genes encode TFs, cell adhesion and axon guidance molecules, receptors, and ion channels, many of which have been cited as having functions related to cell fate specification, motor pool sorting and positioning, axon pathfinding and synaptogenesis^{34,40} (Fig. 4e; Supplementary Fig. 5f).”

and lines 257-262

Upon checking the molecular pathways enriched for variable genes in LMC MNs, we noticed a GO term for neuropeptides, which are small proteins produced by neurons that usually act on G protein-coupled receptors. Neuropeptides can function via autocrine or paracrine signaling with neurotransmitters in a single neuron type to modulate and expand neuronal function⁴¹. Although neuropeptides have been used to categorize cell types in the brain and dorsal horn of the spinal cord^{42,43}, their expression and functions in MNs have not been characterized.”

The authors should consider exploring if any of this distinction is retained in the adult by comparing to data from Blum et al., 2021 and Alkaslasi et al., 2021, and whether (or how much

of) this code is erased after developmental connections are complete. A parallel question is whether there is any correspondence of the LMC or MMC clusters detected here with the adult skeletal MN clusters described in Blum et al or Alkaslasi et al.

Response:

We thank the reviewer for these insightful comments. To investigate any correspondence between the LMC and MMC subclusters and adult skeletal MN clusters, we first performed similarity analysis using the CIDER tool, which measures the Pearson's correlation coefficient of differentially expressed signatures between two cell groups. Interestingly, we found low similarity between the embryonic and adult cluster transcriptomes (new Supplementary Fig. 7). Subsequently, we examined the gene expression patterns of our subcluster-specific TFs and neuropeptides in the adult MN clusters described in Blum et al.²⁰ (new Supplementary Fig. 8). We found that most of these subcluster-specific TFs did not exhibit distinct expression levels across the adult skeletal MN subclusters, but a few neuropeptides showed selective enrichment. We applied the same approach to the Alkaslasi et al.²¹ dataset and revealed a similar outcome. **Thus, while the embryonic motor pool TF/neuropeptide code is largely distinct from that of adult MN subtypes, adult skeletal MN subtypes still manifest a different set of neuropeptide combinations, implying the neuropeptides might function differentially in both embryonic and adult stages** (new Supplementary Fig. 7, 8).

We describe this interesting finding in the revised manuscript as follows (lines 277-285):

“Furthermore, we observed a significantly stronger correlation between the subcluster-specific TFs and neuropeptides compared to the correlation between randomly selected TFs and neuropeptides (Fig. 5g), implying that motor pool TFs in combination with neuropeptides may exert subtype-specific physiological roles that warrant future experimental validation. Strikingly, upon comparing our embryonic MN dataset to previously published adult MN scRNAseq results^{20,21}, we found that though the embryonic LMC subtype TF/neuropeptide code is largely distinct from that of adult MN subtypes, adult skeletal MN subtypes still manifest a different set of neuropeptide combinations, implying that the neuropeptides might function differentially in both embryonic and adult stages (Supplementary Fig. 7, 8).”

Figure 5c: the dotted region showing distinction between m and l is in the wrong position. And a related note: in the methods, the authors should provide an explanation for how the dotted lines separating lateral vs medial LMC are defined.

Response:

We apologize for this mistake. We have now added details of the methodology to the respective legend and corrected the position of m/l in the figure.

Figure 5e: Pou3f1 is red in the label but magenta in the image.

Response:

Apologies. We have now replaced the image with a multiplexed fluorescence in situ hybridization image to illustrate co-expression between TFs and neuropeptides (new Fig. 5e).

Figure 5f: the idea for a combinatorial code of markers defining each motor pool is compelling. To make this data fully convincing, the authors should perform multiplexed in situ hybridization for some of these combinations of transcription factors + neuropeptides to demonstrate their existence. For example in Fig 5e, it would be better to see *Penk* and *Pou3f1* expressed in the same cells, not separate staining in neighboring sections.

Response:

We have now performed multiplexed *in situ* hybridization (RNAscope) and, indeed, we observed that *Penk* is expressed in *Pou3f1*⁺ cells, whereas *Sst* and *Grp* are expressed in *Etv4*⁺ cells (new Fig. 5e).

Figure 5g - please define the arrows.

Response:

Apologies. We have now added the full names of the nerves indicated by the arrows in the figure legend.

Fig 5g is very nice evidence of the specificity of *Npy* to just a subset of brachial nerves. What do *Npy*-Cre projections look like in the lumbar region? (re. Fig 6c).

Response:

To address this question, we have further explored *Npy*-cre projections in the lumbar region (new Fig. 5h, right). Similar to the brachial segment, *Npy*⁺ axons were detected in only a subset of lumbar muscle-targeting nerves (i.e., *Ig* (inferior gluteal), *T* (tibial), and *Dp* (deep peroneal)).

Figure 6 nomenclature is confusing: there is no definition of “crl” shown in 6f and 6g. Also in the text: line 217 - add definition of "crl" as show in Fig 6f and 6g

Apologies. We have now added the respective description to the figure legends. “crl” represents all lateral LMC MNs, as we could not distinguish the rostrocaudal identities for these lateral LMC subclusters based on *Hoxd10*⁺ and *Hoxd10*⁻ expression.

Lines 285-287 - “In contrast, neuropeptide-related terms were more enriched in the brachial LMC MNs (Fig. 6b), and we confirmed higher expression of *Grp* and *Npy* neuropeptides in the brachial LMC MNs by means of in situ hybridization (Fig. 6c).”

To support this claim, the authors should add a quantification for data represented in Fig. 6c.

Response:

To strengthen this conclusion, we have now performed additional experiments to confirm some of the brachial-enriched (e.g., *Grp* and *Npy*) and lumbar-enriched (i.e., *Igf*) genes identified from our single-cell analysis by means of a fluorescence *in situ* hybridization (RNAscope) approach that allowed us to conduct quantification analysis and to examine along the brachial and lumbar spinal cord sections (new Fig. 6c-g and Supplementary Fig. 9a, 9b). Consistent with our scRNAseq analyses, *Grp* was expressed in LMC subsets within the brachial spinal cord, but was absent from lumbar segments. Likewise, *Npy* was significantly more enriched in the most rostral brachial spinal cord sections. In contrast, we detected a higher ratio of *Igf1*-expressing cells in lumbar segments (new Fig. 6f, 6g). Together, these new results corroborate the differentially-expressing neuropeptide gene sets in the brachial and lumbar segments.

In our revised manuscript, we have revised the section to (lines 308-316):

“To validate that the observed differences between limbs are bona fide biological variations rather than batch sampling artifacts, we performed RNAscope multiplexed fluorescence *in situ* hybridization and quantified cell numbers to confirm higher expression of *Grp* and *Npy* neuropeptides in the brachial LMC MNs (Fig. 6c and 6d; Supplementary Fig. 9a). Additionally, we identified important neuropeptide genes such as *Igf1*², known to exert a neuroprotective function, as being differentially enriched in *Hoxd10*⁺ medial LMC MNs in the lumbar segment (Fig. 6c and 6f). Similarly, we observed that these neuropeptide-expressing populations are dynamically distributed across the rostrocaudal axis of the brachial or lumbar spinal cord (Fig. 6d-6g; Supplementary Fig. 9a, 9b), corroborating their aforementioned cluster-specific expression (Fig. 5b).”

Lines 309-312: “but 5 and 2 LMC subclusters were specific to the brachial or lumbar segment, respectively. For example, the *Grp*⁺ cl3 and r15 subclusters are present in the brachial segment alone, whereas the *Nkx6.1*⁺ 311 crl6 subcluster predominantly occurs in the lumbar segment (Fig. 6g, 6i; Supplementary Fig. 7b).” The authors should show this specificity in tissue.

Response:

Given that the crl6 cluster is represented by *Nkx6.1*⁺, *Isl1*⁻ and *Foxp1*⁺ expression, we co-immunostained for them and quantified their presence along the rostrocaudal axis within the brachial and lumbar segments. For both segments, *Nkx6.1*⁺ cells are located in the LMCm (*Isl1*⁺) cluster. Notably, we detected a lumbar-specific population of *Nkx6.1*⁺ *Isl1*⁻ cells (marked by an arrow) solely in the caudal lumbar spinal cord, likely reflecting the lumbar-specific crl6 cluster (new Fig. 7h). We have corroborated our findings by quantifying *Nkx6.1* (annotated by *) with and without *Isl1* expression in the LMC MNs (*Foxp1*⁺) (new Fig. 7i).

In our revised manuscript, we revised the section to (lines 342 – 349):

“Nineteen subclusters, including *Etv4*⁺, were found to have correspondences between the brachial and lumbar segments (**Fig. 7f; Supplementary Fig. 11c**), but five and two LMC subclusters were specific to the brachial or lumbar segment, respectively. For example, the *Grp*⁺ cl3 and rl5 subclusters are present in the brachial segment alone (**Fig. 7g; Supplementary Fig. 11d**). Although *Nkx6.1* is known to be expressed in certain *Isl1*⁺ LMC MNs in brachial and lumbar segments³, we further identified a *Nkx6.1*⁺ *Isl1*⁻ subpopulation, presumably the *crl6* cluster, which is only located in the caudal part of the lumbar segment (**Fig. 7g-7i**).”

Is each point one section in 7d? If so, please could the authors show the data by animal.

Response:

The data points were shown as per animal, and we have now standardized the data to n = 10 embryos per MMC subtype.

For clarity, we have also revised our figure legend as follows:

Quantification of subtype ratio in *Mecom*⁺ MMC neurons in brachial E13.5 spinal cord. Results are shown as mean ± SD from n = 10 embryos for each MMC subtype. Each embryo is represented by average counts from ≥ 3 sections.

Figure 7g: the yellow and orange arrows are much too small to see. Please provide higher magnification.

Response:

We now provide a higher resolution image and quantification of *Nrp2* transcript expression in MMC subtypes (*Satb2*⁺ and *Nr2f2*⁺ cells), as detected using RNAscope fluorescence *in situ* hybridization and 40x confocal imaging (**Fig. 8g and 8h**).

Lines 347-350: “Collectively, our results indicate that, similar to LMC MNs, MMC subtypes exhibit preferential mediolateral distributions and they selectively express TFs, receptors and axon guidance molecules that might facilitate high fidelity axonal innervations to the correct epaxial muscles.”

This interpretation sounds reasonable, but the data shown in Fig 7h are not very compelling. From the figure, it looks like *Unc5c* is also present in the *Nrp2*⁺ branch. Also, the pink line tracing *Nrp2* obscures the actual signal, which is not ideal.

Response:

We think that the co-expressing axonal branch might reflect the *Bcl11b*⁺ MMC subcluster. According to our single-cell dataset, *Unc5c* and *Nrp2* were both detected in the *Bcl11b*⁺ MMC

subcluster (**Fig. 8f**). However, further experimental validations of combinatorial markers are necessary, as the *Bcl11b*⁺ subcluster possesses fewer specific markers.

We now replace Fig. 8i with higher resolution image and use arrows to indicate differences in receptor expression in axial motor nerve branches.

Figure 8c: This figure would be stronger by showing *Nr2f2* + *Satb2* together in same image. If there is a reason why this was not shown please explain why this is not feasible.

Response:

We performed *Satb2* immunostaining after *in situ* hybridization of *Nr2f2* on the same spinal cord section. Protein co-immunostaining was initially carried out with commercially available *Nr2f2* antibody, but it did not work on chicken spinal cord, so we carried out *Nr2f2 in situ* hybridization.

Figure 8f-g: higher magnification for these images is required to see the cells indicated by the arrows.

Response:

Agreed. We have now replaced them with higher magnification images in our revised manuscript (**Fig. 9f, 9g**).

- 1 Hounkpe, B. W., Chenou, F., de Lima, F. & De Paula, E. V. HRT Atlas v1.0 database: redefining human and mouse housekeeping genes and candidate reference transcripts by mining massive RNA-seq datasets. *Nucleic Acids Res* **49**, D947-D955, doi:10.1093/nar/gkaa609 (2021).
- 2 Zuccaro, E., Piol, D., Basso, M. & Pennuto, M. Motor Neuron Diseases and Neuroprotective Peptides: A Closer Look to Neurons. *Front Aging Neurosci* **13**, 723871, doi:10.3389/fnagi.2021.723871 (2021).
- 3 De Marco Garcia, N. V. & Jessell, T. M. Early motor neuron pool identity and muscle nerve trajectory defined by postmitotic restrictions in *Nkx6.1* activity. *Neuron* **57**, 217-231, doi:10.1016/j.neuron.2007.11.033 (2008).
- 4 Arber, S., Ladle, D. R., Lin, J. H., Frank, E. & Jessell, T. M. ETS gene *Er81* controls the formation of functional connections between group Ia sensory afferents and motor neurons. *Cell* **101**, 485-498, doi:10.1016/s0092-8674(00)80859-4 (2000).
- 5 Mendelsohn, A. I., Dasen, J. S. & Jessell, T. M. Divergent Hox Coding and Evasion of Retinoid Signaling Specifies Motor Neurons Innervating Digit Muscles. *Neuron* **93**, 792-805.e794, doi:10.1016/j.neuron.2017.01.017 (2017).
- 6 Livet, J. *et al.* ETS gene *Pea3* controls the central position and terminal arborization of specific motor neuron pools. *Neuron* **35**, 877-892, doi:10.1016/s0896-6273(02)00863-2 (2002).

- 7 Korsunsky, I. *et al.* Fast, sensitive and accurate integration of single-cell data with Harmony. *Nat Methods* **16**, 1289-1296, doi:10.1038/s41592-019-0619-0 (2019).
- 8 Lin, J. H. *et al.* Functionally related motor neuron pool and muscle sensory afferent subtypes defined by coordinate ETS gene expression. *Cell* **95**, 393-407 (1998).
- 9 Stifani, N. Motor neurons and the generation of spinal motor neuron diversity. *Front Cell Neurosci* **8**, 293, doi:10.3389/fncel.2014.00293 (2014).
- 10 Becht, E. *et al.* Dimensionality reduction for visualizing single-cell data using UMAP. *Nat Biotechnol*, doi:10.1038/nbt.4314 (2018).
- 11 Caldeira, V., Dougherty, K. J., Borgius, L. & Kiehn, O. Spinal Hb9::Cre-derived excitatory interneurons contribute to rhythm generation in the mouse. *Sci Rep* **7**, 41369, doi:10.1038/srep41369 (2017).
- 12 Li, C. J. *et al.* MicroRNA filters Hox temporal transcription noise to confer boundary formation in the spinal cord. *Nat Commun* **8**, 14685, doi:10.1038/ncomms14685 (2017).
- 13 Li, C. J. *et al.* MicroRNA governs bistable cell differentiation and lineage segregation via a noncanonical feedback. *Mol Syst Biol* **17**, e9945, doi:10.15252/msb.20209945 (2021).
- 14 Kharchenko, P. V., Silberstein, L. & Scadden, D. T. Bayesian approach to single-cell differential expression analysis. *Nat Methods* **11**, 740-742, doi:10.1038/nmeth.2967 (2014).
- 15 Tung, Y. T. *et al.* Mir-17 approximately 92 Governs Motor Neuron Subtype Survival by Mediating Nuclear PTEN. *Cell Rep* **11**, 1305-1318, doi:10.1016/j.celrep.2015.04.050 (2015).
- 16 Perkel, J. M. Single-cell analysis enters the multiomics age. *Nature* **595**, 614-616, doi:10.1038/d41586-021-01994-w (2021).
- 17 Jin, S., Zhang, L. & Nie, Q. scAI: an unsupervised approach for the integrative analysis of parallel single-cell transcriptomic and epigenomic profiles. *Genome Biol* **21**, 25, doi:10.1186/s13059-020-1932-8 (2020).
- 18 Patel, T., Hammelman, J., Closser, M., Gifford, D. K. & Wichterle, H. General and cell-type-specific aspects of the motor neuron maturation transcriptional program. *bioRxiv*, 2021.2003.2005.434185, doi:10.1101/2021.03.05.434185 (2021).
- 19 Ozel, M. N. *et al.* Neuronal diversity and convergence in a visual system developmental atlas. *Nature* **589**, 88-95, doi:10.1038/s41586-020-2879-3 (2021).
- 20 Blum, J. A. *et al.* Single-cell transcriptomic analysis of the adult mouse spinal cord reveals molecular diversity of autonomic and skeletal motor neurons. *Nat Neurosci* **24**, 572-583, doi:10.1038/s41593-020-00795-0 (2021).
- 21 Alkaslasi, M. R. *et al.* Single nucleus RNA-sequencing defines unexpected diversity of cholinergic neuron types in the adult mouse spinal cord. *Nat Commun* **12**, 2471, doi:10.1038/s41467-021-22691-2 (2021).
- 22 Tang, K., Rubenstein, J. L., Tsai, S. Y. & Tsai, M. J. COUP-TFII controls amygdala patterning by regulating neuropilin expression. *Development* **139**, 1630-1639, doi:10.1242/dev.075564 (2012).
- 23 Alcamo, E. A. *et al.* Satb2 regulates callosal projection neuron identity in the developing cerebral cortex. *Neuron* **57**, 364-377, doi:10.1016/j.neuron.2007.12.012 (2008).

- 24 Amin, N. D. *et al.* A hidden threshold in motor neuron gene networks revealed by modulation of miR-218 dose. *Neuron* **109**, 3252-3267 e3256, doi:10.1016/j.neuron.2021.07.028 (2021).
- 25 Dasen, J. S., Tice, B. C., Brenner-Morton, S. & Jessell, T. M. A Hox regulatory network establishes motor neuron pool identity and target-muscle connectivity. *Cell* **123**, 477-491, doi:10.1016/j.cell.2005.09.009 (2005).
- 26 Salim, A. *et al.* RUV-III-NB: normalization of single cell RNA-seq data. *Nucleic Acids Res*, doi:10.1093/nar/gkac486 (2022).
- 27 Tyler, S. R., Bunyavanich, S. & Schadt, E. E. PMD Uncovers Widespread Cell-State Erasure by scRNAseq Batch Correction Methods. *bioRxiv*, 2021.2011.2015.468733, doi:10.1101/2021.11.15.468733 (2021).
- 28 Zhang, L. & Nie, Q. scMC learns biological variation through the alignment of multiple single-cell genomics datasets. *Genome Biol* **22**, 10, doi:10.1186/s13059-020-02238-2 (2021).
- 29 Luecken, M. D. *et al.* Benchmarking atlas-level data integration in single-cell genomics. *Nat Methods* **19**, 41-50, doi:10.1038/s41592-021-01336-8 (2022).
- 30 Hirono, K., Kohwi, M., Clark, M. Q., Heckscher, E. S. & Doe, C. Q. The Hunchback temporal transcription factor establishes, but is not required to maintain, early-born neuronal identity. *Neural Dev* **12**, 1, doi:10.1186/s13064-017-0078-1 (2017).
- 31 Catela, C., Shin, M. M., Lee, D. H., Liu, J. P. & Dasen, J. S. Hox Proteins Coordinate Motor Neuron Differentiation and Connectivity Programs through Ret/Gfralpha Genes. *Cell Rep* **14**, 1901-1915, doi:10.1016/j.celrep.2016.01.067 (2016).
- 32 Luria, V., Krawchuk, D., Jessell, T. M., Laufer, E. & Kania, A. Specification of motor axon trajectory by ephrin-B:EphB signaling: symmetrical control of axonal patterning in the developing limb. *Neuron* **60**, 1039-1053, doi:10.1016/j.neuron.2008.11.011 (2008).
- 33 Shirasaki, R., Lewcock, J. W., Lettieri, K. & Pfaff, S. L. FGF as a target-derived chemoattractant for developing motor axons genetically programmed by the LIM code. *Neuron* **50**, 841-853, doi:10.1016/j.neuron.2006.04.030 (2006).
- 34 Price, S. R., De Marco Garcia, N. V., Ranscht, B. & Jessell, T. M. Regulation of motor neuron pool sorting by differential expression of type II cadherins. *Cell* **109**, 205-216, doi:10.1016/s0092-8674(02)00695-5 (2002).
- 35 Demireva, E. Y., Shapiro, L. S., Jessell, T. M. & Zampieri, N. Motor neuron position and topographic order imposed by beta- and gamma-catenin activities. *Cell* **147**, 641-652, doi:10.1016/j.cell.2011.09.037 (2011).
- 36 Bello, S. M., Millo, H., Rajebhosale, M. & Price, S. R. Catenin-dependent cadherin function drives divisional segregation of spinal motor neurons. *J Neurosci* **32**, 490-505, doi:10.1523/JNEUROSCI.4382-11.2012 (2012).
- 37 Dewitz, C., Duan, X. & Zampieri, N. Organization of motor pools depends on the combined function of N-cadherin and type II cadherins. *Development* **146**, doi:10.1242/dev.180422 (2019).
- 38 Kupari, J. *et al.* Single cell transcriptomics of primate sensory neurons identifies cell types associated with chronic pain. *Nat Commun* **12**, 1510, doi:10.1038/s41467-021-21725-z (2021).

- 39 Zhong, W. *et al.* The neuropeptide landscape of human prefrontal cortex. *Proc Natl Acad Sci U S A* **119**, e2123146119, doi:10.1073/pnas.2123146119 (2022).
- 40 Catela, C. & Kratsios, P. Transcriptional mechanisms of motor neuron development in vertebrates and invertebrates. *Dev Biol* **475**, 193-204, doi:10.1016/j.ydbio.2019.08.022 (2021).
- 41 Nusbaum, M. P., Blitz, D. M. & Marder, E. Functional consequences of neuropeptide and small-molecule co-transmission. *Nat Rev Neurosci* **18**, 389-403, doi:10.1038/nrn.2017.56 (2017).
- 42 Zeisel, A. *et al.* Molecular Architecture of the Mouse Nervous System. *Cell* **174**, 999-1014.e1022, doi:10.1016/j.cell.2018.06.021 (2018).
- 43 Häring, M. *et al.* Neuronal atlas of the dorsal horn defines its architecture and links sensory input to transcriptional cell types. *Nat Neurosci* **21**, 869-880, doi:10.1038/s41593-018-0141-1 (2018).

REVIEWER COMMENTS

Reviewer #1 (Remarks to the Author):

The authors thoroughly addressed most of my comments.

Regarding Fig. 7c: I thank the authors for adding this panel as a response to my comment ("It would be nice to see what the rm/cm clusters look like on the UMAP.."). However, I think I did not communicate clearly. Although general rostral/caudal identity is useful, I would have liked to see the actual rm/cm cluster identities plotted on the UMAP, e.g., cl1, cl2, etc. A single-cell visualization to supplement the cluster-level heatmap and alluvial diagrams. The conclusions of this figure nevertheless seem sound.

Reviewer #2 (Remarks to the Author):

The authors have addresses all my comments from the previous round of revisions. This is a well executed study that makes interesting observations regarding the establishment of motor neuron diversity. Conclusions throughout the study are well supported by the underlying data and the provided single cell sequencing data will be a valuable resource for the scientific community in the coming years. I have couple of small comments/questions that the authors should clarify before publication.

1) line 132 '... and consequently regarded as false-negative cells.' I am not sure what the authors want to say here. Which cells are regarded as false-negative due to sparse Mnx1 expression?

2) lines 209-210 'We observed that PGC MNs also exhibit subtypes (Fig. 1c; Supplementary Fig. 4c)'. Based on which criteria are there distinct PGC subtypes?

3) lines 347-349 '..., we further identified a Nkx6.1+ Isl1- subpopulation, presumably the crl6 cluster, which is only located in the caudal part of the lumbar segment (Fig. 7g-7i).' Based on which criteria where these Nkx6.1+ Isl1- cells identified as motor neurons?

4) lines 364-365 '... whereas some Bcl11b+ MMC MNs were co-expressed with either Nr2f2 or Satb2 (~15% each) and shared their subtype transcriptomes (Fig. 8c; Supplementary Fig. 12a, 12b).' In the heatmap in Figure S12b both the Bcl11b Satb2 and Bcl11b Nr2f2 neurons appear much more similar to Bcl11b neurons than to either Satb2 or Nr2f2 only neurons. Why are the authors saying that these double-positive neurons share their subtype transcriptomes with Nr2f2 and Satb2 neurons? Also, can the authors introduce little spaces between the individual subgroups in the heatmap in Supplementary Figure 12b to make the different subgroups more readily visible?

5) The mediolateral positioning of the SATB2 and NR2F2 MMC neurons seems to be inverted between mice and human. In mice and chicken SATB2 neurons appear more medial (e.g. Figures 8c and e and 9c) while in Figure 10i SATB2 seems to be more lateral (or at least more towards the right side of the image). Is this correct or simply due to a recording of the other (left) side of the spinal cord? If reproducible, this may be interesting to point out, given the importance of motor neuron positioning for the correct establishment of neuronal circuits in other motor neuron subtypes.

6) While the authors have deposited some of their code on Github, their repository does not include the complete code to reproduce the data analysis. In fact, most of the code snippets provided on GitHub rely on precomputed R data files. While the details about preprocessing and clustering are provided in the Materials and Methods section, given that the single cell RNAseq dataset and its analysis are at the core of this manuscript I think it would facilitate the re-use of the data if the authors make their entire code publically available or at least provide the expression matrix and all metadata obtained

from the analyses described in the manuscript via GEO.

7) Supplementary Figure 5a,b: It is sometimes unclear to which dots on the plot the labels belong. Using small lines to connect the dots and labels as they have done for Hoxc4 in Supplementary Figure 5a could be used to solve this issue.

Reviewer #3 (Remarks to the Author):

The authors have done a fantastic job addressing all my comments with the addition of carefully executed experiments and rephrasing to add clarity to their text. Every single comment was rigorously addressed and/or discussed. They have provided high quality images demonstrating co-expression of new and established markers and quantifications to show distribution of cell types along the rostral-caudal axis that really help validate the conclusions they had drawn from their RNAseq data. I agree with their discussion of the potential shortcomings of single cell data integration; this is a welcome addition. I am also really impressed with the authors' attempt at exceedingly difficult experiments to retrogradely label specific motor neuron pools from axial muscle injections in the embryo. Congratulations to the authors on an excellent manuscript! This work was a pleasure to read and will be a really useful resource for the community.

Reviewer #1:

The authors thoroughly addressed most of my comments.

Regarding Fig. 7c: I thank the authors for adding this panel as a response to my comment ("It would be nice to see what the rm/cm clusters look like on the UMAP.."). However, I think I did not communicate clearly. Although general rostral/caudal identity is useful, I would have liked to see the actual rm/cm cluster identities plotted on the UMAP, e.g., cl1, cl2, etc. A single-cell visualization to supplement the cluster-level heatmap and alluvial diagrams. The conclusions of this figure nevertheless seem sound.

Response:

Apologies that we misunderstood the original suggestion. We have now colored the cells in the UMAP (Figure 7a) based on the individual subclusters and revised the figure legend accordingly.

Reviewer #2:

The authors have addresses all my comments from the previous round of revisions. This is a well executed study that makes interesting observations regarding the establishment of motor neuron diversity. Conclusions throughout the study are well supported by the underlying data and the provided single cell sequencing data will be a valuable resource for the scientific community in the coming years. I have couple of small comments/questions that the authors should clarify before publication.

1) line 132 '... and consequently regarded as false-negative cells.' I am not sure what the authors want to say here. Which cells are regarded as false-negative due to sparse *Mnx1* expression?

Response:

Apologies for the confusing term. We have now clarified the sentence as follows (lines 130-132):

Although *Mnx1* labels MNs and a population of spinal glutamatergic interneurons (IN) specifically in the spinal cord, we detected sparse *Mnx1* expression in our single-cell dataset, likely due to a limitation of scRNA-seq sensitivity.

2) lines 209-210 'We observed that PGC MNs also exhibit subtypes (Fig. 1c; Supplementary Fig. 4c)'. Based on which criteria are there distinct PGC subtypes?

Response:

We apologize for the confusion. We have now added the subclustering results for PGC MNs based on transcriptome similarities from our single-cell dataset in **Supplementary Fig. 4c**. This result is consistent with observations from a previous study¹, which indicated that differential *Isl1* expression could distinguish PGCa and PGCb subtypes. Accordingly, we have revised our sentences as follows (lines 209-213):

In agreement with observations from a previous study reporting that differential *Isl1* expression could differentiate PGC MNs into PGCa and PGCb subtypes³³, we observed a similar pattern in our single-cell data (**Fig. 1c**; **Supplementary Fig. 4c**). However, in this study, our objective was to analyze LMC and MMC MN subtypes, so we did not endeavor to characterize the PGC subclusters further.

3) lines 347-349 '... we further identified a *Nkx6.1*⁺ *Isl1*⁻ subpopulation, presumably the *cr16* cluster, which is only located in the caudal part of the lumbar segment (Fig. 7g-7i).' Based on which criteria were these *Nkx6.1*⁺ *Isl1*⁻ cells identified as motor neurons?

Response:

We consider the *Nkx6.1*⁺ *Isl1*⁻ cells to be LMC MNs based on their co-expression of the *Foxp1* marker (a known LMC marker). In these figures, we outlined the LMC region by a circle based on the distribution of *Foxp1*-expressing cells (**Fig. R1**), but we only showed *Nkx6.1*/*Isl1* immunostaining results for clarity. To make this point clearer, we have now added the following statement to the figure legend of Fig. 7h:

Dashed lines outline the spinal cord boundary, and LMC MNs are demarcated by circles based on *Foxp1* expression.

4) lines 364-365 '... whereas some *Bcl11b*⁺ MMC MNs were co-expressed with either *Nr2f2* or *Satb2* (~15% each) and shared their subtype transcriptomes (Fig. 8c; Supplementary Fig. 12a, 12b).' In the heatmap in Figure S12b both the *Bcl11b* *Satb2* and *Bcl11b* *Nr2f2* neurons appear much more similar to *Bcl11b* neurons than to either *Satb2*

or Nr2f2 only neurons. Why are the authors saying that these double-positive neurons share their subtype transcriptomes with Nr2f2 and Satb2 neurons? Also, can the authors introduce little spaces between the individual subgroups in the heatmap in Supplementary Figure 12b to make the different subgroups more readily visible?

Response:

We appreciate the reviewer's comment on this issue. We have now revised the heatmap in **Supplementary Fig. 12b** by introducing space to separate the MMC subtypes. In our previous revision, we responded to reviewer #1 that no specific marker genes were identified for the Bcl11b⁺Satb2⁺ or Bcl11b⁺Nr2f2⁺ clusters, respectively (**Supplementary Fig. 12b**). Therefore, we do not consider these double-positive cells as being independent subclusters. We apologize for the confusing sentences. We only intended to suggest that the Bcl11b⁺Satb2⁺ and Bcl11b⁺Nr2f2⁺ co-expressing cells do not manifest salient hallmarks, rather than displaying a similarity to the Bcl11b, Satb2, or Nr2f2 subtypes.

Thus, we have revised the sentences as follows (lines 364-367):

Satb2 and Nr2f2 were expressed in MMC MNs in a mutually exclusive manner (**Fig. 8c**). Although some Bcl11b⁺ cells were co-expressed with Satb2 or Nr2f2, these double-positive cells (Bcl11b⁺Satb2⁺ or Bcl11b⁺Nr2f2⁺) did not manifest salient hallmarks (**Fig. 8c; Supplementary Fig. 12a, 12b**).

5) The mediolateral positioning of the SATB2 and NR2F2 MMC neurons seems to be inverted between mice and human. In mice and chicken SATB2 neurons appear more medial (e.g. Figures 8c and e and 9c) while in Figure 10i SATB2 seems to be more lateral (or at least more towards the right side of the image). Is this correct or simply due to a recording of the other (left) side of the spinal cord? If reproducible, this may be interesting to point out, given the importance of motor neuron positioning for the correct establishment of neuronal circuits in other motor neuron subtypes.

Response:

The reviewer has made a sharp observation. We were intrigued when we first noted this same phenomenon. Unfortunately, due to an IRB agreement issue and difficulties in sourcing human embryos over the past three years due to the COVID pandemic, we could not get additional early-stage human embryos to properly quantify and verify this possibility. We hope to collect other data in future to test this hypothesis, but for now we mention this point in the Discussion as follows (lines 545-550):

“Interestingly, we observed that the SATB2 and NR2F2 mediolateral positioning we observed in mouse embryos might be inverted in humans. Since the transcriptional program underlying MMC subtype identities might be critical to MMC subtype topographical organization, additional experiments on human embryos will be necessary in future to investigate if this inversion leads to a reorganization of the MMC topographical distribution and whether this change is functionally important.”

6) While the authors have deposited some of their code on Github, their repository does not include the complete code to reproduce the data analysis. In fact, most of the code snippets provided on GitHub rely on precomputed R data files. While the details about preprocessing and clustering are provided in the Materials and Methods section, given that the single cell RNAseq dataset and its analysis are at the core of this manuscript I think it would facilitate the re-use of the data if the authors make their entire code publically available or at least provide the expression matrix and all metadata obtained from the analyses described in the manuscript via GEO.

Response:

Thank you for this reminder. We have now provided all related metadata, including both cell annotations and UMAP coordinates, in the public GitHub repository (<https://github.com/sqjin/MotorNeuron>) and the Gene Expression Omnibus repository (GEO accession number: GSE183759). The GEO repository will be released for download upon publication.

7) Supplementary Figure 5a,b: It is sometimes unclear to which dots on the plot the labels belong. Using small lines to connect the dots and labels as they have done for Hoxc4 in Supplementary Figure 5a could be used to solve this issue.

Response:

Agreed. We have now updated the figure as suggested.

Reviewer #3:

The authors have done a fantastic job addressing all my comments with the addition of carefully executed experiments and rephrasing to add clarity to their text. Every single comment was rigorously addressed and/or discussed. They have provided high quality images demonstrating co-expression of new and established markers and quantifications to show distribution of cell types along the rostral-caudal axis that really help validate the conclusions they had drawn from their RNAseq data. I agree with their discussion of the potential shortcomings of single cell data integration; this is a welcome addition. I am also really impressed with the authors' attempt at exceedingly difficult experiments to retrogradely label specific motor neuron pools from axial muscle injections in the embryo. Congratulations to the authors on an excellent manuscript! This work was a pleasure to read and will be a really useful resource for the community.

Response:

We thank the reviewer for his/her insightful suggestions, which significantly improved the original manuscript.

- 1 Amin, N. D. *et al.* A hidden threshold in motor neuron gene networks revealed by modulation of miR-218 dose. *Neuron* **109**, 3252-3267 e3256, doi:10.1016/j.neuron.2021.07.028 (2021).

REVIEWERS' COMMENTS

Reviewer #1 (Remarks to the Author):

My points have now been addressed.

Reviewer #2 (Remarks to the Author):

The authors have completely addressed all my comments. I congratulate them to this extremely well executed study.